# Online robust locally differentially private learning for nonparametric regression

**Chenfei Gu[1]***  **Qiangqiang Zhang[2]***  **Ting Li[1]†**  **Jinhan Xie[3]†**  **Niansheng Tang[3]**

[1]School of Statistics and Data Science, Shanghai University of Finance and Economics
[2]Zhongtai Securities Institute for Financial Studies, Shandong University
[3]Yunnan Key Laboratory of Statistical Modeling and Data Analysis, Yunnan University
gu.chenfei@live.sufe.edu.cn, qiangqiangzhang@mail.sdu.edu.cn,
tingli@mail.shufe.edu.cn, {jinhanxie, nstang}@ynu.edu.cn

## Abstract

The growing prevalence of streaming data and increasing concerns over data privacy pose significant challenges for traditional nonparametric regression methods, which are often ill-suited for real-time, privacy-aware learning. In this paper, we tackle these issues by first proposing a novel one-pass online functional stochastic gradient descent algorithm that leverages the Huber loss (H-FSGD), to improve robustness against outliers and heavy-tailed errors in dynamic environments. To further accommodate privacy constraints, we introduce a locally differentially private extension, Private H-FSGD (PH-FSGD), designed to real-time, privacy-preserving estimation. Theoretically, we conduct a comprehensive non-asymptotic convergence analysis of the proposed estimators, establishing finite-sample guarantees and identifying optimal step size schedules that achieve optimal convergence rates. In particular, we provide practical insights into the impact of key hyperparameters, such as step size and privacy budget, on convergence behavior. Extensive experiments validate our theoretical findings, demonstrating that our methods achieve strong robustness and privacy protection without sacrificing efficiency.

## 1   Introduction

Nonparametric regression, which models the relationship between a response variable and its predictors without imposing a specific functional form, is a fundamental tool in statistical data analysis. It has been extensively studied over the past several decades (e.g., Siegel [1957], Härdle [1990], Wasserman and Lafferty [2005], Takezawa [2005]) and is particularly well-suited for capturing complex and nonlinear structures in data. More recently, nonparametric modeling has provided powerful insights into complex and dynamic systems across a range of applications, including deep learning [Schmidt-Hieber, 2020, Zhang and Wang, 2024], climatology [Huth and Pokorná, 2004, Deb and Jana, 2024], and economics [Donnelly et al., 2011, Salibian-Barrera, 2023].

Traditional nonparametric regression methods typically assume full access to the entire dataset beforehand and require it to be stored entirely in memory. Within this batch learning framework, model estimation is conducted only once based on the full dataset. However, this paradigm faces substantial limitations in streaming data environments, where observations arrive sequentially and continuously over time. In such settings, storing and processing the entire data stream simultaneously is often infeasible. For example, data generated in real time by autonomous vehicles or large-scale sensor networks in smart cities accumulate rapidly and far exceed the capacity of available memory

---

*Equal contribution.
†Corresponding authors: Ting Li and Jinhan Xie.

39th Conference on Neural Information Processing Systems (NeurIPS 2025).

resources. In contrast to classical batch learning, online learning methods are designed to dynamically update model estimates using only the currently available data, thereby enabling real-time decision-making in nonparametric regression. To date, such methods have been extensively studied in the literature; see, for example, Gu and Lafferty [2012], Huang et al. [2013], Kuzborskij and Cesa-Bianchi [2017], Xue and Yao [2022], Yang et al. [2024], Quan and Lin [2024]. Beyond these approaches, a line of work has focused on functional stochastic gradient descent (FSGD) approximation algorithms developed within the framework of reproducing kernel Hilbert spaces (RKHS) or more general Hilbert spaces; see Kivinen et al. [2004], Dieuleveut and Bach [2016], Zhang and Simon [2022], Liu et al. [2023], Zhang and Simon [2023], Chen and Klusowski [2024], Fonseca et al. [2024]. Nevertheless, a common limitation across the above literature is the implicit assumption of unrestricted access to raw individual, level data throughout the learning process.

As data complexity and volume continue to grow, so do the challenges associated with safeguarding individual privacy and maintaining public trust, particularly in applications involving potentially sensitive user data, such as patient records in healthcare or behavioral logs in e-commerce platforms. Differential privacy (DP), one of the most widely adopted frameworks [Dwork et al., 2006a,b], provides rigorous guarantees that the output of a statistical analysis does not reveal sensitive information about any individual in the dataset. Owing to its rigorous mathematical definitions and practical applicability, DP has been successfully applied in numerous fields, including medical imaging, healthcare analytics, and intelligent transportation systems [Dankar and El Emam, 2013, Ziller et al., 2024, Bhadani, 2024]. In the literature, two primary variants of DP have been extensively studied: the central differential privacy (CDP) model assumes the existence of a trusted server that can securely collect, store, and process raw data from users, and the local differential privacy (LDP) model eliminates the need for such a trusted entity by requiring each user's data to be privatized at the source, before being transmitted to any data aggregator or processor (see e.g. Dwork et al. [2014], Duchi et al. [2018], Berrett and Yu [2021], Li et al. [2023], Duchi and Ruan [2024]). However, privacy protection inevitably introduces tension with two other key objectives: model robustness against adversarial perturbations and statistical utility. This fundamental trade-off, known as the privacy-robustness-utility trilemma, has been extensively studied across different learning paradigms, including distributed learning [Allouah et al., 2023], adversarial learning with certified guarantees [Phan et al., 2020], and decentralized Byzantine-robust systems [Ye et al., 2024]. Although substantial progress has been made under both paradigms, most existing methods are dedicated to finite-dimensional learning problems such as fitting a parametric regression model.

Recently, increasing attention has been directed toward privacy-preserving estimation in infinite-dimensional settings, where either the inputs, outputs, or both are functions in nature. Most existing work in this area has been developed under the CDP framework. For example, Hall et al. [2013] proposed to add an appropriately calibrated Gaussian process to release functional data while preserving privacy. Building this idea, Mirshani et al. [2019] developed the Gaussian mechanism to a more general framework capable of releasing a broad class of functional estimators. Reimherr and Awan [2019] further generalized this line of work by introducing privacy mechanisms based on centered elliptical processes. In parallel, Awan et al. [2019] studied the exponential mechanism in separable Hilbert spaces, with applications in functional data analysis, shape analysis, and nonparametric statistics. More recently, Lin and Reimherr [2024] introduced the independent component Laplace process mechanism to achieve pure DP for functional summaries in separable infinite-dimensional Hilbert spaces. Cai et al. [2024] examined the statistical optimality of federated nonparametric regression under DP constraints. In cases where the infinite-dimensional functional space can be effectively approximated by a finite number of basis functions, several methods have utilized the post-processing property of DP to design privacy-preserving procedures. For example, Cai et al. [2023] introduced CDP-based techniques for nonparametric regression under basis-function representations. Extending this line of work, Xue et al. [2024] established statistically optimal estimation procedures for distributed functional mean estimation and varying coefficient models under a variety of DP frameworks. Despite these advances, existing approaches are primarily designed for batch learning and often rely on a trusted data curator. To the best of our knowledge, no scalable and statistically sound method has yet been developed to perform online private nonparametric regression under the LDP framework. This gap leads to the following fundamental question:

*Can one design an online, private, nonparametric regression algorithm that is robust to heavy-tailed noise and simultaneously satisfies LDP, without compromising statistical efficiency?*

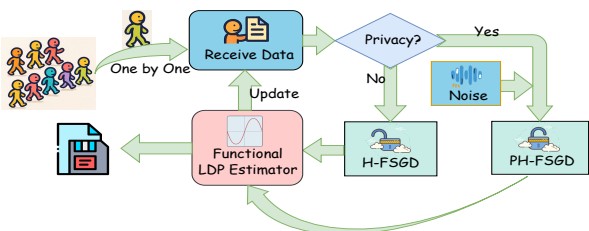

Figure 1: Flowchart of the proposed online robust privacy-preserving estimation framework. Data is received one by one, optionally perturbed with noise for privacy, and then used to update a functional LDP estimator via either H-FSGD or PH-FSGD.

The main goal of this paper is to address the question outlined above. To this end, we develop a fully online robust LDP framework for real-time estimation in nonparametric regression. Specifically, we introduce two novel algorithms, i.e., *H-FSGD* and *PH-FSGD*, that enable efficient and privacy-preserving learning in streaming data environments. In contrast to minimizer-optimal loss alignment approaches [Allouah et al., 2023] for addressing the privacy–robustness–utility trilemma, our method achieves inherent outlier robustness via a Huber loss framework. A flowchart illustrating the structure of the proposed framework is provided in Figure 1. A comparative summary of our method against representative recent works in nonparametric regression is provided in Table 1. For brevity, we include one example from each category of related methods. Our main contribution can be summarized as follows:

- **Online robust LDP estimation framework**: Our framework provides rigorous per-iteration LDP guarantees for nonparametric regression in an online setting, addressing a key limitation of existing methods that typically require access to the entire dataset. By incorporating Huber loss into our framework, the proposed algorithms attain robustness to outliers and heavy-tailed errors, thereby enabling robust privacy-preserving real-time estimation in dynamic environments.

- **One-pass algorithms**: The proposed algorithms are both designed to operate in an online, one-pass manner, yielding LDP estimators with $O(1)$ time and space complexity per iteration for nonparametric regression. By eliminating the need to store or re-access historical data, our approach avoids the $O(n)$ computational and memory overhead typically associated with maintaining past kernel evaluations. This design enables high computational efficiency and inherent scalability, making the algorithms particularly well-suited for large-scale or streaming data environments.

- **Non-asymptotic analysis**: We systematically establish non-asymptotic convergence rates for our estimators, with or without LDP, under both constant and decaying step-size regimes. Our analysis operates in a general framework that recovers the best approximation of the true function within the RKHS. The convergence rate depends on the sample size, step size, and the smoothness of both the RKHS and the original space containing the best approximation. Specifically, under a constant step-size scheme, the proposed estimators attain the minimax optimal rate not only when the original function space matches or is smoother than the estimation space, but also in certain cases when it is less smooth.

Table 1: A comparison of recent results on nonparametric regression.

| Method | Online | One-pass | Robust | Optimal rate | Privacy |
|---|---|---|---|---|---|
| Hall et al. [2013] | ✗ | ✗ | ✗ | ? | ✓ |
| Dieuleveut and Bach [2016] | ✓ | ✓ | ✗ | ✓ | ✗ |
| Liu et al. [2023] | ✓ | ✓ | ✗ | ✓ | ✗ |
| Quan and Lin [2024] | ✓ | ✓ | ✗ | ✓ | ✗ |
| Proposed | ✓ | ✓ | ✓ | ✓ | ✓ |

## 2    Problem formulation

We observe an independently and identically distributed stream of data $\{(X_n, Y_n)\}_{n=1}^{\infty}$ generated from the regression model

$$Y_n = f^{\star}(X_n) + e_n,$$

where $X_n \in \mathcal{X}$ is the $n$-th copy of the covariate $X$ with marginal distribution $P_X$, $Y_n \in \mathbb{R}$ is the response, $e_n$ is the noise with $\mathbb{E}(e) = 0$, $\mathrm{Var}(e) = \sigma^2$ and $\mathrm{Cov}(e, X) = 0$, and the regression function $f^{\star}$ belongs to $L^2(P_X)$ with $\|f^{\star}\|_{\infty} < \infty$. For our theoretical analysis, we assume finite-variance noise to derive tractable bounds on the gradient variance. This assumption encompasses many common heavy-tailed distributions, such as the Student–$t$ distribution with degrees of freedom $\nu > 2$, the Laplace distribution, and the symmetric Pareto distribution with shape parameter $\alpha > 2$. Empirically, however, our procedure remains robust even under infinite-variance conditions, such as Cauchy noise (see Subsection F.4).

Our objective is to estimate $f^{\star}$ in a streaming setting that operates in a single pass, where samples arrive sequentially and storing the entire dataset is infeasible. Online methods that incrementally process data are increasingly adopted for large-scale problems due to their computational and memory efficiency [Zhang and Lei, 2025]. At the same time, many application domains that generate streaming data, such as healthcare [Mohammed et al., 2013], medical records [Liu et al., 2024], and customer analytics [Hard et al., 2019], require formal privacy protection, which motivates the design of algorithms that provide provable privacy guarantees while remaining computationally efficient. These desiderata raise several intertwined challenges: the target is an infinite-dimensional function, yet the algorithm is restricted to a single update per sample, which rules out classical offline kernel methods requiring multiple passes and global optimization [Cai and Yuan, 2011]); most differential-privacy techniques are developed for finite-dimensional parameters [Dwork et al., 2014] and do not transfer straightforwardly to infinite-dimensional, single-pass function estimation; and heavy-tailed noise can make squared-loss-based online methods [Dieuleveut and Bach, 2016] unstable, necessitating robustness in the loss function.

To address robustness in the presence of heavy tails and to facilitate private updates, we consider Huber regression in an RKHS. Concretely, we study the population optimization

$$\min_{f \in \mathcal{H}} \ \mathbb{E}\, L_{\tau}\big(Y - f(X)\big), \tag{1}$$

where $L_{\tau}$ is the Huber loss with parameter $\tau > 0$:

$$L_{\tau}(u) = \tfrac{1}{2}u^2\, \mathbb{I}\{|u| \leq \tau\} + \Big(\tau|u| - \tfrac{1}{2}\tau^2\Big)\mathbb{I}\{|u| > \tau\}. \tag{2}$$

The Huber loss combines the efficiency of squared loss for small residuals with the robustness of absolute loss for large residuals; importantly for our setting, for fixed $\tau$ it yields uniformly bounded gradients, which both stabilizes online updates and simplifies the design of per-iteration noise under LDP without relying on ad-hoc gradient clipping (see (6) for gradient details).

We take $\mathcal{H}$ to be a RKHS on $\mathcal{X}$ with kernel $K(\cdot, \cdot)$ and inner product $\langle \cdot, \cdot \rangle_{\mathcal{H}}$. To allow for model misspecification we do not require $f^{\star} \in \mathcal{H}$; instead we target the best RKHS approximation

$$f_{\mathcal{H}} := \arg\min_{f \in \overline{\mathcal{H}}} \mathbb{E}\big[(Y - f(X))^2\big],$$

where $\overline{\mathcal{H}}$ denotes the closure of $\mathcal{H}$ in $L^2(P_X)$. The RKHS restriction is a standard device in nonparametric regression that converts the infinite-dimensional estimation problem into a tractable functional estimation framework while permitting a misspecified truth [Wahba, 1990, Dieuleveut and Bach, 2016, Zhang et al., 2023].

We place the following standard regularity conditions on the covariate distribution and the kernel.

**Assumption 1.** *Suppose that the distribution of $P_X$ has full support in $\mathcal{X}$.*

**Assumption 2.** *The kernel is continuous and uniformly bounded on the diagonal:* $\sup_{x \in \mathcal{X}} K(x, x) \leq B^2 < \infty$ *for some $B > 0$.*

Assumptions 1–2 are common in nonparametric RKHS regression and ensure basic well-posedness of the estimation task [Cai and Yuan, 2011, Zhou et al., 2020, Liu and Li, 2023].

Since privacy is a core concern in our work, we adopt the LDP framework, which removes the need for a trusted curator by randomizing data at the user side prior to collection. Formally, for $\varepsilon > 0$ and $\delta \geq 0$, a randomized mechanism $M : \mathcal{X} \to \mathcal{Y}$ is $(\varepsilon, \delta)$-LDP if for any $x, x' \in \mathcal{X}$ and measurable $E \subset \mathcal{Y}$ it holds that

$$\mathbb{P}\big(M(x) \in E\big) \leq e^{\varepsilon}\mathbb{P}\big(M(x') \in E\big) + \delta,$$

where the probability is taken over the randomness of $M$ [Xiong et al., 2020]. Within the online RKHS framework, the LDP mechanism leverages the boundedness of Huber gradients to calibrate noise precisely to the sensitivity of each update, ensuring rigorous privacy guarantees while preserving statistical efficiency.

Our aim is to construct a computationally efficient, single-pass sequence of estimators $\{f_n\}_{n \geq 1} \subset \mathcal{H}$ that can be updated incrementally upon receipt of $(X_{n+1}, Y_{n+1})$ and that satisfies $\|\bar{f}_n - f_{\mathcal{H}}\|_{L^2(P_X)} \to 0$ as $n \to \infty$ in both the non-private and the LDP settings. The algorithms leverage the Huber loss to achieve robustness to heavy-tailed noise and to yield bounded per-iteration sensitivity suitable for LDP.

More background information regarding RKHS and DP is presented in Appendix B.

We introduce the notation used throughout the paper. Let $\{a_n\}_{n \geq 1}$ and $\{b_n\}_{n \geq 1}$ are two sequences of non-negative numbers. $a_n \lesssim b_n$ or $a_n = O(b_n)$ indicates that $a_n \leq Cb_n$ for some constant $C > 0$ independent of $n$. $a_n \gtrsim b_n$ indicates that $a_n \geq Cb_n$ for some constant $C > 0$ independent of $n$. $a_n \asymp b_n$ represents $a_n \lesssim b_n$ and $a_n \gtrsim b_n$. Denote $P_X$ as the distribution of $X$ over the space $\mathcal{X}$, and $L^2(P_X) = \{f : \mathcal{X} \to \mathbb{R}| \int_{\mathcal{X}} f(x)^2 dP_X(x) < \infty\}$. The norm $\|\cdot\|_{L^2_{P_X}}$ for $f \in L^2(P_X)$ is defined as $\|f\|^2_{L^2_{P_X}} = \int_{\mathcal{X}} f(x)^2 dP_X(x) = \mathbb{E}(f(X))$.

# 3 Methodology

In this section, we propose the online robust nonparametric estimation within the RKHS framework to the minimization problem (1), which is universal for non-privacy-preserving and privacy-preserving settings.

## 3.1 Robust functional SGD

We first propose a Huber functional stochastic gradient descent (H-FSGD) algorithm to solve (1) in the streaming data setting without DP. Inspired by functional SGD methods for squared loss [Liu et al., 2023], H-FSGD extends this approach to the Huber loss. Given an initial estimate $\hat{f}_0$ (e.g., $\hat{f}_0(\cdot) = \bar{f}_0(\cdot) = 0$), the estimate is recursively updated upon the arrival of each new sample $(X_n, Y_n)$ as follows:

$$\hat{f}_n = \hat{f}_{n-1} - \gamma_n \widehat{\nabla L_\tau}(\hat{f}_{n-1})(X_n, Y_n), \qquad \bar{f}_n = \frac{n-1}{n}\bar{f}_{n-1} + \frac{1}{n}\hat{f}_n, \tag{3}$$

where $\gamma_n$ is the step size, and $\widehat{\nabla L_\tau}$ is an estimator of the Fréchet gradient of the Huber loss evaluated at $(X_n, Y_n)$. This update generalizes classical SGD to the functional setting with a robust loss, and the Polyak average $\bar{f}_n$ further improves stability and accuracy by averaging over the update trajectory [Ruppert, 1988, Polyak and Juditsky, 1992].

By the reproducing property of the kernel $K$, any $f \in \mathcal{H}$ satisfies $f(X) = \langle f, K_X \rangle_{\mathcal{H}}$, where $K_X(\cdot) = K(X, \cdot)$. The Fréchet derivative of $\langle f, K_X \rangle_{\mathcal{H}}$ with respect to $f$ is $K_X$. Therefore, the gradient estimator of the Huber loss at a sample $(x, y)$ is given by

$$\widehat{\nabla L_\tau}(f)(x, y) = -\ell_\tau(y - f(x))K_x = -w_\tau(y - f(x))(y - f(x))K_x, \tag{4}$$

where $\ell_\tau(u) := uI\{|u| \leq \tau\} + \tau \cdot \text{sign}(u)I\{|u| > \tau\}$, and $w_\tau := \min\{1, \tau/|u|\}$. By utilizing the reproducing property $\hat{f}_{n-1}(X) = \langle \hat{f}_{n-1}, K_X \rangle_{\mathcal{H}}$, the recursion in (3) can be written as

$$\hat{f}_n = \hat{f}_{n-1} + \gamma_n w_\tau \left(Y_n - \langle \hat{f}_{n-1}, K_{X_n} \rangle_{\mathcal{H}}\right)\left(Y_n - \langle \hat{f}_{n-1}, K_{X_n} \rangle_{\mathcal{H}}\right)K_{X_n},$$

$$\bar{f}_n = \frac{n-1}{n}\bar{f}_{n-1} + \frac{1}{n}\hat{f}_n. \tag{5}$$

The proposed H-FSGD algorithm updates iteratively without storing historical raw data. In practice, we retain the current estimate $\hat{f}_{n-1}$ evaluated on a fixed grid $\{t_j\}_{j=1}^J$. Upon receiving the $n$-th sample $(X_n, Y_n)$, the prediction $\hat{f}_{n-1}(X_n)$ can be computed as $\langle \hat{f}_{n-1}, K_{X_n} \rangle_{\mathcal{H}}$, requiring only the current estimate $\hat{f}_{n-1}$ and $K_{X_n}$ calculated at the new $X_n$, which offers greater flexibility and lower memory usage.

The parameter $\tau$ in the Huber loss establishes a trade-off between robustness and bias in the estimation, which is consistent with classical literature [Fan et al., 2017]. In practice, we use a data-driven procedure to choose the parameter $\tau$, motivated by classical Huber loss methods [Holland and Welsch, 1977]. Firstly, obtain an initial estimator $\hat{f}^{\mathrm{LS}}$ of $f_{\mathcal{H}}$ via the existing least-squares functional SGD [Dieuleveut and Bach, 2016] and small samples. Secondly, calculate the prediction errors $\{\mathrm{resi}_i\}_i$ based on the estimator $\hat{f}^{\mathrm{LS}}$, and get the robust estimation of $\sigma$ via the median absolute deviation estimator, i.e.,

$$\hat{\sigma} = \mathrm{Median}\{|\mathrm{resi}_i|\}/0.6745.$$

Lastly, construct $\tau = 1.345\hat{\sigma}$, where $1.345$ is the default value in R package ('rlm' function) to achieve 90% efficiency for normally distributed noise.

We summarize the selection of $\tau$ in Algorithm 2, and the proposed H-FSGD in Algorithm 3. Please see Appendix C.

## 3.2 Robust locally differentially private functional SGD

Protecting sensitive information in real-time data streams is paramount to prevent unintended disclosures as each new observation is processed. Unlike Hall et al. [2013], which applies DP to the entire algorithm in a centralized manner, our approach integrates privacy protection into each iteration step. This design eliminates the need for a trusted data collector and achieves LDP by ensuring that data are privatized at the source before any aggregation occurs.

To ensure rigorous LDP guarantees, we augment Algorithm 3 by adding per-iteration Gaussian process noise. Under Assumption 2 and the definition of the Huber loss, for any $n \in \mathbb{N}$, and pair of input individual values $z = (x, y), z' = (x', y')$, we have

$$\sup_{z,z'} \|\widehat{\nabla L}(\hat{f}_{n-1})(x, y) - \widehat{\nabla L}(\hat{f}_{n-1})(x', y')\|_{\mathcal{H}} \leq 2\tau \sup_{x \in \mathcal{X}} \|K_x\|_{\mathcal{H}} \leq 2\tau B. \tag{6}$$

The quantity $2\tau B$ corresponds to the sensitivity in the standard LDP framework [Xiong et al., 2020]. Let $\xi_n$ be the sample path of a Gaussian process having mean zero and covariance function $\frac{8\tau^2 B^2 \log(2/\delta_n)}{\varepsilon_n^2} K$, where $(\varepsilon_n, \delta_n)$ is the privacy budget at the $n$-th iteration. Applying Proposition 1, we propose the private H-FSGD (PH-FSGD) as following, initialized at $\bar{f}_0 = \hat{f}_0 = 0$,

$$\begin{aligned}
\hat{f}_n &= \hat{f}_{n-1} + \gamma_n w_\tau \left( Y_n - \langle \hat{f}_{n-1}, K_{X_n} \rangle_{\mathcal{H}} \right) \left( Y_n - \langle \hat{f}_{n-1}, K_{X_n} \rangle_{\mathcal{H}} \right) K_{X_n} + \gamma_n \xi_n, \\
\bar{f}_n &= \frac{n-1}{n} \bar{f}_{n-1} + \frac{1}{n} \hat{f}_n.
\end{aligned} \tag{7}$$

It is worth noting that PH-FSGD supports varying privacy constraints across iterations, with per-iteration noise ensuring that each data point is protected at the source.

We summarize our approach as in Algorithm 1. We make the following remarks.

**Remark 1.** Algorithm 1 supports mixed privacy regimes through iteration-specific privacy budgets. Large $\varepsilon_n$ values effectively imply non-private updates. While our current experiments focus on fully private or non-private settings, the algorithm naturally accommodates hybrid cases by adjusting per-iteration noise.

**Remark 2.** While our method is developed in the infinite-dimensional RKHS, the computational implementation employs grid discretization as a finite approximation, which is a standard approach in functional data analysis to balance computational feasibility with theoretical fidelity. Following established practice (e.g., Dieuleveut and Bach [2016], Liu et al. [2023]), we use dense grids to ensure accurate function recovery, with approximation error diminishing as grid density increases.

**Remark 3.** When outliers are not a concern ($\tau \to \infty$), the Huber loss reduces to squared loss. In the context of DP, the privacy guarantee is governed by the sensitivity of the gradient. Without additional

---

**Algorithm 1** PH-FSGD

---

1: **Input:** The streaming data $\{(X_n, Y_n)\}_{n\in\mathbb{N}}$, the initial estimates $\bar{f}(\cdot) = \hat{f}(\cdot) = 0$, the step size sequences $\{\gamma_n\}_{n\in\mathbb{N}}$, the tuning parameter $\tau > 0$, the reproducing kernel $K$, the bounded parameter $B > 0$, the privacy parameters $\{\varepsilon_n\}_{n\in\mathbb{N}}$, $\{\delta_n\}_{n\in\mathbb{N}}$, and the function grids $\{t_j\}_{j=1}^{J}$.

2: **for** $n = 1, 2, \ldots$ **do**

3:   Generate the noise $\{\xi_n(t_j)\}_{j=1}^{J}$ from $N_J(\mathbf{0}, \frac{8\tau^2 B^2 \log(2/\delta_n)}{\varepsilon_n^2} K^{(t)})$, where $K^{(t)}$ is a $J \times J$ matrix with its components $(K^{(t)})_{ij} = K(t_i, t_j)$.

4:   Calculate the residual: $\text{res}_n = Y_n - \langle \hat{f}_{n-1}, K_{X_n} \rangle_{\mathcal{H}}$.

5:   Perform the noisy gradient descent at each function grid $t_j$ for $j = 1, \ldots, J$ as follows.

6:     **if** $|\text{res}_n| \leq \tau$

7:       **then** $\hat{f}_n(t_j) = \hat{f}_{n-1}(t_j) + \gamma_n \text{res}_n K(X_n, t_j) + \gamma_n \xi_n(t_j)$.

8:     **elseif** $\text{res}_n > \tau$

9:       **then** $\hat{f}_n(t_j) = \hat{f}_{n-1}(t_j) + \gamma_n \tau K(X_n, t_j) + \gamma_n \xi_n(t_j)$.

10:     **else** $\hat{f}_n(t_j) = \hat{f}_{n-1}(t_j) - \gamma_n \tau K(X_n, t_j) + \gamma_n \xi_n(t_j)$.

11:   Update $\bar{f}_n$ at each function grid:

$$\bar{f}_n(t_j) = \frac{n-1}{n} \bar{f}_{n-1}(t_j) + \frac{1}{n} \hat{f}_n(t_j), j = 1, \ldots, J.$$

12: **end for**

13: **Output:** The estimators $\{\bar{f}_n(t_j)\}_{j=1}^{J}$ at each function grid $t_j$ and each iteration $n$.

---

assumptions, it is standard practice to apply gradient clipping to ensure bounded sensitivity under the squared loss [Song et al., 2021]. If the response variable $Y$ is further assumed to be bounded, then the gradient sensitivity is naturally finite, and the privacy guarantee directly depends on its magnitude. Thus, squared loss remains privacy-compatible with proper sensitivity control.

**Theorem 1.** *The estimators $\hat{f}_n$ and $\bar{f}_n$ at each iteration $n \in \mathbb{N}$ in Algorithm 1 satisfy $(\max_{1\leq i\leq n}\{\varepsilon_i\}, \max_{1\leq i\leq n}\{\delta_i\})$-LDP for $n \in \mathbb{N}$.*

Theorem 1 ensures that each update in the proposed PH-FSGD algorithm satisfies $(\max_{1\leq i\leq n}\{\varepsilon_i\}, \max_{1\leq i\leq n}\{\delta_i\})$-LDP by adding Gaussian process noise calibrated to the gradient's sensitivity. This protects individual sample privacy at each iteration without storing raw data. Cumulative privacy over time can be analyzed via the composition result in Proposition 2.

## 4 Theoretical properties

In this section, we establish non-asymptotic convergence rates for the averaged estimator $\bar{f}_n$. We first introduce the necessary assumptions.

**Assumption 3.** *Suppose that the reproducing kernel $K$ satisfies $K \preccurlyeq \Sigma$, where $\Sigma = \mathbb{E}(K_X \otimes K_X)$, and the symbol $\preccurlyeq$ denotes the order between self-adjoint operators.*

**Assumption 4.** *There exists a constant $M > 0$ such that $\|f\|_{\mathcal{H}} \leq M$ for all $f \in \mathcal{H}$.*

**Assumption 5.** *Suppose that $Cov(e, X) = 0$. In addition, denote $p_e$ is the probability density function of $e$. There exist constants $m > 0$ and $\kappa > 0$ such that $\inf_{t\in(-m,m)} p_e(t) \geq \kappa$.*

**Assumption 6.** *Denote $\{\nu_i\}_{i\in\mathbb{N}}$ as the sequence of non-zero eigenvalues of the operator $\Sigma$ in the decreasing order. Suppose that $\nu_i \leq s^2 i^{-\alpha}$ for some $\alpha > 1$ and some positive constant $s$.*

**Assumption 7.** *Assume that $f_{\mathcal{H}} \in \Sigma^r(L_{P_X}^2)$ with $r \geq 0$, where $\Sigma^r(L_{P_X}^2) = \left\{\sum_{i=1}^{\infty} b_i \phi_i \text{ such that } \sum_{i=1}^{\infty} \frac{b_i^2}{\nu_i^{2r}} < \infty\right\}$ with the eigenvalues $\{b_i\}_i$ and the eigenvectors $\{\phi_i\}_i$. As a consequence, $\|\Sigma^{-r}(f_{\mathcal{H}})\|_{L_{P_X}^2} < \infty$.*

Assumptions 3 and 4 are automatically satisfied by any continuous bounded Mercer kernel on a compact domain. Common examples include the Gaussian, Laplace, Periodic, and Polynomial kernels. Assumption 5 is mild, as it covers any continuous distribution with positive density at zero,

such as the normal distribution, $t$-distribution, or zero-mean uniform distribution. Assumptions 6 and 7 are standard in the RKHS literature [Dieuleveut and Bach, 2016, Fischer and Steinwart, 2020, Zhang et al., 2023]. Assumption 6 imposes smoothness via the spectral decay of the covariance operator $\Sigma$; a larger decay exponent $\alpha$ leads to faster eigenvalue decay, effectively reducing the RKHS's effective dimension and imposing stronger smoothness constraints. Assumption 7 relates to the regularity of $f_{\mathcal{H}}$. A larger source condition exponent $r$ indicates greater smoothness. In particular, when $r = 1/2$, the source space $\Sigma^{1/2}(L_{P_X}^2)$ coincides with the $\mathcal{H}$. For $r \geq 1/2$, the minimizer $f_{\mathcal{H}}$ lies in $\mathcal{H}$, whereas for $r < 1/2$, it may reside only in the closure of $\mathcal{H}$.

We establish the consistency of our proposed H-FSGD and PH-FSGD estimators. By comparing (5) and (7), we observe that the non-private estimator is a special case of the private one with $\xi_n = 0$ for all $n$. Accordingly, we focus on the consistency of the private estimators here and defer the analysis of their non-private counterparts to Appendix D. The privacy budget is allowed to vary across iterations. A notable special case arises when the estimator reduces to $(\varepsilon, \delta)$-LDP with a uniform privacy budget across all individuals, i.e., $\varepsilon = \varepsilon_1 = \cdots = \varepsilon_n = \cdots$ and similarly for $\delta$. Since handling heterogeneous privacy budgets requires no additional theoretical complexity beyond the uniform case, we assume a common budget $(\varepsilon, \delta)$ for simplicity.

The following lemma plays a central role in establishing the consistency of the proposed estimators.

**Lemma 1.** *Suppose that Assumptions 2, 4 and 5 are fulfilled. Denote $\mathcal{T}_f = \mathbb{E}\left[w_\tau(Y_n - f(X_n))K_{X_n} \otimes K_{X_n}\right]$ for any $f \in \mathcal{H}$. There exists a constant $c > 0$ such that $\mathcal{T}_f \succcurlyeq c\Sigma$ for any $n$ and any $f$, where $\succcurlyeq$ denotes the order between self-adjoint operators.*

Theorem 2 establishes the convergence rate of the privatized estimator $\bar{f}_n$ with constant step size.

**Theorem 2** (Constant step size). *Suppose Assumptions 1–7 hold. Take any constant choice $\gamma_i = \gamma = \Gamma(n)$, for $1 \leq i \leq n$. If $c^{-1}\gamma B^2 < 1$ with $c$ defined in Lemma 1, then*

$$\mathbb{E}\|\bar{f}_n - f_{\mathcal{H}}\|_{L_{P_X}^2}^2$$
$$\leq O\left(\left(\sigma^2 + \frac{8\tau^2 B^2 \log(2/\delta)}{\varepsilon^2}\right)\left(\gamma^{1/\alpha}n^{-1+1/\alpha} + n^{-1}\right) + (1 + q(\gamma, n))\gamma^{-2r}n^{-2\min\{r,1\}}\left\|\Sigma^{-r}f_{\mathcal{H}}\right\|_{L_{P_X}^2}^2\right),$$
$$(8)$$

*where $q(\gamma, n) = 0$ for $r \leq 1/2$, and $q(\gamma, n) = \gamma^{(1+\alpha)(2r-1)/\alpha}n^{(2r-1)/\alpha}$ for $r > 1/2$.*

On the right-hand side of (8), the first term represents the variance component, while the second corresponds to the bias. The variance term is associated with both the intrinsic noise level $\sigma^2$ and an additional factor $8\tau^2 B^2 \log(2/\delta)/\varepsilon^2$, which arises from the privacy constraint. A larger step size amplifies the variance, whereas the bias term decreases as the step size increases. This reveals a fundamental bias-variance trade-off in the choice of step size to control estimation error. Compared to regularized kernel ridge regression (KRR) with penalty parameter $\lambda$ in offline [Yang et al., 2017], the variance of KRR scales as $\lambda^{-1/\alpha}n^{-1}$, and the bias as $\lambda^{2r}$. By choosing $\lambda = (\gamma n)^{-1}$, the convergence rate of KRR matches that of our PH-FSGD estimators. Our current theoretical bounds assume sufficiently fine discretization to guarantee the grid-based solution closely approximates the RKHS optimum. While small $J$ may introduce non-negligible error, our empirical results confirm that larger $J$ effectively mitigates this discrepancy. Extending the theory to explicitly account for finite grid effects remains an important direction for future work.

To optimize the bias-variance trade-off, we choose the step size $\gamma$ as prescribed in Corollary 1.

**Corollary 1** (Constant step size). *Under Assumptions 1–7, take the constant step size $\gamma_i = \gamma = \Gamma(n) \asymp n^{-\zeta}$ for $i = 1, 2, \ldots$.*

*(i) When $0 < r \leq (\alpha - 1)/(2\alpha)$, take $\zeta = 0$, then $\mathbb{E}\|\bar{f}_n - f_{\mathcal{H}}\|_{L_{P_X}^2}^2 \leq O\left(n^{-2r}\right)$;*

*(ii) When $(\alpha - 1)/(2\alpha) < r \leq 1$, take $\zeta = (2r\alpha + 1 - \alpha)/(2r\alpha + 1)$, then $\mathbb{E}\|\bar{f}_n - f_{\mathcal{H}}\|_{L_{P_X}^2}^2 \leq O\left(n^{-2r\alpha/(2r\alpha+1)}\right)$;*

*(iii) When $1 < r \leq (\alpha + 2)/2$, take $\zeta = (\alpha + 1)/(2r\alpha + 1)$, then $\mathbb{E}\|\bar{f}_n - f_{\mathcal{H}}\|_{L_{P_X}^2}^2 \leq O\left(n^{-(2r\alpha-2r+2)/(2r\alpha+1)}\right)$;*

*(iv) When $r > (\alpha + 2)/2$, take $\zeta = 1/(1 + \alpha)$ then $\mathbb{E}\|\bar{f}_n - f_{\mathcal{H}}\|_{L_{P_X}^2}^2 \leq O\left(n^{-\alpha/(1+\alpha)}\right)$.*

The results above focus on the effect of the sample size $n$, all privacy parameters $(\varepsilon, \delta)$ are treated as constants and are thus absorbed into the $O(\cdot)$ notation. For results with explicit dependence on the privacy parameters in the convergence rates, we refer to Corollary 2. When $0 < r \leq (\alpha - 1)/(2\alpha)$, where the original space of $f_{\mathcal{H}}$ is less smooth than the estimation RKHS, our estimator remains consistent, with the estimation error vanishing as $n \to \infty$, depending on $r$. For $(\alpha - 1)/(2\alpha) < r \leq 1$, the estimator achieves the minimax optimal rate [Zhang et al., 2023]. This region includes the classical case $\Sigma^r(L_{P_X}^2) = \mathcal{H}$ with $r = 1/2$, smoother scenarios with $r > 1/2$, and even less smooth cases where $(\alpha - 1)/(2\alpha) < r < 1/2$. The convergence rate improves as either $\alpha$ or $r$ increases, reflecting the intuition that greater smoothness in the estimation or original space leads to faster rates. When $r > 1$, although Assumption 7 is stronger than $r = 1$, we do not improve the bound, which is the saturation phenomenon [Engl et al., 1996].

In Appendix D, we also examine consistency under non-constant step sizes. Theoretical results for non-private estimators are also included, showing similar patterns to the private case when the privacy term is set to zero. All technical proofs are provided in Appendix J.

## 5 Experiments

This section evaluates the finite-sample performance of the proposed H-FSGD and PH-FSGD estimators under two cases: (Case 1) the true function $f^*(x)$ is a sine function; (Case 2) $f^*(x)$ is a linear combination of two Beta density functions. Two types of errors are considered, the Gaussian distribution $N(0, 0.25)$ and $t$ distribution with degree of freedom 3. Simulation details are provided in Appendix F. We assess both non-private and privacy-preserving settings, repeating each setup 200 times independently and using the mean squared error (MSE) as the evaluation metric.

**Example 5.1 (Non-private synthetic data).** In this example, we evaluate the H-FSGD estimator and compare it with two baselines: one-pass FSGD using least-squares loss (denoted L2-FSGD) and least-squares FSGD with access to historical data (denoted offline) from Dieuleveut and Bach [2016]. All three methods are tested under both constant and non-constant step-size schemes. We report the MSEs of the averaged estimators at sample sizes $n = 2000, 5000$, and $10000$, and include function fitting plots at $n = 10000$ to illustrate model performance. Box plots and corresponding function fitting results for Cases 1 and 2 under constant step sizes are shown in Figure 2, while the results under non-constant step sizes are presented in Figure 5 in Appendix F for space considerations.

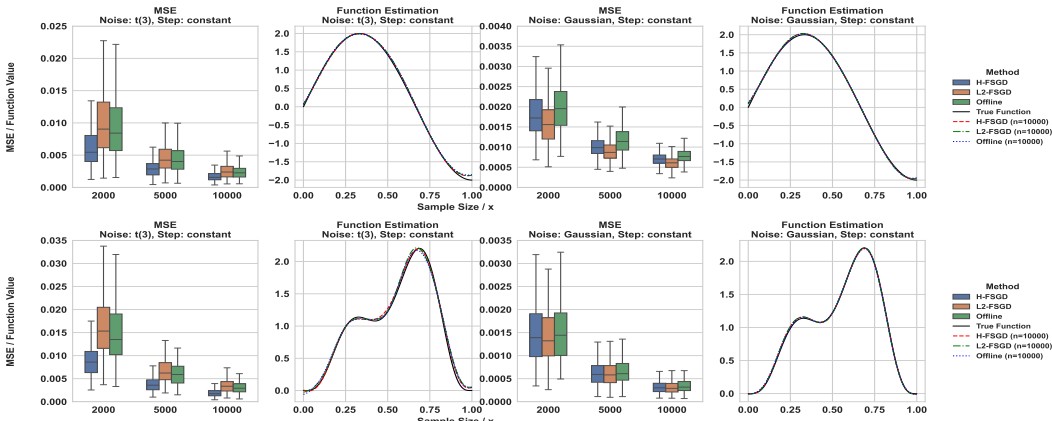

Figure 2: Box-plots and function fitting plots for Case 1 (top panels) and Case 2 (bottom panels) with the constant step size scheme in Example 5.1.

Figures 2 and 5 demonstrate that when the error follows a heavy-tailed distribution such as the $t(3)$ distribution, our proposed H-FSGD method significantly outperforms the least-squares-based FSGD in both the median and interquartile range of the MSEs, especially when the sample size is small. Under Gaussian errors, H-FSGD matches or exceeds the performance of the baseline methods. Moreover, as the sample size grows, the MSEs of all three methods declines, and by $n = 10000$ they all achieve near-perfect function fits.

**Example 5.2** (**Private synthetic data**).  In this example, we evaluate our PH-FSGD method under two LDP settings: $(3, 0.1)$-LDP and $(2, 0.2)$-LDP. Box-plots of MSEs and corresponding function fitting results under both data-generating cases and constant step size schemes are shown in Figures 3 and 6 (in Appendix F), respectively.

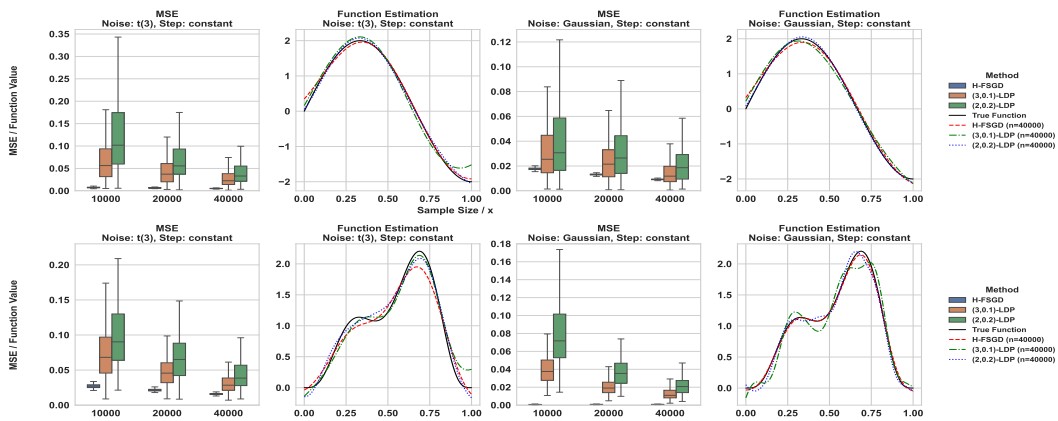

Figure 3: Box-plots and function fitting plots for Case 1 (top panels) and Case 2 (bottom panels) with the constant step size scheme in Example 5.2.

As shown in Figures 3 and 6, the non-private estimator consistently achieves the lowest median MSE and tightest variability, while the two LDP variants incur progressively larger error as privacy strength increases ($(3, 0.1)$-LDP represents moderate privacy, and $(2, 0.2)$-LDP represents strong privacy). These results reflect the inherent trade-off between privacy and statistical efficiency: stronger privacy (i.e., more noise) enhances protection but also leads to greater estimation error and slower convergence. Despite strong privacy constraints, all methods still recover the true function shape well. The differences in box plots and curve fits diminish as the sample size increases, demonstrating that larger datasets can effectively offset the accuracy loss from LDP.

**Computational time.**    We compare the computational efficiency of different methods on a laptop equipped with a 3.20 GHz AMD Ryzen 7 5800H CPU and 16GB RAM. Computational times are recorded for sample sizes ranging from $n = 4000$ to $n = 40000$, as shown in Figure 4.

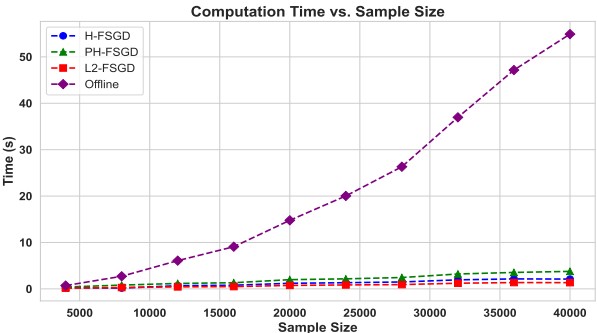

Figure 4:  Change of computation times of our proposed H-FSGD and PH-FSGD, and baselines L2-FSGD and Offline as the sample sizes increasing from $4000$ to $40000$.

Figure 4 confirms that both our proposed H-FSGD and PH-FSGD methods are one-pass algorithms with linear computational complexity $O(n)$, whereas the offline method is computationally intensive, exhibiting cubic complexity $O(n^3)$ as the sample size $n$ increases. Further experimental results, encompassing analyses of step-size sensitivity, performance beyond theoretical assumptions, robustness under contamination models, and a real-data application, are detailed in Appendix F and Appendix G.

## Acknowledgements

We sincerely thank the reviewers, ACs, SACs, and PCs for their time, constructive feedback, and thoughtful discussions. Niansheng Tang's research was supported by the National Key R&D Program of China (No. 102022YFA1003701). Chenfei Gu's research was supported by the Fundamental Research Funds for the Central Universities (No. CXJJ-2024-448). Qiangqiang Zhang's research was supported by the National Key R&D Program of China (No. 2023YFA1008701) and the National Natural Science Foundation of China (Nos. 12371148, 12326603, and 12431017). Ting Li's research was partially supported by the National Natural Science Foundation of China (No. 12571304), the Shanghai Pujiang Programme (No. 24PJC030), CCF-DiDi GAIA Collaborative Research Funds and the Program for Innovative Research Team of Shanghai University of Finance and Economics. Jinhan Xie's research was supported by the National Natural Science Foundation of China (No. 12501388).

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

# Appendix

## A   Notation

Table 2: List of notation

| Notation | Meaning |
|---|---|
| $X_n$ | the $n$-th copy of the random covariate $X$ |
| $Y_n$ | the $n$-th copy of the response $Y$ |
| $e_n$ | the $n$-th copy of the error $e$ |
| $\sigma^2$ | the variance of the error $e$ |
| $f^\star$ | the target nonparametric function lying in the space $L^2(P_X)$ |
| $K$ | the reproducing kernel |
| $\mathcal{H}$ | a RKHS |
| $\langle \cdot, \cdot \rangle_{\mathcal{H}}$ | the inner product in $\mathcal{H}$ |
| $\bar{\mathcal{H}}$ | the closure of $\mathcal{H}$ |
| $f_{\mathcal{H}}$ | the best approximation of $f^\star$ in the RKHS $\bar{\mathcal{H}}$ |
| $L_\tau$ | the Huber loss with the parameter $\tau$ |
| $\widehat{\nabla L_\tau}$ | the estimator of the Frechet gradient of the Huber loss |
| $B^2$ | the uniform upper bound of $K(x,x)$ for all $x$ |
| $\gamma_n$ | the step size at the $n$-th iteration |
| $\hat{f}_n$ | the current private estimator of $f_{\mathcal{H}}$ at the $n$-th iteration |
| $\bar{f}_n$ | the averaged private estimator of $f_{\mathcal{H}}$ at the $n$-th iteration |
| $\bar{f}_n^0$ | the averaged non-private estimator of $f_{\mathcal{H}}$ at the $n$-th iteration |
| $\varepsilon_n, \delta_n$ | the privacy budget at the $n$-th iteration |
| $\{t_j\}_{j=1}^J$ | the function grids |
| $\Sigma$ | the covariance operator associated with the kernel $K$, i.e., $\Sigma = \mathbb{E}(K_X \otimes K_X)$ |
| $\alpha$ | the parameter characterizing the decay rate of the eigenvalues of the covariance operator $\Sigma$ |
| $r$ | the parameter quantifying the regularity of the target function $f_{\mathcal{H}}$ with respect to the eigenbasis of $\Sigma$ |

## B   Background on RKHS and LDP

### B.1   RKHS and linear operators

**Definition 1.** *[Gu and Gu, 2013] A Hilbert space $\mathcal{H}$ of functions $f : \mathcal{X} \to \mathbb{R}$ is said to be an RKHS if the elements of $\mathcal{H}$ are functions defined on a compact topological space $\mathcal{X}$, and there is a bivariate function $K(\cdot, \cdot) : \mathcal{X} \times \mathcal{X} \to \mathbb{R}$ having the following properties:*

*(1) For all $x \in \mathcal{X}$, the function $K_x = K(x, \cdot)$ is in $\mathcal{H}$.*

*(2) The reproducing property holds, i.e., for all $x \in \mathcal{X}$ and $f \in \mathcal{H}$, $f(x) = \langle f, K_x \rangle_{\mathcal{H}}$,*

*where $\langle \cdot, \cdot \rangle_{\mathcal{H}}$ is the associated inner product of $\mathcal{H}$. In this case, $K$ is the reproducing kernel of $\mathcal{H}$.*

Define an operator $\Sigma : L^2(P_X) \to L^2(P_X)$, such that for any $f \in L^2(P_X)$,

$$\Sigma f = \mathbb{E}[K_X f(X)].$$

Then for any $z \in \mathcal{X}$, $\Sigma f(z) = \mathbb{E}[K(X, z)g(X)]$. If $f \in \mathcal{H}$, the reproducing property gives that $\Sigma = \mathbb{E}[K_X \otimes K_X]$, where $K_X \otimes K_X$ is an operator defined as $(K_X \otimes K_X)f = \langle f, K_X \rangle_{\mathcal{H}} K_X = f(X)K_X$ for $f \in \mathcal{H}$. It follows that for any $f, g \in \mathcal{H}$,

$$\langle f, \Sigma g \rangle_{\mathcal{H}} = \mathbb{E}[f(X)g(X)].$$

This operator links the RKHS inner product to the expected product of functions, facilitating theoretical analysis. Assumptions 1–2 make $\Sigma$ be an valid linear operator for the whole space $L^2(P_X)$.

The closure of $\mathcal{H}$ in $L^2(P_X)$, denoted by $\bar{\mathcal{H}}$, is defined as the set of all limits of sequences in $\mathcal{H}$. Formally,

$$\bar{\mathcal{H}} = \{f \in L^2(P_X) : \exists \{f_k\} \subset \mathcal{H} \text{ such that } \|f_k - f\|_{L^2(P_X)} \to 0\}.$$

Intuitively, $\bar{\mathcal{H}}$ consists of all functions in $L^2(P_X)$ that can be approximated arbitrarily well (in $L^2(P_X)$ norm) by a sequence of functions in $\mathcal{H}$. This construction is standard in the kernel methods literature [Dieuleveut and Bach, 2016], and it allows us to avoid assuming that $f^\star$ belongs to $\mathcal{H}$ or that $\mathcal{H}$ is dense in $L^2(P_X)$.

## B.2  Differential privacy

A random algorithm, denoted as $M$, can be intuitively considered as protecting privacy if it prevents an attacker from distinguishing whether a specific datum $x$ belongs to the dataset $X$ when the algorithm is applied to $X$. To formalize this notion, the concept of CDP is introduced.

**Definition 2** (CDP, [Dwork et al., 2006a,b]). *Let $\varepsilon > 0$ and $\delta \geq 0$. A dataset $X = (x_1, x_2, \ldots, x_n) \in \mathcal{X}^n$ consists of $n$ data from some space $\mathcal{X}$. Two datasets $X$ and $X'$ are called neighbors if they differ by only one entry, denoted as $X \sim X'$. A random algorithm $M : \mathcal{X}^n \to \mathcal{Y}$ is said to be $(\varepsilon, \delta)$-DP if for any neighboring datasets $X$ and $X'$, and any measurable set $E \subset \mathcal{Y}$,*

$$\mathbb{P}(M(X) \in E) \leq e^\varepsilon \mathbb{P}(M(X') \in E) + \delta,$$

*where the probabilities are computed over the randomness of the mechanism $M$.*

This definition ensures that the probabilities of obtaining certain outcomes under the algorithm $M$ applied to datasets $X$ and $X'$ are similar. However, CDP relies on a trusted curator, which may lead to internal exposure risks and undermine the goal of privacy protection. To address this issue, LDP eliminates such risks by randomizing data prior to collection.

**Definition 3** (LDP, [Xiong et al., 2020]). *Let $\varepsilon > 0$ and $\delta \geq 0$. A randomized algorithm $M : \mathcal{X} \to \mathcal{Y}$ is said to be $(\varepsilon, \delta)$-LDP if for any pair of input individual values $x, x' \in \mathcal{X}$, and any measurable set $E \subset \mathcal{Y}$,*

$$\mathbb{P}(M(x) \in E) \leq e^\varepsilon \mathbb{P}(M(x') \in E) + \delta,$$

*where the probabilities are computed over the randomness of the mechanism $M$.*

LDP can be regarded as a stricter variant of DP, where individuals add noise to their data before sharing, ensuring that each user's information remains confidential without relying on a trusted data collector. We next introduce the following Gaussian mechanism to construct a function estimator with DP.

**Proposition 1.** *[Hall et al., 2013] Let $D \in \mathcal{D}$ be an input dataset. Suppose that the family of functions $\{f_D : D \in \mathcal{D}\}$ lies in the reproducing kernel Hilbert space (RKHS) $\mathcal{H}$ with the reproducing kernel $K$, the inner product $\langle \cdot, \cdot \rangle_{\mathcal{H}}$ and the corresponding norm $\| \cdot \|_{\mathcal{H}}$. Two datasets $D$ and $D'$ are called neighbors if they differ by only one entry, denoted as $D \sim D'$. Let $\sup_{D \sim D'} \|f_D - f_{D'}\|_{\mathcal{H}} \leq \Delta_0$, $c(\delta) \geq (2\log(2/\delta))^{1/2}$ for $\delta > 0$. Take $G$ as the sample path of a Gaussian process having mean zero and covariance function $K$. Then the release of*

$$\widetilde{f}_D = f_D + \frac{\Delta_0 c(\delta)}{\varepsilon} G$$

*is $(\varepsilon, \delta)$-differential private.*

Some useful properties for the construction of LDP algorithms are stated below.

**Proposition 2.** *[Xiong et al., 2020]*
*(1) Parallel composition for LDP: Let $M_i$, $i = 1, \ldots, k$ be $(\varepsilon_i, \delta_i)$-LDP mechanisms and $X_1, \ldots, X_k$ be disjoint. Then $M(X_1, \ldots, X_k) = (M_1(X_1), \ldots, M_k(X_k))$ is $(\max_i \varepsilon_i, \max_i \delta_i)$-LDP.*
*(2) Postprocessing property for LDP: Let $M_1$ be an $(\varepsilon, \delta)$-LDP mechanism, and $M_2$ be a mechanism without any privacy constraints. Then the composition of $M_1$ and $M_2$, i.e., $M_2(M_1(\cdot))$ is $(\varepsilon, \delta)$-LDP.*

# C   Algorithms for H-FSGD

---

**Algorithm 2** Choice of $\tau$

---

1: **Input:** Small sample data $\{(X_n, Y_n)\}_{n=1}^{N_\tau}$, the initial estimates $\bar{f}(\cdot) = \hat{f}(\cdot) = 0$, the step size sequences $\{\gamma_n\}_{n \in \mathbb{N}}$, the reproducing kernel $K$, and the function grids $\{t_j\}_{j=1}^J$.
2: Use existing least-squares functional SGD [Liu et al., 2023] to obtain an estimator $\hat{f}^{\text{LS}}$.
3: Calculate the prediction errors $\{\text{res}_i\}_i$ based on the estimator $\hat{f}^{\text{LS}}$.
4: Estimate the standard deviation of noise via $\hat{\sigma} = \text{Median}\{|\text{res}_i|\}/0.6745$.
5: Choose $\tau$ as $\tau = 1.345\hat{\sigma}$.
6: **Output:** $\tau$.

---

**Algorithm 3** H-FSGD

---

1: **Input:** The streaming data $\{(X_n, Y_n)\}_{n \in \mathbb{N}}$, the initial estimates $\bar{f}(\cdot) = \hat{f}(\cdot) = 0$, the step size sequences $\{\gamma_n\}_{n \in \mathbb{N}}$, the tuning parameter $\tau > 0$ via Algorithm 2, the reproducing kernel $K$, and the function grids $\{t_j\}_{j=1}^J$.
2: **for** $n = 1, 2, \ldots$ **do**
3:     Calculate the residual: $\text{res}_n = Y_n - \langle \hat{f}_{n-1}, K_{X_n} \rangle_{\mathcal{H}}$.
4:     Perform the gradient descent at each function grid $t_j$ for $j = 1, \ldots, J$ as follows.
5:         **if** $|\text{res}_n| \leq \tau$
6:             **then** $\hat{f}_n(t_j) = \hat{f}_{n-1}(t_j) + \gamma_n \text{res}_n K(X_n, t_j)$.
7:         **elseif** $\text{res}_n > \tau$
8:             **then** $\hat{f}_n(t_j) = \hat{f}_{n-1}(t_j) + \gamma_n \tau K(X_n, t_j)$.
9:         **else** $\hat{f}_n(t_j) = \hat{f}_{n-1}(t_j) - \gamma_n \tau K(X_n, t_j)$.
10:     Update $\bar{f}_n$ at each function grid:

$$\bar{f}_n(t_j) = \frac{n-1}{n} \bar{f}_{n-1}(t_j) + \frac{1}{n} \hat{f}_n(t_j), j = 1, \ldots, J.$$

11: **end for**
12: **Output:** The estimators $\left\{ \bar{f}_n(t_j) \right\}_{j=1}^J$ at each function grid $t_j$ and each iteration $n$.

---

# D   Additional theoretical results

We state the following refinement of Corollary 1 to make the dependence on the privacy parameters $(\varepsilon, \delta)$ explicit in the convergence rates.

**Corollary 2** (Constant step size, explicit dependence on privacy parameters). *Under Assumptions 1–7, take the constant step size $\gamma_i = \gamma = \Gamma(n) \asymp n^{-\zeta}$ for $i = 1, 2, \ldots$.*

*(i) When $0 < r \leq (\alpha - 1)/(2\alpha)$, take $\zeta = 0$, then $\mathbb{E}\|\bar{f}_n - f_{\mathcal{H}}\|_{L^2_{P_X}}^2 \leq O\left(n^{-2r}\right)$;*

*(ii) When $(\alpha - 1)/(2\alpha) < r \leq 1$, take $\zeta = (2r\alpha + 1 - \alpha)/(2r\alpha + 1)$, then $\mathbb{E}\|\bar{f}_n - f_{\mathcal{H}}\|_{L^2_{P_X}}^2 \leq O\left(\left(\sigma^2 + \frac{8\tau^2 B^2 \log(2/\delta)}{\varepsilon^2}\right) n^{-2r\alpha/(2r\alpha+1)}\right)$;*

*(iii) When $1 < r \leq (\alpha + 2)/2$, take $\zeta = (\alpha + 1)/(2r\alpha + 1)$, then $\mathbb{E}\|\bar{f}_n - f_{\mathcal{H}}\|_{L^2_{P_X}}^2 \leq O\left(\left(\sigma^2 + \frac{8\tau^2 B^2 \log(2/\delta)}{\varepsilon^2}\right) n^{-(2r\alpha - 2r + 2)/(2r\alpha+1)}\right)$;*

*(iv) When* $r > (\alpha + 2)/2$, *take* $\zeta = 1/(1 + \alpha)$ *then* $\mathbb{E}\|\bar{f}_n - f_{\mathcal{H}}\|^2_{L^2_{P_X}} \leq$ $O\left(\left(\sigma^2 + \frac{8\tau^2 B^2 \log(2/\delta)}{\varepsilon^2}\right) n^{-\alpha/(1+\alpha)}\right).$

We consider the consistency of privatized estimators in the setting of non-constant step sizes.

**Theorem 3** (Non-constant step size)**.** *Suppose Assumptions 1–7 hold. Take the step sizes* $\gamma_i \asymp i^{-\zeta}$ *with* $\zeta \in (0, 1)$ *satisfying* $2r - 1/(1 - \zeta) < 0$, *for* $i = 1, 2, \ldots$. *If* $c^{-1}\gamma_0 B^2 < 1$ *with* $c$ *defined in Lemma 1, then:*
*(i)* $0 \leq \zeta \leq 1/2$,

$$\mathbb{E}\|\bar{f}_n - f_{\mathcal{H}}\|^2_{L^2_{P_X}} \leq O\left(\left(\sigma^2 + \frac{8\tau^2 B^2 \log(2/\delta)}{\varepsilon^2}\right) \gamma_n^{1/\alpha} n^{-1+1/\alpha} + \gamma_n^{-1}(n\gamma_n)^{-2r} \left\|\Sigma^{-r} f_{\mathcal{H}}\right\|^2_{L^2_{P_X}}\right);$$

*(ii)* $1/2 < \zeta < 1$,

$$\mathbb{E}\|\bar{f}_n - f_{\mathcal{H}}\|^2_{L^2_{P_X}} \leq O\left(\left(\sigma^2 + \frac{8\tau^2 B^2 \log(2/\delta)}{\varepsilon^2}\right) (n\gamma_n)^{-2+1/\alpha} + \gamma_n^{-1}(n\gamma_n)^{-2r} \left\|\Sigma^{-r} f_{\mathcal{H}}\right\|^2_{L^2_{P_X}}\right).$$

Consistent with Theorem 2, the first term in (i) and (ii) reflects the variance order, while the second term captures the bias order. Theorem 3 indicates that for $\zeta \in [0, 1/2]$, both the bias and variance terms respond to changes of the step size similarly to those in Theorem 2. In contrast, for a smaller step size with $\zeta \in (1/2, 1)$, the bias term retains this behavior, but the variance term increases as the step size decreases.

We can also choose an optimal step size and achieve the optimal convergence rate.

**Corollary 3** (Non-constant step size)**.** *Under Assumptions 1–7, take the step sizes* $\gamma_i \asymp i^{-\zeta}$ *for* $i = 1, 2, \ldots$.
*(i) When* $0 < r \leq (\alpha - 1)/(2\alpha)$, *take* $\zeta = 0$, *then* $\mathbb{E}\|\bar{f}_n - f_{\mathcal{H}}\|^2_{L^2_{P_X}} \leq O\left(n^{-2r}\right);$
*(ii) When* $(\alpha - 1)/(2\alpha) < r < (1 + \alpha)/(2\alpha)$, *take* $\zeta = (2r\alpha + 1 - \alpha)/(2r\alpha + 1 + \alpha)$, *then* $\mathbb{E}\|\bar{f}_n - f_{\mathcal{H}}\|^2_{L^2_{P_X}} \leq O\left(n^{-(2r\alpha+\alpha-1)/(2r\alpha+1+\alpha)}\right);$
*(iii) When* $r \geq (1 + \alpha)/(2\alpha)$, *take* $\zeta = 1/(1 + \alpha)$, *then* $\mathbb{E}\|\bar{f}_n - f_{\mathcal{H}}\|^2_{L^2_{P_X}} \leq O\left(n^{-\alpha/(1+\alpha)}\right).$

A comparison between Corollary 3 and Corollary 1 reveals that when $0 < r \leq (\alpha - 1)/(2\alpha)$ or $r \geq (\alpha + 2)/2$, the convergence rates of the non-constant and constant step size schemes are identical. In contrast, for $(\alpha - 1)/(2\alpha) < r < (\alpha + 2)/2$, the convergence rate achieved under the non-constant step size scheme is slower than that under the constant step size setting.

The consistency of the privatized estimators can be easily reduced to the estimators without DP, as shown in Corollary 4 and 5.

**Corollary 4** (Constant step size, without privacy)**.** *Suppose Assumptions 1–7 hold. Consider the estimator* $\bar{f}_n^0$ *without privacy protection defined via the recursion (5). Take any constant choice* $\gamma_i = \gamma = \Gamma(n)$, *for* $1 \leq i \leq n$. *If* $c^{-1}\gamma B^2 < 1$ *with* $c$ *defined in Lemma 1, then*

$$\mathbb{E}\|\bar{f}_n^0 - f_{\mathcal{H}}\|^2_{L^2_{P_X}} \leq O\left(\sigma^2 \left(\gamma^{1/\alpha} n^{-1+1/\alpha} + n^{-1}\right) + (1 + q(\gamma, n))\gamma^{-2r} n^{-2\min\{r,1\}} \left\|\Sigma^{-r} f_{\mathcal{H}}\right\|^2_{L^2_{P_X}}\right).$$

*Further, take* $\Gamma(n) \asymp n^{-\zeta}$.

*(i) When* $0 < r \leq (\alpha - 1)/(2\alpha)$, *take* $\zeta = 0$, *then* $\mathbb{E}\|\bar{f}_n^0 - f_{\mathcal{H}}\|^2_{L^2_{P_X}} \leq O\left(n^{-2r}\right);$

*(ii) When* $(\alpha - 1)/(2\alpha) < r \leq 1$, *take* $\zeta = (2r\alpha + 1 - \alpha)/(2r\alpha + 1)$, *then* $\mathbb{E}\|\bar{f}_n^0 - f_{\mathcal{H}}\|^2_{L^2_{P_X}} \leq O\left(n^{-2r\alpha/(2r\alpha+1)}\right);$

*(iii) When* $1 < r \leq (\alpha + 2)/2$, *take* $\zeta = (\alpha + 1)/(2r\alpha + 1)$, *then* $\mathbb{E}\|\bar{f}_n^0 - f_{\mathcal{H}}\|^2_{L^2_{P_X}} \leq O\left(n^{-(2r\alpha-2r+2)/(2r\alpha+1)}\right);$

*(iv) When* $r > (\alpha + 2)/2$, *take* $\zeta = 1/(1 + \alpha)$ *then* $\mathbb{E}\|\bar{f}_n^0 - f_{\mathcal{H}}\|^2_{L^2_{P_X}} \leq O\left(n^{-\alpha/(1+\alpha)}\right).$

**Corollary 5** (Non-constant step size, without privacy). *Suppose Assumptions 1–7 hold. Consider the estimator $\bar{f}_n^0$ without privacy protection defined via the recursion (5). Take the step sizes $\gamma_i \asymp i^{-\zeta}$ with $\zeta \in (0,1)$, for $i = 1, 2, \ldots$. If $c^{-1}\gamma_0 B^2 < 1$ with $c$ defined in Lemma 1, and $2r - 1/(1-\zeta) < 0$, then:*
*(i) $0 \leq \zeta \leq 1/2$,*

$$\mathbb{E}\|\bar{f}_n^0 - f_{\mathcal{H}}\|_{L_{P_X}^2}^2 \leq O\left(\sigma^2 \gamma_n^{1/\alpha} n^{-1+1/\alpha} + \gamma_n^{-1}(n\gamma_n)^{-2r}\left\|\Sigma^{-r}f_{\mathcal{H}}\right\|_{L_{P_X}^2}^2\right);$$

*(ii) $1/2 < \zeta < 1$,*

$$\mathbb{E}\|\bar{f}_n^0 - f_{\mathcal{H}}\|_{L_{P_X}^2}^2 \leq O\left(\sigma^2 (n\gamma_n)^{-2+1/\alpha} + \gamma_n^{-1}(n\gamma_n)^{-2r}\left\|\Sigma^{-r}f_{\mathcal{H}}\right\|_{L_{P_X}^2}^2\right).$$

*Further,*
*(i) When $0 < r \leq (\alpha-1)/(2\alpha)$, take $\zeta = 0$, then $\mathbb{E}\|\bar{f}_n^0 - f_{\mathcal{H}}\|_{L_{P_X}^2}^2 \leq O\left(n^{-2r}\right)$;*
*(ii) When $(\alpha-1)/(2\alpha) < r < (1+\alpha)/(2\alpha)$, take $\zeta = (2r\alpha + 1 - \alpha)/(2r\alpha + 1 + \alpha)$, then $\mathbb{E}\|\bar{f}_n^0 - f_{\mathcal{H}}\|_{L_{P_X}^2}^2 \leq O\left(n^{-(2r\alpha+\alpha-1)/(2r\alpha+1+\alpha)}\right)$;*
*(iii) When $r \geq (1+\alpha)/(2\alpha)$, take $\zeta = 1/(1+\alpha)$, then $\mathbb{E}\|\bar{f}_n^0 - f_{\mathcal{H}}\|_{L_{P_X}^2}^2 \leq O\left(n^{-\alpha/(1+\alpha)}\right)$.*

We summarize the theoretical results to facilitate a comparison of the guarantees under private and non-private settings across different decay-rate regimes $r$, encompassing both constant and non-constant step-size schemes.

**Constant step size.** In the private setting, the error bound is

$$O\left(\left(\sigma^2 + \frac{8\tau^2 B^2 \log(2/\delta)}{\varepsilon^2}\right)\left(\gamma^{1/\alpha}n^{-1+1/\alpha} + n^{-1}\right) + (1+q(\gamma,n))\gamma^{-2r}n^{-2\min\{r,1\}}\left\|\Sigma^{-r}f_{\mathcal{H}}\right\|_{L_{P_X}^2}^2\right),$$

whereas in the non-private setting, the error bound is

$$O\left(\sigma^2\left(\gamma^{1/\alpha}n^{-1+1/\alpha} + n^{-1}\right) + (1+q(\gamma,n))\gamma^{-2r}n^{-2\min\{r,1\}}\left\|\Sigma^{-r}f_{\mathcal{H}}\right\|_{L_{P_X}^2}^2\right).$$

The optimal choices of $\zeta$ and the corresponding convergence rates across different ranges of $r$ are summarized in Table 3.

Table 3: Constant step size: optimal $\zeta$ and convergence rates.

| $r$ range | Optimal $\zeta$ in $\gamma_i \asymp n^{-\zeta}$ | Private / non-private convergence rate |
|---|---|---|
| $(0, (\alpha-1)/(2\alpha)]$ | $0$ | $O(n^{-2r})$ |
| $((\alpha-1)/(2\alpha), 1]$ | $(2r\alpha+1-\alpha)/(2r\alpha+1)$ | $O\left(n^{-2r\alpha/(2r\alpha+1)}\right)$ |
| $(1, (\alpha+2)/2]$ | $(\alpha+1)/(2r\alpha+1)$ | $O\left(n^{-(2r\alpha-2r+2)/(2r\alpha+1)}\right)$ |
| $((\alpha+2)/2, \infty)$ | $1/(1+\alpha)$ | $O\left(n^{-\alpha/(1+\alpha)}\right)$ |

**Non-constant step size.** In the private setting, the error bound is

$$\begin{cases} O\left(\left(\sigma^2 + \frac{8\tau^2 B^2 \log(2/\delta)}{\varepsilon^2}\right)\gamma_n^{1/\alpha}n^{-1+1/\alpha} + \gamma_n^{-1}(n\gamma_n)^{-2r}\left\|\Sigma^{-r}f_{\mathcal{H}}\right\|_{L_{P_X}^2}^2\right), & 0 \leq \zeta \leq 1/2, \\ O\left(\left(\sigma^2 + \frac{8\tau^2 B^2 \log(2/\delta)}{\varepsilon^2}\right)(n\gamma_n)^{-2+1/\alpha} + \gamma_n^{-1}(n\gamma_n)^{-2r}\left\|\Sigma^{-r}f_{\mathcal{H}}\right\|_{L_{P_X}^2}^2\right), & 1/2 < \zeta < 1, \end{cases}$$

while in the non-private setting, the error bound is

$$\begin{cases} O\left(\sigma^2 \gamma_n^{1/\alpha}n^{-1+1/\alpha} + \gamma_n^{-1}(n\gamma_n)^{-2r}\left\|\Sigma^{-r}f_{\mathcal{H}}\right\|_{L_{P_X}^2}^2\right), & 0 \leq \zeta \leq 1/2, \\ O\left(\sigma^2 (n\gamma_n)^{-2+1/\alpha} + \gamma_n^{-1}(n\gamma_n)^{-2r}\left\|\Sigma^{-r}f_{\mathcal{H}}\right\|_{L_{P_X}^2}^2\right), & 1/2 < \zeta < 1. \end{cases}$$

The optimal $\zeta$ and convergence rates are summarized in Table 4.

Table 4: Non-constant step size: optimal $\zeta$ and convergence rates.

| $r$ range | Optimal $\zeta$ in $\gamma_i \asymp n^{-\zeta}$ | Private / non-private convergence rate |
|---|---|---|
| $(0, (\alpha-1)/(2\alpha)]$ | $0$ | $O(n^{-2r})$ |
| $((\alpha-1)/(2\alpha), (1+\alpha)/(2\alpha))$ | $(2r\alpha+1-\alpha)/(2r\alpha+1+\alpha)$ | $O\big(n^{-(2r\alpha+\alpha-1)/(2r\alpha+1+\alpha)}\big)$ |
| $[(1+\alpha)/(2\alpha), \infty)$ | $1/(1+\alpha)$ | $O\big(n^{-\alpha/(1+\alpha)}\big)$ |

Overall, the privacy term introduces an additional additive factor of $(8\tau^2 B^2 \log(2/\delta))/(\varepsilon^2)$, which increases the variance constant but does not affect the asymptotic rates under optimal step sizes.

## E Robustness-privacy-utility trilemma

The interplay among the Huber parameter, the privacy budget, and the step size forms a fundamental robustness–privacy–utility trilemma. In the non-private setting, using a smaller Huber parameter $\tau$ improves robustness to outliers but increases bias, which is a classical bias–robustness trade-off [Fan et al., 2017]. In the private setting, however, a larger $\tau$ amplifies gradient sensitivity, requiring stronger noise for given $(\varepsilon, \delta)$-LDP, which raises gradient variance and necessitates smaller $\gamma_0$ for stability. Similarly, a tighter privacy budget (e.g., a smaller $\varepsilon$) demands stronger noise, again making a smaller $\gamma_0$ critical for stable convergence. The step size itself is typically chosen according to a decaying schedule, either $\gamma_i = \gamma_0 n^{-\zeta}$ for constant step size with sample size $n$ or $\gamma_i = \gamma_0 i^{-\zeta}$ for non-constant step size. As shown in Corollaries 1–5, the optimal choice of the decay exponent $\zeta$ depends on the smoothness parameters $r$ and $\alpha$. Under strong privacy noise, smaller $\gamma_0$ improves stability, but overly small steps slow convergence.

In practice, the three hyperparameters must be co-tuned carefully. The Huber parameter can be set following the procedure described in Algorithm 2, which builds on established methods in the Huber loss literature [Holland and Welsch, 1977]. The privacy budget is usually determined by application-specific requirements, but, whenever possible, relaxing the budget (choosing a larger $\varepsilon$) reduces the amount of noise added and thereby improves utility. In the absence of theory-guided tuning, we recommend a grid search over $(\gamma_0, \zeta)$ for the step size on a validation set, consistent with practical and widely used approaches in optimization [Ge et al., 2019, Attia and Koren, 2025].

## F Additional experimental results

### F.1 Experiment details

We generate i.i.d. samples from the model $Y = f^\star(X) + e$. We consider two examples of true functions $f^\star$ as follows:

$$\text{Case 1: } f_1^\star(x) = \sin(3\pi x/2),$$

$$\text{Case 2: } f_2^\star(x) = \frac{2}{3}\beta_{10,5}(x) + \frac{1}{3}\beta_{5,10}(x),$$

where $\beta_{p,q} = \frac{x^{p-1}(1-x)^{q-1}}{B(p,q)}$ with $B(p,q) = \frac{\Gamma(p)\Gamma(q)}{\Gamma(p+q)}$ denoting the beta function and $\Gamma$ is the gamma function with $\Gamma(p) = p!$ for $p \in \mathbb{N}_+$, and $\psi_{a,b}$ denotes the density function of $N(a, b^2)$. The first case is simple, and the second case is designed to mimic complex true function. The noise $e$ is set to two cases: $t(3)$ representing heavy-tailed noise, and $N(0, 0.25)$ representing the regular situation.

In simulation, we use the RKHS with the Gaussian kernel $K(x, y) = \exp\{-(x-y)^2/(2h^2)\}$ and the inner product: for any $f, g \in \mathcal{H}_K$, $\langle f, g \rangle = \int \frac{\hat{f}(w)\overline{\hat{g}(w)}}{\hat{K}(w)} dw$, where $\hat{f}(w) = \int_\mathbb{R} f(x)e^{-iwx}dx$, and $\hat{K}(w) = \sqrt{2\pi}h \exp\{-h^2 w^2/2\}$.

We conduct all three methods under both constant and non-constant step-size schemes. For the constant step size setting, we use a fixed step size of the form $\gamma_0 n^{-\zeta}$ with total sample sizes $n$, where $n$ is the total sample size; for the non-constant setting, the step size is set as $\gamma_i = \gamma_0 i^{-\zeta}$ for the $i$-th iteration.

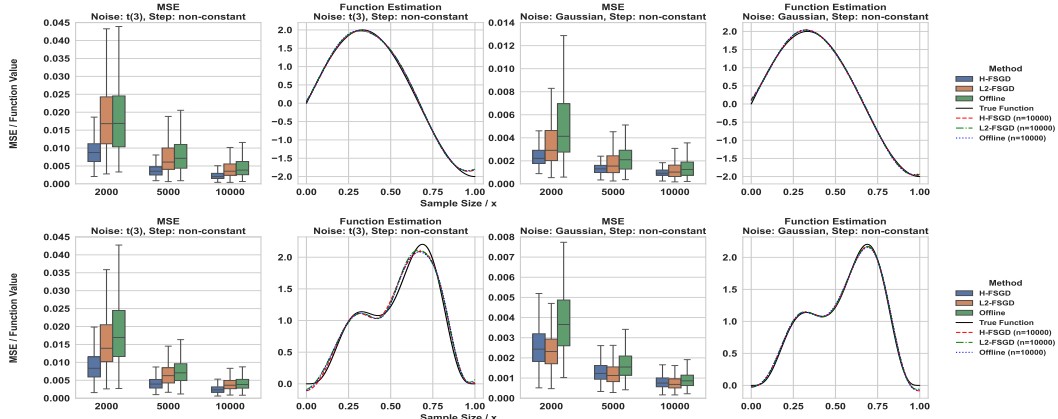

Figure 5: Box-plots and function fitting plots for Case 1 (top panels) and Case 2 (bottom panels) with the non-constant step size scheme in Example 5.1.

## F.2 Additional main results

As shown in Figure 5, when the error distribution exhibits heavy tails such as the $t(3)$ distribution, our H-FSGD algorithm also delivers markedly improved robustness compared to the standard least-squares-based FSGD under the non-constant step size scheme. The advantage is most pronounced at small sample sizes. In the case of Gaussian noise, H-FSGD maintains superior or competitive accuracy relative to benchmark methods. Furthermore, all methods exhibit decreasing MSEs with growing sample size, and they convergence to nearly ideal function approximations by $n = 10000$.

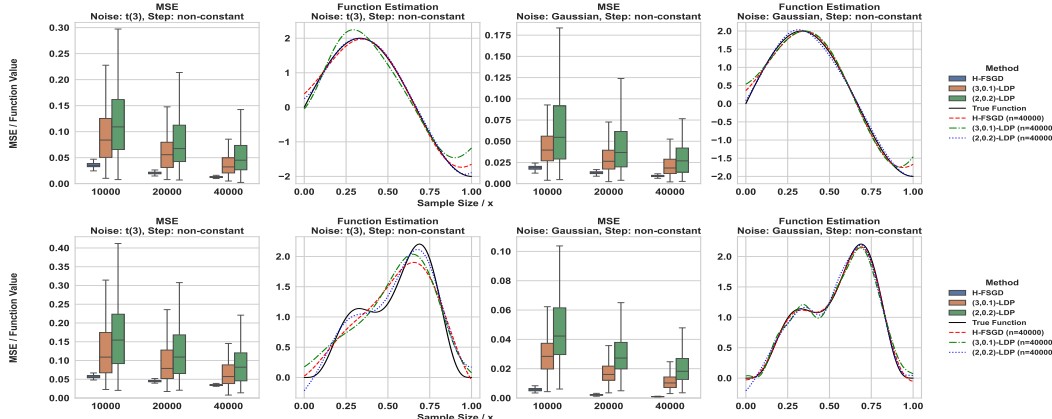

Figure 6: Box-plots and function fitting plots for Case 1 (top panels) and Case 2 (bottom panels) with the non-constant step size scheme in Example 5.2.

Figure 6 demonstrate similar patterns to Figure 3. Two LDP estimators exhibit increasing estimation error as the level of privacy protection intensifies, which illustrates the classic tension between privacy and utility. Nonetheless, all estimators, including those under stringent LDP, are still capable of capturing the underlying functional form. As sample size grows, the performance gap between private and non-private methods narrows significantly. This trend underscores how larger datasets can mitigate the negative impact of privacy-preserving mechanisms on estimation accuracy.

## F.3 Sensitivity of step sizes

We have conducted additional simulation studies to examine the sensitivity of estimation performance with respect to different values of step size. We examine the H-FSGD performance under Case 1 for both constant and non-constant step size settings, with $\gamma_0 \in [4, 24]$ and $\zeta \in [0.3, 0.8]$. Figure 7 illustrates the heatmaps of average MSEs over 50 repetitions when the sample size is $n = 10000$.

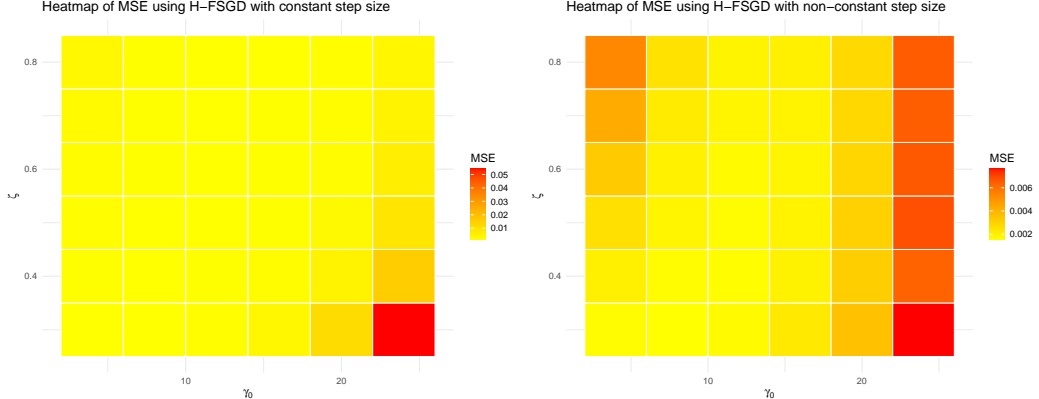

Figure 7: Sensitivity analysis of the step-size parameters $\gamma_0$ and $\zeta$. Results under the constant step-size scheme are presented in the left panel, whereas those under the non-constant step-size scheme are shown in the right panel.

Figure 7 indicates that the proposed estimators exhibit robustness to the choice of $\gamma_0$ and $\zeta$ under both constant and non-constant step size settings, with wide parameter regions yielding stable and comparable MSEs.

## F.4 Performance beyond theoretical assumptions

While our theoretical analysis requires finite-variance noise condition, we also conduct experiments with heavier-tailed noise, including Student-$t(2.5)$ and Cauchy$(0,1)$ noises. Tables 5 and 6 show MSE ($\times 10^{-3}$) means and standard deviations over 200 repetitions for $n = 10000, 20000$, and $40000$, under both constant and non-constant step sizes.

Table 5: MSE ($\times 10^{-3}$) with standard deviations in parentheses ($\times 10^{-3}$), computed over 200 repetitions under $t(2.5)$ noise.

| Step size | Constant | | | Non-constant | | |
|---|---|---|---|---|---|---|
| $n$ | 10000 | 20000 | 40000 | 10000 | 20000 | 40000 |
| H-FSGD | 2.15 (0.758) | 1.25 (0.415) | 0.669 (0.214) | 2.26 (1.06) | 1.41 (0.682) | 0.907 (0.467) |
| L2-FSGD | 3.97 (2.25) | 2.04 (1.20) | 1.11 (0.551) | 6.13 (4.40) | 3.38 (2.89) | 2.02 (1.71) |
| Offline | 4.21 (2.17) | 2.17 (1.26) | 1.16 (0.569) | 6.75 (5.49) | 3.65 (3.25) | 2.12 (1.85) |

Our experimental results demonstrate that H-PSGD maintains superior robustness compared to both L2-FSGD and offline methods, even when handling infinite-variance distributions that fall outside our current theoretical framework. While formal guarantees for these extreme cases remain to be established, the observed empirical performance strongly motivates future theoretical investigation of such challenging scenarios.

## F.5 Robustness in contamination models

While our method is designed for heavy-tailed noise via the Huber loss, we also assess its robustness under Huber's $\varepsilon^\star$-contamination model, where an $\varepsilon^\star$-fraction of data is adversarially corrupted. Using Case 1 (true model) and Case 2 (contaminated model), we simulate performance across varying $\varepsilon^\star$. Table 7 reports the mean and standard deviation of the MSE ($\times 10^{-2}$) over 50 repetitions for both constant and non-constant step sizes at $n = 10000$.

Table 6: MSE ($\times 10^{-3}$) with standard deviations in parentheses ($\times 10^{-3}$), computed over 200 repetitions under Cauchy$(0, 1)$ noise.

| Step size | Constant | | | Non-constant | | |
|---|---|---|---|---|---|---|
| $n$ | 10000 | 20000 | 40000 | 10000 | 20000 | 40000 |
| H-FSGD | 3.64 (1.57) | 1.97 (0.816) | 1 (0.435) | 5.22 (3.25) | 2.98 (1.81) | 1.65 (1.03) |
| L2-FSGD | 5.83e+07 (8.2e+08) | 1.89e+07 (2.65e+08) | 5.7e+06 (7.89e+07) | 2.24e+08 (3.17e+09) | 1.01e+08 (1.42e+09) | 4.36e+07 (6.15e+08) |
| Offline | 1.73e+08 (2.45e+09) | 4.86e+07 (6.85e+08) | 1.33e+07 (1.86e+08) | 3.67e+08 (5.19e+09) | 1.38e+08 (1.95e+09) | 5.32e+07 (7.51e+08) |

Table 7: Performance comparison under different contamination levels.

| Step size | $\varepsilon^\star$ | 0 | 0.1 | 0.2 | 0.3 | 0.4 |
|---|---|---|---|---|---|---|
| Constant | H-FSGD | 0.0722 (0.0158) | 0.657 (0.0985) | 2.85 (0.251) | 8.08 (0.497) | 20.4 (1.27) |
| | L2-FSGD | 0.0619 (0.0150) | 2.46 (0.263) | 9.49 (0.611) | 21.2 (0.928) | 37.4 (1.16) |
| Non-constant | H-FSGD | 0.0939 (0.0326) | 0.683 (0.105) | 2.89 (0.260) | 8.11 (0.517) | 20.2 (1.23) |
| | L2-FSGD | 0.152 (0.136) | 2.55 (0.310) | 9.63 (0.683) | 21.3 (1.01) | 37.5 (1.23) |

| Step size | $\varepsilon^\star$ | 0.5 | 0.6 | 0.7 | 0.8 | 0.9 |
|---|---|---|---|---|---|---|
| Constant | H-FSGD | 56.7 (3.91) | 114 (3.47) | 155 (2.49) | 186 (2.21) | 211 (2.10) |
| | L2-FSGD | 58.4 (1.60) | 84 (1.70) | 114 (1.74) | 149 (2.15) | 189 (2.14) |
| Non-constant | H-FSGD | 56.4 (4.23) | 114 (3.78) | 155 (2.61) | 186 (2.32) | 211 (2.16) |
| | L2-FSGD | 58.5 (1.64) | 84.2 (1.83) | 114 (1.94) | 149 (2.25) | 189 (2.18) |

As shown in Table 7, H-FSGD remains stable up to $\varepsilon^\star = 0.3$, with degradation beginning around $\varepsilon^\star = 0.4$, suggesting a breakdown point near $40\%$. In contrast, L2-FSGD deteriorates earlier at $\varepsilon^\star = 0.3$. Across all settings, H-FSGD consistently outperforms L2-FSGD.

## G   Applications

Many real-world applications simultaneously require privacy protection, robustness to heavy-tailed noise, and nonparametric modeling. Examples include:

- Healthcare analytics. Wearable fitness data (e.g., heart rate, sleep patterns) demands LDP since raw physiological signals can re-identify users; exhibits heavy-tailed noise due to irregular activities (e.g., sudden spikes in heart rate during exercise or sensor artifacts); and necessitates nonparametric regression because the relationship between metrics (e.g., sleep duration vs. recovery rate) is often nonlinear and complex.

- Financial fraud detection. Transaction histories require LDP when shared due to their identifiable nature, legitimate spending patterns contaminated by fraudulent outliers create heavy-tailed distributions, and the adversarial nature of fraud evolution necessitates nonparametric methods to detect novel attack patterns beyond rigid rule-based systems.

- Consumer behavior analysis. Browsing logs need LDP protection against profiling, purchase amounts exhibit heavy-tailed distributions dominated by rare large orders, and market segmentation reveals irregular price elasticity patterns that nonparametric models can adequately capture.

To better validate our method's practical utility, we conducted experiments on the real data "Health and fitness dataset" from Kaggle website [3]. Our goal is to investigate how endurance levels affect overall fitness. We select 40000 samples as the training set and 1000 samples as the test set. We compared our H-FSGD and PH-FSGD methods with baseline methods on out-of-sample $R^2$ performance, as displayed in Table 8.

Table 8: Out-of-sample $R^2$ of proposed and baseline methods.

| Step size | H-FSGD | L2-FSGD | Offline | (2,0.2)-LDP | (3,0.1)-LDP | lm |
|---|---|---|---|---|---|---|
| Constant | 0.622 | 0.617 | 0.609 | 0.589 | 0.609 | 0.581 |
| Non-constant | 0.623 | 0.618 | 0.609 | 0.526 | 0.596 | 0.581 |

The results show that H-FSGD consistently outperforms L2-FSGD and Offline methods, showing strong robustness. All non-private nonparametric methods outperform linear regression (lm) in capturing complex relationships. Under LDP, PH-FSGD remains competitive, matching or exceeding lm, confirming its effectiveness under privacy constraints. These findings validate our method's ability to model nonlinear patterns while preserving privacy.

## H   Discussions

### H.1   Computational complexity

Our theoretical analysis is conducted directly in the RKHS, where finite-sample bounds depend on the kernel eigenvalue decay rate $\alpha$ rather than the grid size $J$. The grid size $J$ primarily influences computational costs, reflecting the inherent tradeoff between accuracy and efficiency.

In the univariate setting, the computational complexity remains tractable: constructing the covariance matrix and performing Cholesky decomposition require $O(J^3)$ setup time, while $n$ functional SGD iterations incur an additional $O(nJ^2)$ cost with $O(J^2)$ storage. However, we fully acknowledge that the situation changes drastically in the multivariate case. A $d$-dimensional problem would demand $O(J^{3d})$ setup time, $O(nJ^{2d})$ operations, and $O(J^{2d})$ storage, rendering naive grid-based approaches impractical for $d \geq 3$.

To mitigate this curse of dimensionality, additive kernel methods [Raskutti et al., 2012] provide a scalable alternative. By decomposing the problem dimension-wise, the computational costs are

---

[3]https://www.kaggle.com/datasets/evan65549/health-and-fitness-dataset

reduced to $O(dJ^3 + ndJ^2)$ with storage $O(dJ^2)$, thereby preserving theoretical guarantees while ensuring practical feasibility.

## H.2 Minibatching integration

Our framework can be easily extended to support minibatching. Specifically, at iteration $n$, if a mini-batch $\{(X_{nt}, Y_{nt})\}_{t=1}^{B_n}$ with size $B_n$ is available, the update rule becomes:

$$\hat{f}_n = \hat{f}_{n-1} + \gamma_n \frac{1}{B_n} \sum_{t=1}^{B_n} w_\tau \left( Y_{nt} - \langle \hat{f}_{n-1}, K_{X_{nt}} \rangle_{\mathcal{H}} \right) \left( Y_{nt} - \langle \hat{f}_{n-1}, K_{X_{nt}} \rangle_{\mathcal{H}} \right) K_{X_{nt}} + \gamma_n \xi_n,$$

$$\bar{f}_n = \frac{n-1}{n} \bar{f}_{n-1} + \frac{1}{n} \hat{f}_n,$$

where $\xi_n$ is the sample path of a Gaussian process having mean zero and covariance function $\frac{8\tau^2 B^2 \log(2/\delta_n)}{\varepsilon_n^2} K$ with $(\varepsilon_n, \delta_n)$ being the privacy budget at the $n$-th iteration. The noise term $\xi_n$ still suffices to ensure DP, as the modified Fréchet gradient has the same sensitivity and the privacy mechanism remains applicable in this case. Moreover, the theoretical analysis can be readily modified by replacing the single-sample gradient with its minibatch-averaged counterpart. Minibatching may in fact help improve the privacy–utility trade-off by reducing gradient variance, allowing for smaller noise magnitudes under the same privacy budget; however, it also requires a trusted data collector to aggregate and access the mini-batch data.

## H.3 Extension to non-i.i.d. data

Addressing concept drift and non-i.i.d. data is both important and challenging. Our current framework, like standard functional SGD theory, assumes i.i.d. data, and does not directly extend to non-i.i.d. settings, which is a limitation shared by many SGD-based methods. challenges include parameter drift, slower convergence [Zhao et al., 2018, Li et al., 2020], and bias from dependent observations. While one could define the population risk as

$$\sum_{n=1}^{\infty} p_n L_\tau (Y_n - f_n(X_n))$$

with distribution weight $p_n$, this is infeasible in online settings due to unknown distribution shifts and the ill-defined infinite-sum objective. A more practical alternative models the data as coming from $M$ sub-populations with different distributions, minimizing the global risk

$$\sum_{m=1}^{M} p_m L_\tau (Y^{(m)} - f(X^{(m)})).$$

This leads to a parallel SGD scheme: at each round $t$, the global estimate $\hat{f}_t$ is sent to local devices, which compute updates

$$\hat{f}_{t+1}^{(m)} = \hat{f}_t - \gamma_t \widehat{\nabla L_\tau}(\hat{f}_t)(X_n^{(m)}, Y_n^{(m)}) + \gamma_t \xi_t,$$

followed by weighted aggregation

$$\hat{f}_{t+1} = \sum_{m=1}^{M} p_m \hat{f}_{t+1}^{(m)}.$$

This setup accommodates distributional heterogeneity and aligns with recent federated learning approaches for non-i.i.d. data [Chen et al., 2020, Ma et al., 2022].

# I Limitations

The first limitation of this work lies in its exclusive focus on the Gaussian mechanism for implementing LDP, without exploring alternative mechanisms such as the exponential mechanism [Awan et al., 2019] or the Laplace mechanism [Lin and Reimherr, 2024]. While the Gaussian mechanism provides

desirable analytical properties and facilitates rigorous theoretical analysis, this choice may reduce the flexibility of our framework in settings where other mechanisms are better suited. We acknowledge this constraint; however, to the best of our knowledge, our work is the first to adapt the Gaussian mechanism into online nonparametric regression under LDP, providing a novel and theoretically grounded approach to privacy-preserving estimation in streaming environments. The second limitation concerns the scope of our loss function, which is restricted to the Huber loss rather than a broader class of M-estimators. This modeling decision is primarily driven by two technical challenges: (i) quantifying sensitivity for general M-estimators under LDP is nontrivial, and (ii) most M-estimators lack closed-form update rules, posing significant challenges for theoretical convergence analysis in online learning frameworks. Future research may address these limitations by extending the proposed framework to incorporate alternative privacy mechanisms and more general robust loss functions.

## J Technical proofs

### J.1 Proof of Equation (4)

By the definition of the Huber loss (2), the estimator of the Fréchet gradient is

$$
\widehat{\nabla L_\tau}(f)(x, y) = \begin{cases} -(y - f(x))\, K_x, & |y - f(x)| \leq \tau \\ -\tau \cdot \operatorname{sign}(y - f(x)) K_x, & |y - f(x)| > \tau \end{cases}
$$
$$
= -\left[ (y - f(x))\, I\left\{ |y - f(x)| \leq \tau \right\} + \tau \cdot \operatorname{sign}(y - f(x)) I\left\{ |y - f(x)| > \tau \right\} \right] K_x
$$
$$
=: -\ell_\tau(y - f(x)) K_x,
$$

where $K_x(\cdot) = K(x, \cdot)$, and $\ell_\tau(u) := uI\left\{ |u| \leq \tau \right\} + \tau \cdot \operatorname{sign}(u) I\left\{ |u| > \tau \right\}$. Note that $\ell_\tau(u) = u w_\tau(u)$ with $w_\tau(u) = \min\{1, \tau/|u|\}$, then

$$
\widehat{\nabla L_\tau}(f)(x, y) = -w_\tau(y - f(x))(y - f(x)) K_x.
$$

### J.2 Proof of outlier robustness

Outlier robustness in statistics and machine learning is typically characterized by the influence function [Hampel et al., 1986, Avella-Medina, 2021], which we formally define below.

**Definition 4** (Robust statistics). *Let $\mathscr{F}$ be a space of probability distributions on $\mathcal{Z} = \mathcal{X} \times \mathbb{R}$, and let $T : \mathscr{F} \to \mathcal{G}$ be a functional mapping each $F \in \mathscr{F}$ to a function $T(F) = f_F \in \mathcal{G}$. For any contamination point $z = (x', y') \in \mathcal{Z}$, define the contaminated distribution $F_\epsilon = (1 - \epsilon)F + \epsilon\Delta_z$, where $\Delta_z$ is the point mass at $z$ and $\epsilon \in (0, 1)$. Then the influence function of $T$ at $z$ under $F$ is the function $\operatorname{IF}(z; T, F) : \mathcal{X} \to \mathbb{R}$ given pointwise by*

$$
\operatorname{IF}(z; T, F)(x) := \lim_{\epsilon \to 0^+} \frac{T(F_\epsilon)(x) - T(F)(x)}{\epsilon} = \lim_{\epsilon \to 0^+} \frac{f_{F_\epsilon}(x) - f_F(x)}{\epsilon}.
$$

*If the influence function is uniformly bounded, then the statistics $T(F)$ are considered robust, as no single outlier can have a disproportionate effect on the estimator.*

We now rigorously show that the Huber loss yields an estimator with a uniformly bounded influence function in the nonparametric setting. Consider the nonparametric regression model $Y = f(X) + e$ and the estimator $\hat{f} = \arg\min_f \mathbb{E}_F[L_\tau(Y - f(X))]$, where the Huber loss is

$$
L_\tau(u) = \begin{cases} \dfrac{1}{2}u^2, & |u| \leq \tau, \\ \tau|u| - \dfrac{1}{2}\tau^2, & |u| > \tau, \end{cases}
$$

with $\psi_\tau(u) = \frac{d}{du}L_\tau(u) = \begin{cases} u, & |u| \leq \tau, \\ \tau \operatorname{sign}(u), & |u| > \tau, \end{cases}$ and $\psi_\tau'(u) = \frac{d}{du}\psi_\tau(u) = \begin{cases} 1, & |u| < \tau, \\ 0, & |u| \geq \tau. \end{cases}$

Define the functional $T_{x_0}$ on the joint distribution $F$ of $(X, Y)$ by $T_{x_0}(F) = f_F(x_0)$, where $f_F(\cdot)$ satisfies the population estimating equation

$$
\Psi_{x_0}(t; F) := \int \psi_\tau(y - t) F_{Y|X=x_0}(dy) = 0.
$$

Let a contaminated distribution be $F_\epsilon = (1 - \epsilon)F + \epsilon\Delta_{(x',y')}$, where $\Delta_{(x',y')}$ is a point mass at $(x', y')$. The influence function at $(x', y')$ is defined by

$$\text{IF}((x',y'); T_{x_0}, F) = \lim_{\epsilon \to 0^+} \frac{T_{x_0}(F_\epsilon) - T_{x_0}(F)}{\epsilon}.$$

Since $\Psi_{x_0}\big(T_{x_0}(F_\epsilon); F_\epsilon\big) = 0$, by differentiating with respect to $\epsilon$ at $\epsilon = 0$ (implicit function theorem), we obtain

$$\frac{\partial \Psi_{x_0}}{\partial t}\bigg|_{t=f_F(x_0)} (T_{x_0}(F_\epsilon) - T_{x_0}(F)) + \Psi_{x_0}\big(f_F(x_0); F_\epsilon\big) = o(\epsilon).$$

Noting $\frac{\partial \Psi_{x_0}}{\partial t} = -\int \psi'_\tau(y - f_F(x_0))F_{Y|X=x_0}(dy)$, and $\Psi_{x_0}(f_F(x_0); F_\epsilon) = \epsilon\psi_\tau(y' - f_F(x_0))I_{\{x'=x_0\}}$, we solve to get

$$\text{IF}((x',y'); T_{x_0}, F) = \frac{\psi_\tau(y' - f_F(x_0))I_{\{x'=x_0\}}}{\int \psi'_\tau(y - f_F(x_0))F_{Y|X=x_0}(dy)}.$$

Note that $|\psi_k(y' - f_F(x_0))| \leq \tau$, $|I_{\{x'=x_0\}}| \leq 1$, and $D(x_0) := \int \psi'_\tau(y - f_F(x_0))F_{Y|X=x_0}(dy) = \mathbb{P}(|Y - f_F(x_0)| < \tau \mid X = x_0)$. Under the usual assumption that the conditional density of $r = Y - f(X)$ is continuous and strictly positive at zero, there exists a universal constant $c > 0$ such that $D(x_0) \geq c$ for all $x_0$. Hence, for all contamination points $(x', y')$ and all $x_0$,

$$|\text{IF}((x',y'); T_{x_0}, F)| \leq \frac{\tau}{c},$$

which does not depend on $(x', y')$, proving that the influence function is uniformly bounded. This establishes that our estimator based on the Huber loss is provably robust to outliers.

### J.3   Proof of Theorem 1

Fixing $\hat{f}_{n-1}$, the Fréchet gradient iterating $\hat{f}_n$ is

$$g_n(X_n, Y_n) = -\ell_\tau(Y_n - \hat{f}_{n-1}(X_n))K_{X_n}.$$

View $g$ as an operator such that $g(x, y) = -\ell_\tau(y - \hat{f}_{n-1}(x))K_x$. By Assumption 2 and the definition of $\ell_\tau(\cdot)$, we have

$$\sup_{z=(x,y),z'=(x',y')} \|g(x,y) - g(x',y')\| \leq 2\tau B.$$

Applying Proposition 1 with single sample $D = \{(X_n, Y_n)\}$, if $\xi_n$ is taken as the sample path of a Gaussian process having mean zero and covariance function $\frac{8\tau^2 B^2 \log(2/\delta_n)}{\varepsilon_n^2}K$, then the next iterate $\hat{f}_n$ is $(\varepsilon_n, \delta_n)$-LDP. Utilizing the parallel composition property of LDP stated in Proposition 2 (1), we observe that releasing each update based on disjoint subsets of the data does not amplify the overall privacy loss. Consequently, for any $n \in \mathbb{N}$, both the estimator $\hat{f}_n$ and its averaged counterpart $\bar{f}_n$ inherit $(\max_{1 \leq i \leq n}\{\varepsilon_i\}, \max_{1 \leq i \leq n}\{\delta_i\})$-LDP from each individual update.

### J.4   Proof of Theorem 2

Lemma 1 and the following Lemmas are usefull to prove Theorem 2.

*Proof of Lemma 1.* Recall that $w_\tau(s) = \min\{1, \frac{\tau}{|s|}\}$. Assumptions 2 and 4 imply that $\|f\|_\infty \leq BM$, then

$$|Y_n - f(X_n)| \leq \|f^\star\|_\infty + \|f\|_\infty + |e_n| \leq \|f^\star\|_\infty + BM + |e_n|,$$

it follows that $w_\tau(Y_n - f(X_n)) \geq \min\{1, \frac{\tau}{|e_n|+BM+\|f^\star\|_\infty}\}$. Utilizing Assumption 5, we have

$$\mathcal{T}_f \succcurlyeq \mathbb{E}\left(\min\left\{1, \frac{\tau}{|e_n| + BM + \|f^\star\|_\infty}\right\}\right)\mathbb{E}(K_{X_n} \otimes K_{X_n})$$

$$\succcurlyeq 2m\kappa \min\left\{1, \frac{\tau}{m + BM + \|f^\star\|_\infty}\right\}\Sigma =: c\Sigma.$$

$\square$

**Lemma 2.** *Let* $\alpha_n = (I - \gamma \mathcal{T}_{n-1}) \alpha_{n-1} + \gamma \Xi_n^\alpha$. $\mathcal{T}_{n-1}$ *satisfies* $\mathcal{T}_{n-1} \preccurlyeq \Sigma$ *with* $\gamma \Sigma \preccurlyeq I$, *where* $\preccurlyeq$ *denotes the order between self-adjoint operators.* $\Xi_n^\alpha \in \mathcal{H}$ *is* $\mathcal{F}_n$ *measurable for a sequence of increasing* $\sigma$-*fields* $\{\mathcal{F}_n\}_n$, $\mathbb{E}(\|\Xi_n^\alpha\|^2 | \mathcal{F}_{n-1})$ *is finite, and* $\mathbb{E}(\Xi_n^\alpha \otimes \Xi_n^\alpha) \preccurlyeq \sigma_\alpha^2 \Sigma$. *Then*

$$\mathbb{E}\langle \bar{\alpha}_n, \Sigma \bar{\alpha}_n \rangle_{L_{P_X}^2} \leq Bias(n, \gamma, \Sigma, \{\mathcal{T}_i\}_i, \alpha_0) + Var(n, \gamma, \Sigma, \{\mathcal{T}_i\}_i, \{\Xi_i^\alpha\}_i),$$

*where* $\bar{\alpha}_n = \sum_{j=1}^n \alpha_n$,

$$Bias(n, \gamma, \Sigma, \{\mathcal{T}_i\}_i, \alpha_0) = \frac{2}{n^2} \mathbb{E} \left\| \Sigma^{1/2} \sum_{j=1}^n \prod_{i=1}^j (I - \gamma \mathcal{T}_{i-1}) \alpha_0 \right\|_\Sigma^2,$$

*and*

$$Var(n, \gamma, \Sigma, \{\mathcal{T}_i\}_i, \{\Xi_i^\alpha\}_i) = \frac{2}{n^2} \mathbb{E} \left\| \Sigma^{1/2} \sum_{j=1}^n \sum_{k=1}^j \prod_{i=k+1}^j (I - \gamma \mathcal{T}_{i-1}) \gamma \Xi_k^\alpha \right\|_\Sigma^2.$$

*Proof of Lemma 2.* By the recursion of $\alpha_n$, we have

$$\bar{\alpha}_n = \frac{1}{n} \left( \sum_{j=1}^n \prod_{i=1}^j (I - \gamma \mathcal{T}_{i-1}) \alpha_0 + \sum_{j=1}^n \sum_{k=1}^j \prod_{i=k+1}^j (I - \gamma \mathcal{T}_{i-1}) \gamma \Xi_k^\alpha \right).$$

Then

$$\mathbb{E} \|\bar{\alpha}_n\|_{L_{P_X}^2}^2 \leq \frac{2}{n^2} \mathbb{E} \left\| \Sigma^{1/2} \sum_{j=1}^n \prod_{i=1}^j (I - \gamma \mathcal{T}_{i-1}) \alpha_0 \right\|_\Sigma^2 + \frac{2}{n^2} \mathbb{E} \left\| \Sigma^{1/2} \sum_{j=1}^n \sum_{k=1}^j \prod_{i=k+1}^j (I - \gamma \mathcal{T}_{i-1}) \gamma \Xi_k^\alpha \right\|_\Sigma^2$$

$$=: \text{Bias}(n, \gamma, \Sigma, \{\mathcal{T}_i\}_i, \alpha_0) + \text{Var}(n, \gamma, \Sigma, \{\mathcal{T}_i\}_i, \{\Xi_i^\alpha\}_i).$$

$\square$

**Lemma 3.** *Under Assumptions 2–5, for any* $r \geq 0$ *and any* $n \geq 0$, $\mathbb{E}(\Xi_n^r \otimes \Xi_n^r) \preccurlyeq c^{-r} \gamma^r B^{2r} \widetilde{\sigma}^2 \Sigma$, *and* $\mathbb{E}\left(\eta_n^{noise,r} \otimes \eta_n^{noise,r}\right) \preccurlyeq c^{-(r+1)} \gamma^{r+1} B^{2r} \widetilde{\sigma}^2 I$, *where* $\widetilde{\sigma}^2 = \sigma^2 + \frac{8\tau^2 B^2 \log(2/\delta)}{\varepsilon^2}$, *and* $c$ *is defined as in Lemma 1.*

*Proof of Lemma 3.* We make an induction on $r$. For $r = 0$, using Cauchy-Schwarz inequality and Assumption 3, we have

$$\mathbb{E}(\Xi_n^0 \otimes \Xi_n^0) \preccurlyeq \mathbb{E}\left[ (Y_n - f_\mathcal{H}(X_n))^2 K_{X_n} \otimes K_{X_n} \right] + \mathbb{E}(\xi_n \otimes \xi_n)$$

$$\preccurlyeq \mathbb{E}(e_n^2) \mathbb{E}(K_{X_n} \otimes K_{X_n}) + \frac{8\tau^2 B^2 \log(2/\delta)}{\varepsilon^2} K$$

$$\preccurlyeq \left( \sigma^2 + \frac{8\tau^2 B^2 \log(2/\delta)}{\varepsilon^2} \right) \Sigma$$

$$=: \widetilde{\sigma}^2 \Sigma,$$

where $\widetilde{\sigma}^2 = \sigma^2 + \frac{8\tau^2 B^2 \log(2/\delta)}{\varepsilon^2}$. The recursion formula of $\eta_n^{\text{noise},0}$ implies that

$$\eta_n^{\text{noise},0} = \sum_{k=1}^n \prod_{i=k+1}^n (I - \gamma \mathcal{T}_{i-1}) \gamma \Xi_k^0.$$

Then

$$\mathbb{E}\left( \eta_n^{\text{noise},0} \otimes \eta_n^{\text{noise},0} | \mathcal{F}_{n-1} \right) = \sum_{k=1}^n \prod_{i=k+1}^n (I - \gamma \mathcal{T}_{i-1}) \gamma^2 \mathbb{E}(\Xi_n^0 \otimes \Xi_n^0) \prod_{i=k+1}^n (I - \gamma \mathcal{T}_{i-1})$$

$$\preccurlyeq \widetilde{\sigma}^2 \sum_{k=1}^n \prod_{i=k+1}^n (I - \gamma \mathcal{T}_{i-1}) \gamma^2 \Sigma \prod_{i=k+1}^n (I - \gamma \mathcal{T}_{i-1}).$$

Utilizing Lemma 1, we have $\mathcal{T}_n \succcurlyeq c\Sigma$ for any $n$, then

$$\prod_{i=k+1}^{n} (I - \gamma\mathcal{T}_{i-1}) - \prod_{i=k}^{n} (I - \gamma\mathcal{T}_{i-1}) = \prod_{i=k+1}^{n} (I - \gamma\mathcal{T}_{i-1})\gamma\mathcal{T}_{k-1} \succcurlyeq c \prod_{i=k+1}^{n} (I - \gamma\mathcal{T}_{i-1})\gamma\Sigma.$$

It follows that

$$\sum_{k=1}^{n} \prod_{i=k+1}^{n} (I - \gamma\mathcal{T}_{i-1})\gamma^2\Sigma \prod_{i=k+1}^{n} (I - \gamma\mathcal{T}_{i-1}) \preccurlyeq \gamma \sum_{k=1}^{n} \prod_{i=k+1}^{n} (I - \gamma\mathcal{T}_{i-1})\gamma\Sigma$$

$$\preccurlyeq c^{-1}\gamma \sum_{k=1}^{n} \left[ \prod_{i=k+1}^{n} (I - \gamma\mathcal{T}_{i-1}) - \prod_{i=k}^{n} (I - \gamma\mathcal{T}_{i-1}) \right] \preccurlyeq c^{-1}\gamma I.$$

Then

$$\mathbb{E}\left(\eta_n^{\text{noise},0} \otimes \eta_n^{\text{noise},0}\right) \preccurlyeq c^{-1}\gamma\widetilde{\sigma}^2 I.$$

Assume that for any $n \geq 0$, $\mathbb{E}(\Xi_n^r \otimes \Xi_n^r) \preccurlyeq c^{-r}\gamma^r B^{2r}\widetilde{\sigma}^2\Sigma$, and $\mathbb{E}\left(\eta_n^{\text{noise},r} \otimes \eta_n^{\text{noise},r}\right) \preccurlyeq c^{-(r+1)}\gamma^{r+1}B^{2r}\widetilde{\sigma}^2 I$. We now consider $\mathbb{E}(\Xi_n^{r+1} \otimes \Xi_n^{r+1})$ and $\mathbb{E}\left(\eta_n^{\text{noise},r+1} \otimes \eta_n^{\text{noise},r+1}\right)$. Recall that $\Xi_n^{r+1} = \left(\mathcal{T}_{n-1} - w_\tau(Y_n - \hat{f}_{n-1}(X_n))K_{X_n} \otimes K_{X_n}\right)\eta_{n-1}^{\text{noise},r}$. By induction and Assumption 2,

$$\mathbb{E}(\Xi_n^{r+1} \otimes \Xi_n^{r+1}) \preccurlyeq c^{-(r+1)}\gamma^{r+1}B^{2r}\widetilde{\sigma}^2\mathbb{E}\left[\left(\mathcal{T}_{n-1} - w_\tau(Y_n - \hat{f}_{n-1}(X_n))K_{X_n} \otimes K_{X_n}\right)^2 \Big| \mathcal{F}_{n-1}\right]$$

$$\preccurlyeq c^{-(r+1)}\gamma^{r+1}B^{2r}\widetilde{\sigma}^2\mathbb{E}\left[(K_{X_n} \otimes K_{X_n})^2\right]$$

$$\preccurlyeq c^{-(r+1)}\gamma^{r+1}B^{2(r+1)}\widetilde{\sigma}^2\Sigma.$$

It follows that

$$\mathbb{E}(\eta_n^{\text{noise},r+1} \otimes \eta_n^{\text{noise},r+1})$$

$$\preccurlyeq c^{-(r+1)}\gamma^{r+1}B^{2(r+1)}\widetilde{\sigma}^2 \sum_{k=1}^{n} \prod_{i=k+1}^{n} (I - \gamma\mathcal{T}_{i-1})\gamma^2\Sigma \prod_{i=k+1}^{n} (I - \gamma\mathcal{T}_{i-1})$$

$$\preccurlyeq c^{-(r+2)}\gamma^{r+2}B^{2(r+1)}\widetilde{\sigma}^2 I.$$

$\square$

**Lemma 4.** *Assume that $(X_n, \hat{f}_n, \Xi_n^\alpha) \in \mathcal{H} \times \mathcal{H} \times \mathcal{H}$ is $\mathcal{F}_n$ measurable for a sequence of increasing $\sigma$-fields $\{\mathcal{F}_n\}_n$. Further, $\mathbb{E}(\Xi_n^\alpha|\mathcal{F}_{n-1}) = 0$, $\mathbb{E}(\|\Xi_n^\alpha\|^2|\mathcal{F}_{n-1}) < \infty$, and $\mathbb{E}(\|K_{X_n}\|^2 K_{X_n} \otimes K_{X_n}|\mathcal{F}_{n-1}) \preccurlyeq B^2\Sigma$ with $\mathbb{E}(K_{X_n} \otimes K_{X_n}|\mathcal{F}_{n-1}) = \Sigma$ for all $n \geq 1$, some constant $B > 0$ and invertible operator $\Sigma$. Let $\alpha_n = \left(I - \gamma w_\tau(Y_n - \hat{f}_{n-1}(X_n))K_{X_n} \otimes K_{X_n}\right)\alpha_{n-1} + \gamma\Xi_n^\alpha$, with $\alpha_0 = 0$ and $\gamma c^{-1}B^2 < 1$. Then*

$$\mathbb{E}\langle\bar{\alpha}_{n-1}, \Sigma\bar{\alpha}_{n-1}\rangle_{L_{P_X}^2} \leq \frac{1}{c(1 - \gamma c^{-1}B^2)}\frac{\gamma}{n}\sum_{k=1}^{n} \mathbb{E}\|\Xi_k^\alpha\|_{L_{P_X}^2}^2.$$

*Proof of Lemma 4.* The recursion formula implies that

$$\|\alpha_n\|_{L_{P_X}^2}^2 \leq \|\alpha_{n-1}\|_{L_{P_X}^2}^2 + 2\gamma^2\|w_\tau(Y_n - \hat{f}_{n-1}(X_n))(K_{X_n} \otimes K_{X_n})\alpha_{n-1}\|_{L_{P_X}^2}^2 + 2\gamma^2\|\Xi_n^\alpha\|_{L_{P_X}^2}^2$$

$$+ 2\gamma\langle\alpha_{n-1}, \Xi_n^\alpha - w_\tau(Y_n - \hat{f}_{n-1}(X_n))(K_{X_n} \otimes K_{X_n})\alpha_{n-1}\rangle_{L_{P_X}^2}.$$

Taking expectations on both sides and utilizing Lemma 1, we have

$$\mathbb{E}(\|\alpha_n\|_{L_{P_X}^2}^2|\mathcal{F}_{n-1})$$

$$\leq \|\alpha_{n-1}\|_{L_{P_X}^2}^2 + 2\gamma^2\mathbb{E}\|(K_{X_n} \otimes K_{X_n})\alpha_{n-1}\|_{L_{P_X}^2}^2 + 2\gamma^2\mathbb{E}\|\Xi_n^\alpha\|_{L_{P_X}^2}^2 - 2\gamma\langle\alpha_{n-1}, \mathcal{T}_{n-1}\alpha_{n-1}\rangle_{L_{P_X}^2}$$

$$\leq \|\alpha_{n-1}\|_{L_{P_X}^2}^2 + 2\gamma^2\langle\alpha_{n-1}, \mathbb{E}(\|K_{X_n}\|^2 K_{X_n} \otimes K_{X_n})\alpha_{n-1}\rangle_{L_{P_X}^2} + 2\gamma^2\mathbb{E}\|\Xi_n^\alpha\|_{L_{P_X}^2}^2 - 2c\gamma\langle\alpha_{n-1}, \Sigma\alpha_{n-1}\rangle_{L_{P_X}^2}$$

$$\leq \|\alpha_{n-1}\|_{L_{P_X}^2}^2 + 2\gamma^2\mathbb{E}\|\Xi_n^\alpha\|_{L_{P_X}^2}^2 - 2c\gamma(1 - \gamma c^{-1}B^2)\langle\alpha_{n-1}, \Sigma\alpha_{n-1}\rangle_{L_{P_X}^2}.$$

Taking another expectation on both sides, we obtain that

$$\mathbb{E}\langle\alpha_{n-1},\Sigma\alpha_{n-1}\rangle_{L^2_{P_X}} \leq \frac{1}{2c\gamma(1-\gamma c^{-1}B^2)}\mathbb{E}\left(\|\alpha_{n-1}\|^2_{L^2_{P_X}} - \|\alpha_n\|^2_{L^2_{P_X}} + 2\gamma^2\|\Xi^\alpha_n\|^2_{L^2_{P_X}}\right).$$

By convexity and $\alpha_0 = 0$, we have

$$\mathbb{E}\langle\bar{\alpha}_{n-1},\Sigma\bar{\alpha}_{n-1}\rangle_{L^2_{P_X}} \leq \frac{1}{c(1-\gamma c^{-1}B^2)}\frac{\gamma}{n}\sum_{k=1}^n\mathbb{E}\|\Xi^\alpha_k\|^2_{L^2_{P_X}}.$$

$\square$

**Lemma 5.** *Consider* $\alpha_n = \left(I - \gamma w_\tau(Y_n - \hat{f}_{n-1}(X_n))K_{X_n}\otimes K_{X_n}\right)\alpha_{n-1}$ *with* $\alpha_0 = -f_{\mathcal{H}}$. *If* $r > 1/2$, *then*

$$\mathbb{E}\langle\bar{\alpha}_n,\Sigma\bar{\alpha}_n\rangle_{L^2_{P_X}} \leq O\left(\left(1+\gamma^{(1+\alpha)(2r-1)/\alpha}n^{(2r-1)/\alpha}\right)(\gamma n)^{-2r}\left\|\Sigma^{-r}f_{\mathcal{H}}\right\|^2_\Sigma\right).$$

*Proof of Lemma 5.* The proof is similar to that of Lemma 13 in Dieuleveut and Bach [2016]. The recursion implies that

$$\alpha_n = M(n, i+1)\alpha_i$$

for any $i = 0, 1, \ldots, n-1$, where $M(n, k) = \prod_{j=k}^n\left(I - \gamma w_\tau(Y_j - \hat{f}_{j-1}(X_j))K_{X_j}\otimes K_{X_j}\right)$. Note that

$$n^2\mathbb{E}\langle\bar{\alpha}_n,\Sigma\bar{\alpha}_n\rangle = \mathbb{E}\sum_{i=0}^n\langle\alpha_i,\Sigma\alpha_i\rangle + 2\mathbb{E}\sum_{i=0}^{n-1}\sum_{j=i+1}^n\langle\alpha_i,\Sigma\alpha_j\rangle.$$

For the second term, by Lemma 1,

$$\mathbb{E}\sum_{i=0}^{n-1}\sum_{j=i+1}^n\langle\alpha_i,\Sigma\alpha_j\rangle \leq \mathbb{E}\sum_{i=0}^{n-1}\sum_{j=i+1}^n\langle\alpha_i,\Sigma(I-c\gamma\Sigma)^{j-i}\alpha_i\rangle$$

$$= \mathbb{E}\sum_{i=0}^{n-1}\langle\alpha_i, \left[c^{-1}\gamma^{-1}\left((I-c\gamma\Sigma)-(I-c\gamma\Sigma)^{n-i+1}\right)\wedge n\Sigma(I-c\gamma\Sigma)\right]\alpha_i\rangle$$

$$\leq \mathbb{E}\sum_{i=0}^n\langle\alpha_i, A_{i,n}\alpha_i\rangle - \mathbb{E}\sum_{i=0}^n\langle\alpha_i,\Sigma\alpha_i\rangle,$$

where $A_{i,n} \preccurlyeq (c^{-1}\gamma^{-1}I \wedge n\Sigma) =: A$. Define the operator $T$ from symmetric matrices to symmetric matrices as

$$TA = c\Sigma A + cA\Sigma - \gamma\mathbb{E}\left[K_{X_n}\otimes K_{X_n}AK_{X_n}\otimes K_{X_n}\right].$$

Then for any symmetric matrix $A$,

$$\mathbb{E}\sum_{i=0}^n(M(i,1))^\top AM(i,1) \leq \sum_{i=0}^n(I-\gamma T)^iA.$$

Similar to the proof of Lemma 14 in Dieuleveut and Bach [2016], for $r > 1/2$, we have

$$\gamma\sum_{i=0}^n(I-\gamma T)^iA \preccurlyeq O\left(\left(\gamma^{-1}+n^{1/\alpha}\gamma^{1/\alpha}\right)^{2r-1}n^{2-2r}\Sigma^{1-2r}\right),$$

and then

$$\mathbb{E}\sum_{i=0}^n\langle\alpha_i, A\alpha_i\rangle \leq O\left(\left(1+\gamma^{1+1/\alpha}n^{1/\alpha}\right)^{2r-1}\gamma^{-2r}n^{2-2r}\|\Sigma^{-r}\alpha_0\|^2_\Sigma\right).$$

Thus,

$$\mathbb{E}\langle\bar{\alpha}_n,\Sigma\bar{\alpha}_n\rangle_{L^2_{P_X}} \leq O\left(\left(1+\gamma^{(1+\alpha)(2r-1)/\alpha}n^{(2r-1)/\alpha}\right)(\gamma n)^{-2r}\left\|\Sigma^{-r}f_{\mathcal{H}}\right\|^2_\Sigma\right).$$

$\square$

**Lemma 6.** *Suppose Assumptions 2, 4–7 hold. Define*

$$Bias(n, \gamma, \Sigma, \{\mathcal{T}_i\}_i, f_{\mathcal{H}}) = \frac{2}{n^2} \mathbb{E} \left\| \Sigma^{1/2} \sum_{j=1}^{n} \prod_{i=1}^{j} (I - \gamma \mathcal{T}_{i-1}) f_{\mathcal{H}} \right\|_{\Sigma}^2.$$

*If $0 \leq r \leq 1$, then*

$$Bias(n, \gamma, \Sigma, \{\mathcal{T}_i\}_i, f_{\mathcal{H}}) \leq 2c^{-2r} (\gamma n)^{-2r} \left\| \Sigma^{-r} f_{\mathcal{H}} \right\|_{\Sigma}^2;$$

*If $r > 1$, then*

$$Bias(n, \gamma, \Sigma, \{\mathcal{T}_i\}_i, f_{\mathcal{H}}) \leq 2c^{-2r} n^{-2} \gamma^{-2r} \left\| \Sigma^{-r} f_{\mathcal{H}} \right\|_{\Sigma}^2.$$

*Proof of Lemma 6.* The proof is similar to that of Lemma 4 in Dieuleveut and Bach [2016]. Utilizing Lemma 1,

$$\begin{aligned}
\text{Bias}(n, \gamma, \Sigma, \{\mathcal{T}_i\}_i, f_{\mathcal{H}}) &\leq \frac{2}{n^2} \left\| \Sigma^{1/2} \sum_{j=1}^{n} \prod_{i=1}^{j} (I - c\gamma\Sigma) f_{\mathcal{H}} \right\|_{\Sigma}^2 \\
&\leq \frac{2}{n^2} \left\| \sum_{j=0}^{n-1} (I - c\gamma\Sigma)^j \Sigma^r \right\|_{\Sigma}^2 \left\| \Sigma^{-r} f_{\mathcal{H}} \right\|_{\Sigma}^2 \\
&= \frac{2}{n^2} c^{-2r} \gamma^{-2r} \left\| \sum_{j=0}^{n-1} (I - c\gamma\Sigma)^j (c\gamma\Sigma)^r \right\|_{\Sigma}^2 \left\| \Sigma^{-r} f_{\mathcal{H}} \right\|_{\Sigma}^2 \\
&\leq \frac{2}{n^2} c^{-2r} \gamma^{-2r} \sup_{0 \leq x \leq 1} \left\{ \sum_{j=0}^{n-1} (1-x)^j x^r \right\}^2 \left\| \Sigma^{-r} f_{\mathcal{H}} \right\|_{\Sigma}^2.
\end{aligned}$$

If $0 \leq r \leq 1$, then $\sup_{0 \leq x \leq 1} \left\{ \sum_{j=0}^{n-1} (1-x)^j x^r \right\} \leq n^{1-r}$, it follows that

$$\text{Bias}(n, \gamma, \Sigma, \{\mathcal{T}_i\}_i, f_{\mathcal{H}}) \leq 2c^{-2r} (\gamma n)^{-2r} \left\| \Sigma^{-r} f_{\mathcal{H}} \right\|_{\Sigma}^2.$$

If $r > 1$, then $\sup_{0 \leq x \leq 1} \left\{ \sum_{j=0}^{n-1} (1-x)^j x^r \right\} = 1$, it follows that

$$\text{Bias}(n, \gamma, \Sigma, \{\mathcal{T}_i\}_i, f_{\mathcal{H}}) \leq 2c^{-2r} n^{-2} \gamma^{-2r} \left\| \Sigma^{-r} f_{\mathcal{H}} \right\|_{\Sigma}^2.$$

$\square$

**Lemma 7.** *Under Assumptions 2, 4–7, we have for $r \geq 0$,*

$$\begin{aligned}
Var(n, \gamma, \Sigma, \{\mathcal{T}_i\}_i, \{\Xi_i^r\}_i) &= \frac{2}{n^2} \mathbb{E} \left\| \Sigma^{1/2} \sum_{j=1}^{n} \sum_{k=1}^{j} \prod_{i=k+1}^{j} (I - \gamma \mathcal{T}_{i-1}) \gamma \Xi_k^r \right\|_{\Sigma}^2 \\
&\leq c^{-r} \gamma^r B^{2r} 2c^{-2} \tilde{\sigma}^2 \left[ c_0(\alpha) (c\gamma)^{1/\alpha} n^{-1+1/\alpha} + n^{-1} \right],
\end{aligned}$$

*where $c_0(\alpha) = \frac{4\alpha^2}{(\alpha+1)(2\alpha-1)}$.*

*Proof of Lemma 7.* Utilizing Lemmas 1 and 3, we have

$$\text{Var}(n, \gamma, \Sigma, \{\mathcal{T}_i\}_i, \{\Xi_i^r\}_i) \leq \frac{2}{n^2} \mathbb{E} \left\| \Sigma^{1/2} \sum_{j=1}^n \sum_{k=1}^j (I - c\gamma\Sigma)^{j-k} \gamma \Xi_k^r \right\|_\Sigma^2$$

$$= \frac{2}{n^2} \sum_{k=1}^n \gamma^2 \mathbb{E} \left[ \text{tr} \left( \left( \sum_{j=k}^n (I - c\gamma\Sigma)^{j-k} \right) \Sigma \left( \sum_{j=k}^n (I - c\gamma\Sigma)^{j-k} \right) \Xi_k^r \otimes \Xi_k^r \right) \right]$$

$$\leq c^{-r}\gamma^r B^{2r}\widetilde{\sigma}^2 \frac{2}{n^2} \sum_{k=1}^n \gamma^2 \text{tr} \left[ \left( \sum_{j=k}^n (I - c\gamma\Sigma)^{j-k} \right) \Sigma \right]^2$$

$$= c^{-r}\gamma^r B^{2r}\widetilde{\sigma}^2 \frac{2c^{-2}}{n^2} \sum_{k=1}^n \text{tr} \left[ I - (I - c\gamma\Sigma)^{n-k+1} \right]^2.$$

Note that

$$\text{tr} \left[ I - (I - c\gamma\Sigma)^j \right]^2 \leq 1 + \int_1^\infty \left[ 1 - (1 - c\gamma u^{-\alpha})^j \right]^2 du$$

$$= 1 + \int_1^{(c\gamma j)^{1/\alpha}} \left[ 1 - (1 - c\gamma u^{-\alpha})^j \right]^2 du + \int_{(c\gamma)^{1/\alpha}}^\infty \left[ 1 - (1 - c\gamma u^{-\alpha})^j \right]^2 du$$

$$\leq 1 + \left( 1 + \frac{1}{2\alpha - 1} \right) (c\gamma j)^{1/\alpha},$$

then

$$\text{Var}(n, \gamma, \Sigma, \{\mathcal{T}_i\}_i, \{\Xi_i^r\}_i) \leq \frac{2}{n^2} \mathbb{E} \left\| \Sigma^{1/2} \sum_{j=1}^n \sum_{k=1}^j (I - c\gamma\Sigma)^{j-k} \gamma \Xi_k^r \right\|_\Sigma^2$$

$$\leq c^{-r}\gamma^r B^{2r}\widetilde{\sigma}^2 \frac{2c^{-2}}{n^2} \sum_{k=1}^n \left[ 1 + \left( 1 + \frac{1}{2\alpha - 1} \right) (c\gamma(n-k+1))^{1/\alpha} \right]$$

$$\leq c^{-r}\gamma^r B^{2r} 2c^{-2}\widetilde{\sigma}^2 \left[ \frac{4\alpha^2 (c\gamma)^{1/\alpha}}{(\alpha+1)(2\alpha-1)} n^{-1+1/\alpha} + n^{-1} \right].$$

$\square$

Following Lemmas 1–7, we give the proof of Theorem 2.

*Proof of Theorem 2.* The recursion (7) is equivalent to

$$\hat{f}_n = \left( I - \gamma w_\tau(Y_n - \hat{f}_{n-1}(X_n))K_{X_n} \otimes K_{X_n} \right) \hat{f}_{n-1} + \gamma Y_n w_\tau(Y_n - \hat{f}_{n-1}(X_n))K_{X_n} + \gamma \xi_n.$$

Denote $\widetilde{K_{X_n} \otimes K_{X_n}}$ as the a.s. extension of $K_{X_n} \otimes K_{X_n} : \mathcal{H} \to \mathcal{H}$ to $L_{P_X}^2 \to \mathcal{H}$, such that $\widetilde{K_{X_n} \otimes K_{X_n}}(f) = f(X_n)K_{X_n}$, and it will be denoted as $K_{X_n} \otimes K_{X_n}$ for simplicity without confusion. Let $\eta_n = \hat{f}_n - f_\mathcal{H}$, and $\Xi_n = (Y_n - f_\mathcal{H}(X_n))w_\tau(Y_n - \hat{f}_{n-1}(X_n))K_{X_n} + \xi_n$. We obtain that $\eta_0 = -f_\mathcal{H}$, and

$$\eta_n = \left( I - \gamma w_\tau(Y_n - \hat{f}_{n-1}(X_n))K_{X_n} \otimes K_{X_n} \right) \eta_{n-1} + \gamma \Xi_n. \qquad (9)$$

We decompose the recursion formula (9) into two simpler recursions $\eta_n^{\text{init}}$ and $\eta_n^{\text{noise}}$ such that $\eta_n = \eta_n^{\text{init}} + \eta_n^{\text{noise}}$. Specifically, the initial component $\{\eta_n^{\text{init}}\}_n$ is defined as $\eta_0^{\text{init}} = -f_\mathcal{H}$, and

$$\eta_n^{\text{init}} = \left( I - \gamma w_\tau(Y_n - \hat{f}_{n-1}(X_n))K_{X_n} \otimes K_{X_n} \right) \eta_{n-1}^{\text{init}};$$

and the noise component $\{\eta_n^{\text{noise}}\}_n$ satisfies $\eta_0^{\text{noise}} = 0$, and

$$\eta_n^{\text{noise}} = \left( I - \gamma w_\tau(Y_n - \hat{f}_{n-1}(X_n))K_{X_n} \otimes K_{X_n} \right) \eta_{n-1}^{\text{noise}} + \gamma \Xi_n.$$

By Minkowski's inequality,

$$\left(\mathbb{E}\|\bar{\eta}_n\|_{L_{P_X}^2}^2\right)^{1/2} \le \left(\mathbb{E}\|\bar{\eta}_n^{\text{init}}\|_{L_{P_X}^2}^2\right)^{1/2} + \left(\mathbb{E}\|\bar{\eta}_n^{\text{noise}}\|_{L_{P_X}^2}^2\right)^{1/2}, \tag{10}$$

where $\bar{\eta}_n = \sum_{j=1}^n \eta_j/n$, $\bar{\eta}_n^{\text{init}} = \sum_{j=1}^n \eta_j^{\text{init}}/n$, and $\bar{\eta}_n^{\text{noise}} = \sum_{j=1}^n \eta_j^{\text{noise}}/n$. Next, we will respectively present the upper bounds of $\mathbb{E}\|\eta_n^{\text{init}}\|_{L_{P_X}^2}^2$ and $\mathbb{E}\|\eta_n^{\text{noise}}\|_{L_{P_X}^2}^2$.

**Noise component.** Denote $\mathcal{T}_{n-1} = \mathbb{E}\left[w_\tau(Y_n - \hat{f}_{n-1}(X_n))K_{X_n} \otimes K_{X_n}|\mathcal{F}_{n-1}\right]$. Define the main recursion of $\eta_n^{\text{noise}}$ as

$$\eta_n^{\text{noise},0} = (I - \gamma\mathcal{T}_{n-1})\,\eta_{n-1}^{\text{noise},0} + \gamma\Xi_n, \text{ with } \eta_0^{\text{noise},0} = 0.$$

Then the residual term is

$$\eta_n^{\text{noise}} - \eta_n^{\text{noise},0} = \left(I - \gamma w_\tau(Y_n - \hat{f}_{n-1}(X_n))K_{X_n} \otimes K_{X_n}\right)\left(\eta_{n-1}^{\text{noise}} - \eta_{n-1}^{\text{noise},0}\right) + \gamma\Xi_n^1, \text{ with } \eta_0^{\text{noise}} - \eta_0^{\text{noise},0} = 0,$$

where $\Xi_n^1 = \left(\mathcal{T}_{n-1} - w_\tau(Y_n - \hat{f}_{n-1}(X_n))K_{X_n} \otimes K_{X_n}\right)\eta_{n-1}^{\text{noise},0}$. For any $r \ge 0$, we further define a sequence $\{\eta_n^{\text{noise},r}\}_n$ as follows:

$$\eta_n^{\text{noise},r} = (I - \gamma\mathcal{T}_{n-1})\,\eta_{n-1}^{\text{noise},r} + \gamma\Xi_n^r, \text{ with } \eta_0^{\text{noise},r} = 0,$$

where $\Xi_n^0 = \Xi_n$, and $\Xi_n^r = \left(\mathcal{T}_{n-1} - w_\tau(Y_n - \hat{f}_{n-1}(X_n))K_{X_n} \otimes K_{X_n}\right)\eta_{n-1}^{\text{noise},r-1}$ for $r \ge 1$. Then $\eta_0^{\text{noise}} - \sum_{i=0}^r \eta_0^{\text{noise},i} = 0$, and

$$\eta_n^{\text{noise}} - \sum_{i=0}^r \eta_n^{\text{noise},i} = \left(I - \gamma w_\tau(Y_n - \hat{f}_{n-1}(X_n))K_{X_n} \otimes K_{X_n}\right)\left(\eta_{n-1}^{\text{noise}} - \sum_{i=0}^r \eta_{n-1}^{\text{noise},i}\right) + \gamma\Xi_n^{r+1}.$$

Minkowski's inequality implies that

$$\left(\mathbb{E}\|\bar{\eta}_n^{\text{noise}}\|_{L_{P_X}^2}^2\right)^{1/2} \le \sum_{i=0}^r \left(\mathbb{E}\|\bar{\eta}_n^{\text{noise},i}\|_{L_{P_X}^2}^2\right)^{1/2} + \left(\mathbb{E}\left\|\bar{\eta}_n^{\text{noise}} - \sum_{i=0}^r \bar{\eta}_n^{\text{noise},i}\right\|_{L_{P_X}^2}^2\right)^{1/2},$$

where $\bar{\eta}_n^{\text{noise},i} = \sum_{j=1}^n \eta_j^{\text{noise},i}/n$. To obtain the upper bound of $\mathbb{E}\|\bar{\eta}_n^{\text{noise},i}\|_{L_{P_X}^2}^2$, we first verify the conditions in Lemma 2. The definition of $w_\tau(\cdot)$ implies that $\mathcal{T}_{n-1} \preccurlyeq \Sigma$ for any $n$. For $i = 0$,

$$\mathbb{E}(\|\Xi_n^0\|^2|\mathcal{F}_{n-1}) = \mathbb{E}\left[\left\|(Y_n - f_{\mathcal{H}}(X_n))w_\tau(Y_n - \hat{f}_{n-1}(X_n))K_{X_n} + \xi_n\right\|^2\Big|\mathcal{F}_{n-1}\right]$$

$$\le \mathbb{E}\left[\|(Y_n - f_{\mathcal{H}}(X_n))K_{X_n}\|^2\Big|\mathcal{F}_{n-1}\right] + \mathbb{E}(\|\xi_n\|^2) < \infty.$$

For $i \ge 1$,

$$\mathbb{E}(\|\Xi_n^i\|^2|\mathcal{F}_{n-1}) = \mathbb{E}\left[\left\|\left(\mathcal{T}_{n-1} - w_\tau(Y_n - \hat{f}_{n-1}(X_n))K_{X_n} \otimes K_{X_n}\right)\eta_{n-1}^{\text{noise},r-1}\right\|^2\Big|\mathcal{F}_{n-1}\right]$$

$$= \left[\mathbb{E}\left(\left\|w_\tau(Y_n - \hat{f}_{n-1}(X_n))K_{X_n} \otimes K_{X_n}\right\|^2\Big|\mathcal{F}_{n-1}\right) - \|\mathcal{T}_{n-1}\|^2\right]\left\|\eta_{n-1}^{\text{noise},r-1}\right\|^2$$

$$\le \mathbb{E}\left(\|K_{X_n} \otimes K_{X_n}\|^2\right)\left\|\eta_{n-1}^{\text{noise},r-1}\right\|^2 < \infty.$$

By Lemma 3, for any $r \ge 0$ and any $n \ge 0$, $\mathbb{E}(\Xi_n^r \otimes \Xi_n^r) \preccurlyeq c^{-r}\gamma^r B^{2r}\widetilde{\sigma}^2\Sigma$. Utilizing Lemma 2 with $\alpha_0 = 0$ and $\alpha_n = \eta_n^{\text{noise},i}$, we have

$$\mathbb{E}\|\bar{\eta}_n^{\text{noise},i}\|_{L_{P_X}^2}^2 \le \text{Var}(n, \gamma, \Sigma, \{\mathcal{T}_j\}_j, \{\Xi_j^i\}_j). \tag{11}$$

For the residual term $\eta_n^{\text{noise}} - \sum_{i=0}^r \eta_n^{\text{noise},i}$, it is easy to verify the conditions in Lemma 4. If $\gamma c^{-1}B^2 < 1$, then

$$\mathbb{E}\left\langle \bar{\eta}_n^{\text{noise}} - \sum_{i=0}^r \bar{\eta}_n^{\text{noise},i}, \Sigma\left(\bar{\eta}_n^{\text{noise}} - \sum_{i=0}^r \bar{\eta}_n^{\text{noise},i}\right)\right\rangle_{L_{P_X}^2} \le \frac{1}{c(1 - \gamma c^{-1}B^2)}\frac{\gamma}{n}\sum_{k=1}^n \mathbb{E}\|\Xi_k^{r+1}\|_{L_{P_X}^2}^2.$$

Utilizing Lemma 3,

$$\sum_{k=1}^{n}\mathbb{E}\|\Xi_k^{r+1}\|_{L_{P_X}^2}^2 \leq \sum_{k=1}^{n}\mathrm{tr}\left(\mathbb{E}(\Xi_k^{r+1}\otimes\Xi_k^{r+1})\right) \leq nc^{-(r+1)}\gamma^{r+1}B^{2(r+1)}\widetilde{\sigma}^2\mathrm{tr}(\Sigma).$$

Then,

$$\mathbb{E}\left\|\bar{\eta}_n^{\mathrm{noise}} - \sum_{i=0}^{r}\bar{\eta}_n^{\mathrm{noise},i}\right\|_{L_{P_X}^2}^2 \leq \frac{1}{1-\gamma c^{-1}B^2}c^{-(r+2)}\gamma^{r+2}B^{2r+2}\widetilde{\sigma}^2\mathrm{tr}(\Sigma). \tag{12}$$

Combining (11), (12), and Lemma 7, we have

$$\left(\mathbb{E}\|\bar{\eta}_n^{\mathrm{noise}}\|_{L_{P_X}^2}^2\right)^{1/2}$$

$$\leq \sum_{i=0}^{r}\left[\mathrm{Var}(n,\gamma,\Sigma,\{\mathcal{T}_j\}_j,\{\Xi_j^i\}_j)\right]^{1/2} + \left[\frac{1}{1-\gamma c^{-1}B^2}c^{-(r+2)}\gamma^{r+2}B^{2r+2}\widetilde{\sigma}^2\mathrm{tr}(\Sigma)\right]^{1/2}$$

$$\leq \left[2c^{-2}\widetilde{\sigma}^2\left(c_0(\alpha)(c\gamma)^{1/\alpha}n^{-1+1/\alpha}+n^{-1}\right)\right]^{1/2}\sum_{i=0}^{r}(c^{-1}\gamma B^2)^{i/2} + \left[\frac{1}{1-c^{-1}\gamma B^2}c^{-(r+2)}\gamma^{r+2}B^{2r+2}\widetilde{\sigma}^2\mathrm{tr}(\Sigma)\right]^{1/2}$$

$$\leq \frac{\sqrt{2}c^{-1}\widetilde{\sigma}}{1-(c^{-1}\gamma B^2)^{1/2}}\left[\left(c_0(\alpha)(c\gamma)^{1/\alpha}\right)^{1/2}n^{-1/2+1/(2\alpha)}+n^{-1/2}\right] + \left[\frac{1}{1-c^{-1}\gamma B^2}c^{-(r+2)}\gamma^{r+2}B^{2r+2}\widetilde{\sigma}^2\mathrm{tr}(\Sigma)\right]^{1/2},$$

where $c_0(\alpha) = \frac{4\alpha^2}{(\alpha+1)(2\alpha-1)}$. Let the recursion step $r \to \infty$, if $c^{-1}\gamma B^2 < 1$, then we have

$$\left(\mathbb{E}\|\bar{\eta}_n^{\mathrm{noise}}\|_{L_{P_X}^2}^2\right)^{1/2} \leq \frac{\sqrt{2}c^{-1}\widetilde{\sigma}}{1-(c^{-1}\gamma B^2)^{1/2}}\left[\left(c_0(\alpha)(c\gamma)^{1/\alpha}\right)^{1/2}n^{-1/2+1/(2\alpha)}+n^{-1/2}\right]. \tag{13}$$

**Initial component.** Recall that $\eta_n^{\mathrm{init}} = \left(I - \gamma w_\tau(Y_n - \hat{f}_{n-1}(X_n))K_{X_n}\otimes K_{X_n}\right)\eta_{n-1}^{\mathrm{init}}$ with $\eta_0^{\mathrm{init}} = -f_{\mathcal{H}}$. Define the main recursion as

$$\eta_n^{\mathrm{init},0} = (I - \gamma\mathcal{T}_{n-1})\eta_{n-1}^{\mathrm{init},0}$$

with $\eta_0^{\mathrm{init},0} = -f_{\mathcal{H}}$. Then the residual term is

$$\eta_n^{\mathrm{init}} - \eta_n^{\mathrm{init},0} = \left(I - \gamma w_\tau(Y_n - \hat{f}_{n-1}(X_n))K_{X_n}\otimes K_{X_n}\right)\left(\eta_{n-1}^{\mathrm{init}} - \eta_{n-1}^{\mathrm{init},0}\right) + \gamma\Xi_n^{\mathrm{init}}$$

with $\eta_0^{\mathrm{init}} - \eta_0^{\mathrm{init},0} = 0$, where $\Xi_n^{\mathrm{init}} = \left(\mathcal{T}_{n-1} - w_\tau(Y_n - \hat{f}_{n-1}(X_n))K_{X_n}\otimes K_{X_n}\right)\eta_{n-1}^{\mathrm{init},0}$. Utilizing Lemmas 2 and 4, we have

$$\mathbb{E}\langle\bar{\eta}_n^{\mathrm{init}},\Sigma\bar{\eta}_n^{\mathrm{init}}\rangle_{L_{P_X}^2} \leq \mathrm{Bias}(n,\gamma,\Sigma,\{\mathcal{T}_i\}_i,f_{\mathcal{H}}),$$

and

$$\mathbb{E}\langle\bar{\eta}_n^{\mathrm{init}} - \bar{\eta}_n^{\mathrm{init},0},\Sigma\left(\bar{\eta}_n^{\mathrm{init}} - \bar{\eta}_n^{\mathrm{init},0}\right)\rangle_{L_{P_X}^2} \leq \frac{1}{c(1-\gamma c^{-1}B^2)}\frac{\gamma}{n}\sum_{k=1}^{n}\mathbb{E}\|\Xi_k^{\mathrm{init}}\|_{L_{P_X}^2}^2,$$

where

$$\mathrm{Bias}(n,\gamma,\Sigma,\{\mathcal{T}_i\}_i,f_{\mathcal{H}}) = \frac{2}{n^2}\mathbb{E}\left\|\Sigma^{1/2}\sum_{j=1}^{n}\prod_{i=1}^{j}(I - \gamma\mathcal{T}_{i-1})f_{\mathcal{H}}\right\|_\Sigma^2.$$

Note that

$$\mathbb{E}\|\Xi_k^{\mathrm{init}}\|_{L_{P_X}^2}^2$$

$$= \mathbb{E}\left\langle f_{\mathcal{H}},\prod_{i=1}^{k}(I - \gamma\mathcal{T}_{i-1})\left(\mathcal{T}_{k-1} - w_\tau(Y_k - \hat{f}_{k-1}(X_k))K_{X_k}\otimes K_{X_k}\right)^2\prod_{i=1}^{k}(I - \gamma\mathcal{T}_{i-1})f_{\mathcal{H}}\right\rangle_{L_{P_X}^2}$$

$$\leq B^2\mathbb{E}\left\langle f_{\mathcal{H}},\left(\prod_{i=1}^{k}(I - \gamma\mathcal{T}_{i-1})\right)^2\Sigma f_{\mathcal{H}}\right\rangle_{L_{P_X}^2}.$$

By Lemma 1, for any $r$, if $c^{-1}\gamma B^2 < 1$ with $c \in (0, 1]$, then $c\gamma B^2 < 1$, it follows that

$$
\begin{aligned}
\frac{\gamma}{n} \sum_{k=1}^n \mathbb{E}\|\Xi_k^{\text{init}}\|_{L^2_{P_X}}^2 &\leq \frac{\gamma B^2}{n} \sum_{k=1}^n \mathbb{E} \left\langle f_{\mathcal{H}}, \left(\prod_{i=1}^k (I - \gamma \mathcal{T}_{i-1})\right)^2 \Sigma f_{\mathcal{H}} \right\rangle_{L^2_{P_X}} \\
&\leq \frac{\gamma B^2}{n} \mathbb{E} \left\langle f_{\mathcal{H}}, \sum_{k=1}^n (I - c\gamma\Sigma)^{2k} \Sigma f_{\mathcal{H}} \right\rangle_{L^2_{P_X}} \\
&\leq \frac{\gamma B^2}{n} \left\| \left[\sum_{k=1}^n (I - c\gamma\Sigma)^{2k} \Sigma^{2r}\right]^{1/2} \Sigma^{-r} f_{\mathcal{H}} \right\|_{L^2_{P_X}}^2 \\
&\leq \frac{\gamma B^2}{n} c^{-2r} \gamma^{-2r} \left\| \sum_{k=1}^n (I - c\gamma\Sigma)^{2k} (c\gamma\Sigma)^{2r} \right\|_{L^2_{P_X}} \left\|\Sigma^{-r} f_{\mathcal{H}}\right\|_{L^2_{P_X}}^2 \\
&\leq \frac{\gamma B^2}{n} c^{-2r} \gamma^{-2r} \sup_{0 \leq x \leq 1} \left\{ \sum_{k=1}^n (1 - x)^{2k} x^{2r} \right\} \left\|\Sigma^{-r} f_{\mathcal{H}}\right\|_{L^2_{P_X}}^2 .
\end{aligned}
$$

If $r \leq 1/2$, then $\sup_{0 \leq x \leq 1} \left\{ \sum_{k=1}^n (1-x)^{2k} x^{2r} \right\} \leq n^{1-2r}$, whose proof can be found in Dieuleveut and Bach [2016]. Then

$$
\mathbb{E}\|\bar{\eta}_n^{\text{init}} - \bar{\eta}_n^{\text{init},0}\|_{L^2_{P_X}}^2 \leq O\left( (\gamma n)^{-2r} \left\|\Sigma^{-r} f_{\mathcal{H}}\right\|_{L^2_{P_X}}^2 \right).
$$

Thus, for $r \leq 1/2$,

$$
\left(\mathbb{E}\|\bar{\eta}_n^{\text{init}}\|_{L^2_{P_X}}^2\right)^{1/2} \leq \text{Bias}(n, \gamma, \Sigma, \{\mathcal{T}_i\}_i, f_{\mathcal{H}})^{1/2} + O\left( (\gamma n)^{-r} \left\|\Sigma^{-r} f_{\mathcal{H}}\right\|_{L^2_{P_X}} \right).
$$

By Lemma 6, if $0 \leq r \leq 1/2$, then

$$
\left(\mathbb{E}\|\bar{\eta}_n^{\text{init}}\|_{L^2_{P_X}}^2\right)^{1/2} \leq O\left( (\gamma n)^{-r} \left\|\Sigma^{-r} f_{\mathcal{H}}\right\|_{L^2_{P_X}} \right). \tag{14}
$$

When $r > 1/2$, utilizing Lemmas 5 and 6, we have

$$
\left(\mathbb{E}\|\bar{\eta}_n^{\text{init}}\|_{L^2_{P_X}}^2\right)^{1/2} \leq O\left( \left(1 + \gamma^{(1+\alpha)(2r-1)/\alpha} n^{(2r-1)/\alpha}\right)^{1/2} \gamma^{-r} n^{-\min\{r,1\}} \left\|\Sigma^{-r} f_{\mathcal{H}}\right\|_{L^2_{P_X}} \right). \tag{15}
$$

To put (14) and (15) together, denote $q(\gamma, n) = 0$ for $r \leq 1/2$, and $q(\gamma, n) = \gamma^{(1+\alpha)(2r-1)/\alpha} n^{(2r-1)/\alpha}$ for $r > 1/2$. Then

$$
\left(\mathbb{E}\|\bar{\eta}_n^{\text{init}}\|_{L^2_{P_X}}^2\right)^{1/2} \leq O\left( (1 + q(\gamma, n))^{1/2} \gamma^{-r} n^{-\min\{r,1\}} \left\|\Sigma^{-r} f_{\mathcal{H}}\right\|_{L^2_{P_X}} \right). \tag{16}
$$

Combining (10), (13), and (16), we obtain that

$$
\begin{aligned}
\left(\mathbb{E}\|\bar{\eta}_n\|_{L^2_{P_X}}^2\right)^{1/2} &\leq \frac{\sqrt{2} c^{-1} \widetilde{\sigma}}{1 - (c^{-1}\gamma B^2)^{1/2}} \left[ \left(c_0(\alpha)(c\gamma)^{1/\alpha}\right)^{1/2} n^{-1/2+1/(2\alpha)} + n^{-1/2} \right] \\
&\quad + O\left( (1 + q(\gamma, n))^{1/2} \gamma^{-r} n^{-\min\{r,1\}} \left\|\Sigma^{-r} f_{\mathcal{H}}\right\|_{L^2_{P_X}} \right).
\end{aligned}
$$

Thus,

$$
\mathbb{E}\|\bar{f}_n - f_{\mathcal{H}}\|_{L^2_{P_X}}^2 \leq O\left( \widetilde{\sigma}^2 \left(\gamma^{1/\alpha} n^{-1+1/\alpha} + n^{-1}\right) + (1 + q(\gamma, n)) \gamma^{-2r} n^{-2\min\{r,1\}} \left\|\Sigma^{-r} f_{\mathcal{H}}\right\|_{L^2_{P_X}}^2 \right)
$$

$$\square$$

## J.5 Proof of Theorem 3

The proof idea of Theorem 3 is similar to that of Theorem 2. We begin by presenting several useful lemmas that will be useful in proving the main theorem. For brevity, we omit proofs that are analogous to previously shown results. The proof of Theorem 3 is given thereafter.

**Lemma 8.** *Let $\alpha_n = (I - \gamma_n \mathcal{T}_{n-1}) \alpha_{n-1} + \gamma_n \Xi_n^\alpha$. $\mathcal{T}_{n-1}$ satisfies $\mathcal{T}_{n-1} \preccurlyeq \Sigma$ with $\gamma_n \Sigma \preccurlyeq I$, where $\preccurlyeq$ denotes the order between self-adjoint operators. $\Xi_n^\alpha \in \mathcal{H}$ is $\mathcal{F}_n$ measurable for a sequence of increasing $\sigma$-fields $\{\mathcal{F}_n\}_n$, $\mathbb{E}(\|\Xi_n^\alpha\|^2|\mathcal{F}_{n-1})$ is finite, and $\mathbb{E}(\Xi_n^\alpha \otimes \Xi_n^\alpha) \preccurlyeq \sigma_\alpha^2 \Sigma$. Then*

$$\mathbb{E}\langle \bar{\alpha}_n, \Sigma \bar{\alpha}_n \rangle_{L_{P_X}^2} \leq Bias(n, \{\gamma_i\}_i, \Sigma, \{\mathcal{T}_i\}_i, \alpha_0) + Var(n, \{\gamma_i\}_i, \Sigma, \{\mathcal{T}_i\}_i, \{\Xi_i^\alpha\}_i),$$

*where $\bar{\alpha}_n = \sum_{j=1}^n \alpha_n$,*

$$Bias(n, \{\gamma_i\}_i, \Sigma, \{\mathcal{T}_i\}_i, \alpha_0) = \frac{2}{n^2} \mathbb{E} \left\| \Sigma^{1/2} \sum_{j=1}^n \prod_{i=1}^j (I - \gamma_i \mathcal{T}_{i-1}) \alpha_0 \right\|_\Sigma^2,$$

*and*

$$Var(n, \{\gamma_i\}_i, \Sigma, \{\mathcal{T}_i\}_i, \{\Xi_i^\alpha\}_i) = \frac{2}{n^2} \mathbb{E} \left\| \Sigma^{1/2} \sum_{j=1}^n \sum_{k=1}^j \prod_{i=k+1}^j (I - \gamma_i \mathcal{T}_{i-1}) \gamma_k \Xi_k^\alpha \right\|_\Sigma^2.$$

**Lemma 9.** *Under Assumptions 2–5, for any $r \geq 0$ and any $n \geq 0$, $\mathbb{E}(\Xi_n^r \otimes \Xi_n^r) \preccurlyeq c^{-r} \gamma_0^r B^{2r} \widetilde{\sigma}^2 \Sigma$, and $\mathbb{E}\left(\eta_n^{noise,r} \otimes \eta_n^{noise,r}\right) \preccurlyeq c^{-(r+1)} \gamma_0^{r+1} B^{2r} \widetilde{\sigma}^2 I$, where $\widetilde{\sigma}^2 = \sigma^2 + \frac{8\tau^2 B^2 \log(2/\delta)}{\varepsilon^2}$, and $c$ is defined as in Lemma 1.*

**Lemma 10.** *Assume that $(X_n, \hat{f}_n, \Xi_n^\alpha) \in \mathcal{H} \times \mathcal{H} \times \mathcal{H}$ is $\mathcal{F}_n$ measurable for a sequence of increasing $\sigma$-fields $\{\mathcal{F}_n\}_n$. Further, $\mathbb{E}(\Xi_n^\alpha|\mathcal{F}_{n-1}) = 0$, $\mathbb{E}(\|\Xi_n^\alpha\|^2|\mathcal{F}_{n-1}) < \infty$, and $\mathbb{E}(\|K_{X_n}\|^2 K_{X_n} \otimes K_{X_n}|\mathcal{F}_{n-1}) \preccurlyeq B^2 \Sigma$ with $\mathbb{E}(K_{X_n} \otimes K_{X_n}|\mathcal{F}_{n-1}) = \Sigma$ for all $n \geq 1$, some constant $B > 0$ and invertible operator $\Sigma$. Let $\alpha_n = \left(I - \gamma_n w_\tau(Y_n - \hat{f}_{n-1}(X_n)) K_{X_n} \otimes K_{X_n}\right) \alpha_{n-1} + \gamma_n \Xi_n^\alpha$, with $\alpha_0 = 0$ and non-increasing $\{\gamma_n\}_n$ satisfying $c^{-1} \gamma_0 B^2 < 1$. Then*

$$\mathbb{E}\langle \bar{\alpha}_{n-1}, \Sigma \bar{\alpha}_{n-1} \rangle_{L_{P_X}^2}$$

$$\leq \frac{1}{2cn(1 - c^{-1}\gamma_0 B^2)} \left[ \sum_{k=1}^{n-1} \mathbb{E}\|\alpha_k\|_{L_{P_X}^2}^2 \left( -\frac{1}{\gamma_k} + \frac{1}{\gamma_{k+1}} \right) + 2 \sum_{k=1}^n \gamma_k \mathbb{E}\|\Xi_k^\alpha\|_{L_{P_X}^2}^2 \right].$$

*Proof of Lemma 10.* Similar to the proof of Lemma 4, we obtain

$$2c\gamma_n(1 - c^{-1}\gamma_n B^2) \mathbb{E}\langle \alpha_{n-1}, \Sigma \alpha_{n-1} \rangle_{L_{P_X}^2} \leq \mathbb{E}\|\alpha_{n-1}\|_{L_{P_X}^2}^2 - \mathbb{E}\|\alpha_n\|_{L_{P_X}^2}^2 + 2\gamma_n^2 \mathbb{E}\|\Xi_n^\alpha\|_{L_{P_X}^2}^2.$$

If $\{\gamma_n\}_n$ is non-increasing, then

$$\mathbb{E}\langle \alpha_{n-1}, \Sigma \alpha_{n-1} \rangle_{L_{P_X}^2} \leq \frac{1}{2c\gamma_n(1 - c^{-1}\gamma_0 B^2)} \left( \mathbb{E}\|\alpha_{n-1}\|_{L_{P_X}^2}^2 - \mathbb{E}\|\alpha_n\|_{L_{P_X}^2}^2 + 2\gamma_n^2 \mathbb{E}\|\Xi_n^\alpha\|_{L_{P_X}^2}^2 \right).$$

By convexity and $\alpha_0 = 0$, we have

$$\mathbb{E}\langle \bar{\alpha}_{n-1}, \Sigma \bar{\alpha}_{n-1} \rangle_{L_{P_X}^2}$$

$$\leq \frac{1}{2cn(1 - c^{-1}\gamma_0 B^2)} \left[ \sum_{k=1}^{n-1} \mathbb{E}\|\alpha_k\|_{L_{P_X}^2}^2 \left( -\frac{1}{\gamma_k} + \frac{1}{\gamma_{k+1}} \right) + 2 \sum_{k=1}^n \gamma_k \mathbb{E}\|\Xi_k^\alpha\|_{L_{P_X}^2}^2 \right].$$

$\square$

**Lemma 11.** *Suppose Assumptions 2, 4–7 hold. Take $\gamma_i \asymp i^{-\zeta}$ with $\zeta \in (0, 1)$. Define*

$$Bias(n, \{\gamma_i\}_i, \Sigma, \{\mathcal{T}_i\}_i, f_{\mathcal{H}}) = \frac{2}{n^2} \mathbb{E} \left\| \Sigma^{1/2} \sum_{j=1}^n \prod_{i=1}^j (I - \gamma_i \mathcal{T}_{i-1}) f_{\mathcal{H}} \right\|_\Sigma^2.$$

*If $r - 1/(1-\zeta) \leq 0$, then*

$$Bias(n, \{\gamma_i\}_i, \Sigma, \{\mathcal{T}_i\}_i, f_{\mathcal{H}}) \leq O\left(n^{-2r(1-\zeta)} \left\|\Sigma^{-r} f_{\mathcal{H}}\right\|_{\Sigma}^2\right);$$

*if $r - 1/(1-\zeta) > 0$, then*

$$Bias(n, \{\gamma_i\}_i, \Sigma, \{\mathcal{T}_i\}_i, f_{\mathcal{H}}) \leq O\left(n^{-2} \left\|\Sigma^{-r} f_{\mathcal{H}}\right\|_{\Sigma}^2\right).$$

*Proof of Lemma 11.* The proof is similar to that of Lemma 6 in Dieuleveut and Bach [2016]. Utilizing Lemma 1, if $\gamma_i = i^{-\zeta}$, then

$$
\text{Bias}(n, \{\gamma_i\}_i, \Sigma, \{\mathcal{T}_i\}_i, f_{\mathcal{H}}) \leq \frac{2}{n^2} \left\| \Sigma^{1/2} \sum_{j=1}^{n} \prod_{i=1}^{j} (I - c\gamma_i\Sigma) f_{\mathcal{H}} \right\|_{\Sigma}^2
$$

$$
\leq \frac{2}{n^2} \left\| \sum_{j=1}^{n} \prod_{i=1}^{j} (I - c\gamma_i\Sigma) \Sigma^r \right\|_{\Sigma}^2 \left\| \Sigma^{-r} f_{\mathcal{H}} \right\|_{\Sigma}^2
$$

$$
\leq \frac{2}{n^2} \sup_{0 \leq x \leq 1/(c\gamma_0)} \left\{ \sum_{j=1}^{n} \prod_{i=1}^{j} (1 - c\gamma_i x) x^r \right\}^2 \left\| \Sigma^{-r} f_{\mathcal{H}} \right\|_{\Sigma}^2
$$

$$
\leq \frac{2}{n^2} \left( C \sup_{0 \leq x \leq 1} \left\{ nx^r \wedge x^{r-1/(1-\zeta)} \right\} \right)^2 \left\| \Sigma^{-r} f_{\mathcal{H}} \right\|_{\Sigma}^2.
$$

If $r - 1/(1-\zeta) \leq 0$, then $\sup_{0 \leq x \leq 1} \left\{ nx^r \wedge x^{r-1/(1-\zeta)} \right\} \leq n^{1-r(1-\zeta)}$, so that

$$\text{Bias}(n, \{\gamma_i\}_i, \Sigma, \{\mathcal{T}_i\}_i, f_{\mathcal{H}}) \leq C_0 n^{-2r(1-\zeta)} \left\| \Sigma^{-r} f_{\mathcal{H}} \right\|_{\Sigma}^2;$$

if $r - 1/(1-\zeta) > 0$, then $\sup_{0 \leq x \leq 1} \left\{ nx^r \wedge x^{r-1/(1-\zeta)} \right\} = 1$, so that

$$\text{Bias}(n, \{\gamma_i\}_i, \Sigma, \{\mathcal{T}_i\}_i, f_{\mathcal{H}}) \leq C_0 n^{-2} \left\| \Sigma^{-r} f_{\mathcal{H}} \right\|_{\Sigma}^2.$$

$\square$

**Lemma 12.** *Under Assumptions 2, 4–7, take $\gamma_i \asymp i^{-\zeta}$ with $\zeta \in (0,1)$, we have for $r \geq 0$, there exist positive constants $C_3$ and $C_4$ such that*

$$
Var(n, \{\gamma_i\}_i, \Sigma, \{\mathcal{T}_i\}_i, \{\Xi_i^r\}_i) = \frac{2}{n^2} \mathbb{E} \left\| \Sigma^{1/2} \sum_{j=1}^{n} \sum_{k=1}^{j} \prod_{i=k+1}^{j} (I - \gamma_i\mathcal{T}_{i-1}) \gamma_k \Xi_k^r \right\|_{\Sigma}^2
$$

$$
\leq \begin{cases} C_3 c^{-r} \gamma_0^r B^{2r} \widetilde{\sigma}^2 n^{(1-\zeta-\alpha)/\alpha}, & \text{if } 0 < \zeta \leq 1/2 \\ C_4 c^{-r} \gamma_0^r B^{2r} \widetilde{\sigma}^2 n^{(2\alpha\zeta+1-\zeta-2\alpha)/\alpha}, & \text{if } 1/2 < \zeta < 1 \end{cases}.
$$

*Proof of Lemma 12.* The proof is similar to that of Lemma 7 in Dieuleveut and Bach [2016]. Utilizing Lemmas 1 and 9, we have

$$
\text{Var}(n, \{\gamma_i\}_i, \Sigma, \{\mathcal{T}_i\}_i, \{\Xi_i^r\}_i) \leq \frac{2}{n^2} \mathbb{E} \left\| \Sigma^{1/2} \sum_{j=1}^{n} \sum_{k=1}^{j} \prod_{i=k+1}^{j} (I - c\gamma_i\Sigma) \gamma_k \Xi_k^r \right\|_{\Sigma}^2
$$

$$
= \frac{2}{n^2} \sum_{k=1}^{n} \gamma_k^2 \mathbb{E} \left[ \text{tr} \left( \left( \sum_{j=k}^{n} \prod_{i=k+1}^{j} (I - c\gamma_i\Sigma) \right) \Sigma \left( \sum_{j=k}^{n} \prod_{i=k+1}^{j} (I - c\gamma_i\Sigma) \right) \Xi_k^r \otimes \Xi_k^r \right) \right]
$$

$$
\leq c^{-r} \gamma_0^r B^{2r} \widetilde{\sigma}^2 \frac{2}{n^2} \sum_{k=1}^{n} \gamma_k^2 \text{tr} \left[ \left( \sum_{j=k}^{n} \prod_{i=k+1}^{j} (I - c\gamma_i\Sigma) \right) \Sigma \right]^2
$$

$$
\leq c^{-r} \gamma_0^r B^{2r} \widetilde{\sigma}^2 \frac{2C'}{n^2} \sum_{k=1}^{n} \gamma_k^2 \sum_{t=1}^{\infty} \left[ \left( \sum_{j=k}^{n} \prod_{i=k+1}^{j} (1 - c\gamma_i t^{-\alpha}) \right) t^{-\alpha} \right]^2.
$$

Take $\gamma_i \asymp i^{-\zeta}$. Utilizing $1 - x \leq \exp(-x)$, we have

$$
\sum_{j=k}^{n} \prod_{i=k+1}^{j} (1 - c\gamma_i t^{-\alpha}) \leq \sum_{j=k}^{n} \exp\{-c' \sum_{i=k+1}^{j} i^{-\zeta} t^{-\alpha}\}
$$

$$
= \sum_{j=k}^{n} \exp\left\{-c' t^{-\alpha} \frac{(j+1)^{1-\zeta} - (k+1)^{1-\zeta}}{1-\zeta}\right\}
$$

$$
\leq \int_{k+1}^{n+1} \exp\left\{-c' t^{-\alpha} \frac{x^{1-\zeta} - (k+1)^{1-\zeta}}{1-\zeta}\right\} dx
$$

$$
\leq c'' \max\{t^{\alpha/(1-\zeta)}, t^{\alpha}(k+1)^{\zeta}\}.
$$

Then

$$
\mathrm{Var}(n, \{\gamma_i\}_i, \Sigma, \{\mathcal{T}_i\}_i, \{\Xi_i^r\}_i) \leq \frac{2}{n^2} \mathbb{E} \left\| \Sigma^{1/2} \sum_{j=1}^{n} \sum_{k=1}^{j} \prod_{i=k+1}^{j} (I - c\gamma_i \Sigma) \gamma_k \Xi_k^r \right\|_{\Sigma}^2
$$

$$
\leq c^{-r} \gamma_0^r B^{2r} \widetilde{\sigma}^2 \frac{2C'}{n^2} \sum_{k=1}^{n} \gamma_k^2 \sum_{t=1}^{\infty} t^{-2\alpha} \left( \min\{n-k, c'' \max\{t^{\alpha/(1-\zeta)}, t^{\alpha}(k+1)^{\zeta}\}\} \right)^2
$$

$$
\leq c^{-r} \gamma_0^r B^{2r} \widetilde{\sigma}^2 \left[ \frac{C''}{n^2} \sum_{k=1}^{n} \gamma_k^2 \sum_{t=1}^{\infty} t^{-2\alpha} \min\{(n-k)^2, t^{2\alpha/(1-\zeta)}\} \right.
$$

$$
\left. + \frac{C''}{n^2} \sum_{k=1}^{n} \gamma_k^2 \sum_{t=1}^{\infty} t^{-2\alpha} \min\{(n-k)^2, t^{2\alpha}(k+1)^{2\zeta}\} \right].
$$

Utilizing the same arguments as in Lemma 7 in Dieuleveut and Bach [2016], we have,

$$
\frac{C''}{n^2} \sum_{k=1}^{n} \gamma_k^2 \sum_{t=1}^{\infty} t^{-2\alpha} \min\{(n-k)^2, t^{2\alpha/(1-\zeta)}\} \leq \begin{cases} C_1 n^{(1-\zeta-\alpha)/\alpha}, & \text{if } \zeta \leq 1/2 \\ C_1 n^{(2\alpha\zeta+1-\zeta-2\alpha)/\alpha}, & \text{if } \zeta > 1/2 \end{cases}.
$$

Also,

$$
\frac{C''}{n^2} \sum_{k=1}^{n} \gamma_k^2 \sum_{t=1}^{\infty} t^{-2\alpha} \min\{(n-k)^2, t^{2\alpha}(k+1)^{2\zeta}\} \leq C_2 n^{(1-\zeta-\alpha)/\alpha}.
$$

Thus,

$$
\mathrm{Var}(n, \{\gamma_i\}_i, \Sigma, \{\mathcal{T}_i\}_i, \{\Xi_i^r\}_i) \leq \begin{cases} C_3 c^{-r} \gamma_0^r B^{2r} \widetilde{\sigma}^2 n^{(1-\zeta-\alpha)/\alpha}, & \text{if } 0 < \zeta \leq 1/2 \\ C_4 c^{-r} \gamma_0^r B^{2r} \widetilde{\sigma}^2 n^{(2\alpha\zeta+1-\zeta-2\alpha)/\alpha}, & \text{if } 1/2 < \zeta < 1 \end{cases}.
$$

$\square$

Following Lemmas 1, 8–12, we give the proof of Theorem 3.

*Proof of Theorem 3.* Denote $\eta_n = \hat{f}_n - f_{\mathcal{H}}$. The recursion (7) implies that

$$
\eta_n = \left( I - \gamma_n w_\tau (Y_n - \hat{f}_{n-1}(X_n)) K_{X_n} \otimes K_{X_n} \right) \eta_{n-1} + \gamma_n \Xi_n, \tag{17}
$$

where $\Xi_n = (Y_n - f_{\mathcal{H}}(X_n)) w_\tau (Y_n - \hat{f}_{n-1}(X_n)) K_{X_n} + \xi_n$. We also decompose the recursion formula (17) into the initial and noise components. Specifically, the initial component $\{\eta_n^{\mathrm{init}}\}_n$ is defined as $\eta_0^{\mathrm{init}} = -f_{\mathcal{H}}$, and

$$
\eta_n^{\mathrm{init}} = \left( I - \gamma_n w_\tau (Y_n - \hat{f}_{n-1}(X_n)) K_{X_n} \otimes K_{X_n} \right) \eta_{n-1}^{\mathrm{init}};
$$

and the noise component $\{\eta_n^{\mathrm{noise}}\}_n$ satisfies $\eta_0^{\mathrm{noise}} = 0$, and

$$
\eta_n^{\mathrm{noise}} = \left( I - \gamma_n w_\tau (Y_n - \hat{f}_{n-1}(X_n)) K_{X_n} \otimes K_{X_n} \right) \eta_{n-1}^{\mathrm{noise}} + \gamma_n \Xi_n.
$$

By Minkowski's inequality,

$$\left(\mathbb{E}\|\bar{\eta}_n\|^2_{L^2_{P_X}}\right)^{1/2} \le \left(\mathbb{E}\|\bar{\eta}_n^{\text{init}}\|^2_{L^2_{P_X}}\right)^{1/2} + \left(\mathbb{E}\|\bar{\eta}_n^{\text{noise}}\|^2_{L^2_{P_X}}\right)^{1/2}, \tag{18}$$

where $\bar{\eta}_n = \sum_{j=1}^n \eta_j/n$, $\bar{\eta}_n^{\text{init}} = \sum_{j=1}^n \eta_j^{\text{init}}/n$, and $\bar{\eta}_n^{\text{noise}} = \sum_{j=1}^n \eta_j^{\text{noise}}/n$. Next, we will respectively present the upper bounds of $\mathbb{E}\|\bar{\eta}_n^{\text{init}}\|^2_{L^2_{P_X}}$ and $\mathbb{E}\|\bar{\eta}_n^{\text{noise}}\|^2_{L^2_{P_X}}$.

**Noise component.** Denote $\mathcal{T}_{n-1} = \mathbb{E}\left[w_\tau(Y_n - \hat{f}_{n-1}(X_n))K_{X_n} \otimes K_{X_n}|\mathcal{F}_{n-1}\right]$. For any $r \ge 0$, define a sequence $\{\eta_n^{\text{noise},r}\}_n$ as follows:

$$\eta_n^{\text{noise},r} = (I - \gamma_n\mathcal{T}_{n-1})\,\eta_{n-1}^{\text{noise},r} + \gamma_n\Xi_n^r, \text{ with } \eta_0^{\text{noise},r} = 0,$$

where $\Xi_n^0 = \Xi_n$, and $\Xi_n^r = \left(\mathcal{T}_{n-1} - w_\tau(Y_n - \hat{f}_{n-1}(X_n))K_{X_n} \otimes K_{X_n}\right)\eta_{n-1}^{\text{noise},r-1}$ for $r \ge 1$. Then $\eta_0^{\text{noise}} - \sum_{i=0}^r \eta_0^{\text{noise},i} = 0$, and

$$\eta_n^{\text{noise}} - \sum_{i=0}^r \eta_n^{\text{noise},i} = \left(I - \gamma_n w_\tau(Y_n - \hat{f}_{n-1}(X_n))K_{X_n} \otimes K_{X_n}\right)\left(\eta_{n-1}^{\text{noise}} - \sum_{i=0}^r \eta_{n-1}^{\text{noise},i}\right) + \gamma_n\Xi_n^{r+1}.$$

Minkowski's inequality implies that

$$\left(\mathbb{E}\|\bar{\eta}_n^{\text{noise}}\|^2_{L^2_{P_X}}\right)^{1/2} \le \sum_{i=0}^r \left(\mathbb{E}\|\bar{\eta}_n^{\text{noise},i}\|^2_{L^2_{P_X}}\right)^{1/2} + \left(\mathbb{E}\left\|\bar{\eta}_n^{\text{noise}} - \sum_{i=0}^r \bar{\eta}_n^{\text{noise},i}\right\|^2_{L^2_{P_X}}\right)^{1/2},$$

where $\bar{\eta}_n^{\text{noise},i} = \sum_{j=1}^n \eta_j^{\text{noise},i}/n$. By Lemma 9, for any $r \ge 0$ and any $n \ge 0$, $\mathbb{E}(\Xi_n^r \otimes \Xi_n^r) \preccurlyeq c^{-r}\gamma_0^r B^{2r}\widetilde{\sigma}^2\Sigma$. Utilizing Lemma 8 with $\alpha_0 = 0$ and $\alpha_n = \eta_n^{\text{noise},i}$, we have

$$\mathbb{E}\|\bar{\eta}_n^{\text{noise},i}\|^2_{L^2_{P_X}} \le \text{Var}(n, \{\gamma_j\}_j, \Sigma, \{\mathcal{T}_j\}_j, \{\Xi_j^i\}_j). \tag{19}$$

For the residual term $\eta_n^{\text{noise}} - \sum_{i=0}^r \eta_n^{\text{noise},i}$, we can utilize Lemma 10 with $\alpha_n^r = \eta_n^{\text{noise}} - \sum_{i=0}^r \eta_n^{\text{noise},i}$ and $\Xi_n^\alpha = \Xi_n^{r+1}$. If $\{\gamma_n\}_n$ is non-increasing and satisfies $c^{-1}\gamma_0 B^2 < 1$, then

$$\mathbb{E}\left\langle \bar{\eta}_n^{\text{noise}} - \sum_{i=0}^r \bar{\eta}_n^{\text{noise},i}, \Sigma\left(\bar{\eta}_n^{\text{noise}} - \sum_{i=0}^r \bar{\eta}_n^{\text{noise},i}\right)\right\rangle_{L^2_{P_X}}$$

$$\le \frac{1}{2cn(1 - c^{-1}\gamma_0 B^2)}\left[\sum_{k=1}^{n-1}\mathbb{E}\|\alpha_k^r\|^2_{L^2_{P_X}}\left(-\frac{1}{\gamma_k} + \frac{1}{\gamma_{k+1}}\right) + 2\sum_{k=1}^n \gamma_k\mathbb{E}\|\Xi_k^{r+1}\|^2_{L^2_{P_X}}\right].$$

Utilizing Lemma 9,

$$\sum_{k=1}^n \gamma_k\mathbb{E}\|\Xi_k^{r+1}\|^2_{L^2_{P_X}} \le \sum_{k=1}^n \gamma_k\text{tr}\left(\mathbb{E}(\Xi_k^{r+1} \otimes \Xi_k^{r+1})\right) \le nc^{-(r+1)}\gamma_0^{r+2}B^{2(r+1)}\widetilde{\sigma}^2\text{tr}(\Sigma).$$

Take $\gamma_i \asymp i^{-\zeta}$. Note that $\|\alpha_i^r\| \le \|\alpha_{i-1}^r\| + \gamma_i\|\Xi_i^{r+1}\| \le \sum_{k=1}^i \gamma_k\|\Xi_k^{r+1}\|$. Then

$$\frac{1}{n}\sum_{k=1}^{n-1}\mathbb{E}\|\alpha_k^r\|^2_{L^2_{P_X}}\left(-\frac{1}{\gamma_k} + \frac{1}{\gamma_{k+1}}\right) \le 2c'\zeta\frac{1}{n}\sum_{i=1}^{n-1}\frac{1}{i\gamma_i}\mathbb{E}\|\alpha_k^r\|^2$$

$$\le 2c'\zeta\frac{1}{n}\sum_{i=1}^{n-1}\frac{1}{i\gamma_i}\left[\left(\sum_{k=1}^i \gamma_k\right)\left(\sum_{k=1}^i \gamma_k\mathbb{E}(\|\Xi_k^{r+1}\|^2)\right)\right]$$

$$\le \frac{c'\zeta}{1 - \zeta}nc^{-(r+1)}\gamma_0^{r+2}B^{2(r+1)}\widetilde{\sigma}^2\text{tr}(\Sigma).$$

It follows that

$$
\begin{aligned}
&\mathbb{E}\left\|\bar{\eta}_n^{\text{noise}} - \sum_{i=0}^r \bar{\eta}_n^{\text{noise},i}\right\|_{L_{P_X}^2}^2 \\
&\leq \frac{1}{2c(1-c^{-1}\gamma_0 B^2)}\left[\frac{c'\zeta}{1-\zeta}nc^{-(r+1)}\gamma_0^{r+2}B^{2(r+1)}\widetilde{\sigma}^2\text{tr}(\Sigma) + 2c^{-(r+1)}\gamma_0^{r+2}B^{2(r+1)}\widetilde{\sigma}^2\text{tr}(\Sigma)\right] \\
&\leq \widetilde{c}nc^{-(r+1)}\gamma_0^{r+2}B^{2(r+1)}\widetilde{\sigma}^2\text{tr}(\Sigma).
\end{aligned}
\tag{20}
$$

Combining (19), (20), and Lemma 12, we have

$$
\begin{aligned}
&\left(\mathbb{E}\|\bar{\eta}_n^{\text{noise}}\|_{L_{P_X}^2}^2\right)^{1/2} \\
&\leq \sum_{i=0}^r\left[\text{Var}(n, \{\gamma_j\}_j, \Sigma, \{\mathcal{T}_j\}_j, \{\Xi_j^i\}_j)\right]^{1/2} + \left[c'nc^{-(r+1)}\gamma_0^{r+2}B^{2(r+1)}\widetilde{\sigma}^2\text{tr}(\Sigma)\right]^{1/2} \\
&\leq \widetilde{\sigma}\left[C_3 n^{(1-\zeta-\alpha)/\alpha}I\{0 < \zeta \leq 1/2\} + C_4 n^{(2\alpha\zeta+1-\zeta-2\alpha)/\alpha}I\{1/2 < \zeta < 1\}\right]^{1/2}\sum_{i=0}^r(c^{-1}\gamma B^2)^{i/2} \\
&+ \left[c'nc^{-(r+1)}\gamma_0^{r+2}B^{2(r+1)}\widetilde{\sigma}^2\text{tr}(\Sigma)\right]^{1/2} \\
&\leq \frac{\widetilde{\sigma}}{1-(c^{-1}\gamma B^2)^{1/2}}\left[C_3 n^{(1-\zeta-\alpha)/\alpha}I\{0 < \zeta \leq 1/2\} + C_4 n^{(2\alpha\zeta+1-\zeta-2\alpha)/\alpha}I\{1/2 < \zeta < 1\}\right]^{1/2} \\
&+ \left[c'nc^{-(r+1)}\gamma_0^{r+2}B^{2(r+1)}\widetilde{\sigma}^2\text{tr}(\Sigma)\right]^{1/2}.
\end{aligned}
$$

Let the recursion step $r \to \infty$, if $c^{-1}\gamma B^2 < 1$, then we have

$$
\left(\mathbb{E}\|\bar{\eta}_n^{\text{noise}}\|_{L_{P_X}^2}^2\right)^{1/2} \leq
\begin{cases}
\dfrac{C_3^{1/2}\widetilde{\sigma}}{1-(c^{-1}\gamma B^2)^{1/2}}n^{(1-\zeta-\alpha)/(2\alpha)}, & \text{if } 0 < \zeta \leq 1/2 \\[2ex]
\dfrac{C_4^{1/2}\widetilde{\sigma}}{1-(c^{-1}\gamma B^2)^{1/2}}n^{(2\alpha\zeta+1-\zeta-2\alpha)/(2\alpha)}, & \text{if } 1/2 < \zeta < 1
\end{cases}
\tag{21}
$$

**Initial component.** The main recursion is

$$
\eta_n^{\text{init},0} = (I - \gamma_n \mathcal{T}_{n-1})\eta_{n-1}^{\text{init},0}
$$

with $\eta_0^{\text{init},0} = -f_{\mathcal{H}}$, and the residual term is

$$
\eta_n^{\text{init}} - \eta_n^{\text{init},0} = \left(I - \gamma_n w_\tau(Y_n - \hat{f}_{n-1}(X_n))K_{X_n} \otimes K_{X_n}\right)\left(\eta_{n-1}^{\text{init}} - \eta_{n-1}^{\text{init},0}\right) + \gamma_n\Xi_n^{\text{init}}
$$

with $\eta_0^{\text{init}} - \eta_0^{\text{init},0} = 0$, where $\Xi_n^{\text{init}} = \left(\mathcal{T}_{n-1} - w_\tau(Y_n - \hat{f}_{n-1}(X_n))K_{X_n} \otimes K_{X_n}\right)\eta_{n-1}^{\text{init},0}$. Utilizing Lemmas 8 and 10, we have

$$
\mathbb{E}\langle\bar{\eta}_n^{\text{init},0}, \Sigma\bar{\eta}_n^{\text{init},0}\rangle_{L_{P_X}^2} \leq \text{Bias}(n, \{\gamma_i\}_i, \Sigma, \{\mathcal{T}_i\}_i, f_{\mathcal{H}}),
$$

and

$$
\begin{aligned}
&\mathbb{E}\langle\bar{\eta}_n^{\text{init}} - \bar{\eta}_n^{\text{init},0}, \Sigma\left(\bar{\eta}_n^{\text{init}} - \bar{\eta}_n^{\text{init},0}\right)\rangle_{L_{P_X}^2} \\
&\leq \frac{1}{2cn(1-c^{-1}\gamma_0 B^2)}\left[\sum_{k=1}^{n-1}\mathbb{E}\|\eta_k^{\text{init}} - \eta_k^{\text{init},0}\|_{L_{P_X}^2}^2\left(-\frac{1}{\gamma_k} + \frac{1}{\gamma_{k+1}}\right) + 2\sum_{k=1}^n\gamma_k\mathbb{E}\|\Xi_k^{\text{init}}\|_{L_{P_X}^2}^2\right],
\end{aligned}
$$

where

$$
\text{Bias}(n, \{\gamma_i\}_i, \Sigma, \{\mathcal{T}_i\}_i, f_{\mathcal{H}}) = \frac{2}{n^2}\mathbb{E}\left\|\Sigma^{1/2}\sum_{j=1}^n\prod_{i=1}^j(I - \gamma_i\mathcal{T}_{i-1})f_{\mathcal{H}}\right\|_\Sigma^2.
$$

Note that

$$\mathbb{E}\|\Xi_k^{\text{init}}\|_{L_{P_X}^2}^2 \le B^2 \mathbb{E}\left\langle f_{\mathcal{H}}, \left(\prod_{i=1}^k (I - \gamma_i \mathcal{T}_{i-1})\right)^2 \Sigma f_{\mathcal{H}}\right\rangle_{L_{P_X}^2}.$$

By Lemma 1, for any $r$, we have

$$\frac{1}{n}\sum_{k=1}^n \gamma_k \mathbb{E}\|\Xi_k^{\text{init}}\|_{L_{P_X}^2}^2 \le \frac{B^2}{n}\sum_{k=1}^n \gamma_k \mathbb{E}\left\langle f_{\mathcal{H}}, \left(\prod_{i=1}^k (I - \gamma_i \mathcal{T}_{i-1})\right)^2 \Sigma f_{\mathcal{H}}\right\rangle_{L_{P_X}^2}$$

$$\le \frac{B^2}{n}c^{-1}\left\|\sum_{k=1}^n \left(\prod_{i=1}^k (I - c\gamma_i \Sigma)\right)^2 c\gamma_k \Sigma^{2r}\right\|_{L_{P_X}^2} \left\|\Sigma^{-r} f_{\mathcal{H}}\right\|_{L_{P_X}^2}^2.$$

The proof of Theorem 3 in Dieuleveut and Bach [2016] implies that, if $2r - \frac{1}{1-\zeta} < 0$, then

$$\left\|\sum_{k=1}^n \left(\prod_{i=1}^k (I - c\gamma_i \Sigma)\right)^2 c\gamma_k \Sigma^{2r}\right\|_{L_{P_X}^2} \le c'' n (n\gamma_n)^{-2r}.$$

It follows that

$$\frac{1}{n}\sum_{k=1}^n \gamma_k \mathbb{E}\|\Xi_k^{\text{init}}\|_{L_{P_X}^2}^2 \le c'' c^{-1} B^2 (n\gamma_n)^{-2r} \left\|\Sigma^{-r} f_{\mathcal{H}}\right\|_{L_{P_X}^2}^2.$$

Denote $\alpha_n = \eta_n^{\text{init}} - \eta_n^{\text{init},0}$. Lemma 10 implies that

$$\mathbb{E}\|\alpha_i\|^2 \le \mathbb{E}\|\alpha_{i-1}\|^2 + 2\gamma_i^2 \mathbb{E}\|\Xi_i^{\text{init}}\|^2 \le \sum_{k=1}^i 2\gamma_k^2 \mathbb{E}\|\Xi_k^{\text{init}}\|^2 \le \sum_{k=1}^i 2\gamma_k \mathbb{E}\|\Xi_k^{\text{init}}\|^2.$$

Taking $\gamma_i \asymp i^{-\zeta}$, we have

$$\frac{1}{n}\sum_{k=1}^{n-1} \mathbb{E}\|\alpha_k\|_{L_{P_X}^2}^2 \left(-\frac{1}{\gamma_k} + \frac{1}{\gamma_{k+1}}\right) \le 2c'\zeta \frac{1}{n}\sum_{i=1}^{n-1} \frac{1}{i\gamma_i}\mathbb{E}\|\alpha_k\|^2$$

$$\le 4c'c''c^{-1}\zeta B^2 \frac{1}{n}\sum_{i=1}^{n-1} i(i\gamma_i)^{-2r-1} \left\|\Sigma^{-r} f_{\mathcal{H}}\right\|_{L_{P_X}^2}^2$$

$$\le c''' B^2 \gamma_n^{-1}(n\gamma_n)^{-2r} \left\|\Sigma^{-r} f_{\mathcal{H}}\right\|_{L_{P_X}^2}^2.$$

It follows that

$$\mathbb{E}\|\bar{\eta}_n^{\text{init}} - \bar{\eta}_n^{\text{init},0}\|_{L_{P_X}^2}^2 \le O\left((n\gamma_n)^{-2r}\left\|\Sigma^{-r} f_{\mathcal{H}}\right\|_{L_{P_X}^2}^2\right) + O\left(\gamma_n^{-1}(n\gamma_n)^{-2r}\left\|\Sigma^{-r} f_{\mathcal{H}}\right\|_{L_{P_X}^2}^2\right)$$

$$\le O\left(\gamma_n^{-1}(n\gamma_n)^{-2r}\left\|\Sigma^{-r} f_{\mathcal{H}}\right\|_{L_{P_X}^2}^2\right).$$

Thus,

$$\left(\mathbb{E}\|\bar{\eta}_n^{\text{init}}\|_{L_{P_X}^2}^2\right)^{1/2} \le \text{Bias}(n, \{\gamma_i\}_i, \Sigma, \{\mathcal{T}_i\}_i, f_{\mathcal{H}})^{1/2} + O\left(\gamma_n^{-1/2}(n\gamma_n)^{-r}\left\|\Sigma^{-r} f_{\mathcal{H}}\right\|_{L_{P_X}^2}\right).$$

Take $\gamma_i \asymp i^{-\zeta}$ with $\zeta \in (0,1)$. By Lemma 11, if $r - (1 + \zeta/2)/(1 - \zeta) \le 0$, then

$$\left(\mathbb{E}\|\bar{\eta}_n^{\text{init}}\|_{L_{P_X}^2}^2\right)^{1/2} \le O\left(\gamma_n^{-1/2}(n\gamma_n)^{-r}\left\|\Sigma^{-r} f_{\mathcal{H}}\right\|_{L_{P_X}^2}\right); \tag{22}$$

if $r - (1 + \zeta/2)/(1 - \zeta) > 0$, then

$$\left(\mathbb{E}\|\bar{\eta}_n^{\text{init}}\|_{L_{P_X}^2}^2\right)^{1/2} \le O\left(n^{-1}\left\|\Sigma^{-r} f_{\mathcal{H}}\right\|_{L_{P_X}^2}\right).$$

Note that $1/(2(1-\zeta)) < (1+\zeta/2)/(1-\zeta)$ for any $\zeta \in (0,1)$. Combining (18), (21), and (22), take $\zeta$ satisfying $2r - 1/(1-\zeta) < 0$, if $\zeta \in (0,1/2]$, then

$$\left(\mathbb{E}\|\bar{\eta}_n\|^2_{L^2_{P_X}}\right)^{1/2} \leq O\left(\widetilde{\sigma} n^{(1-\zeta-\alpha)/(2\alpha)}\right) + O\left(\gamma_n^{-1/2}(n\gamma_n)^{-r}\left\|\Sigma^{-r}f_\mathcal{H}\right\|_{L^2_{P_X}}\right);$$

if $\zeta \in (1/2,1)$, then

$$\left(\mathbb{E}\|\bar{\eta}_n\|^2_{L^2_{P_X}}\right)^{1/2} \leq O\left(\widetilde{\sigma} n^{(2\alpha\zeta+1-\zeta-2\alpha)/(2\alpha)}\right) + O\left(\gamma_n^{-1/2}(n\gamma_n)^{-r}\left\|\Sigma^{-r}f_\mathcal{H}\right\|_{L^2_{P_X}}\right).$$

Thus, taking $\zeta$ satisfying $2r - 1/(1-\zeta) < 0$, we have

$$\mathbb{E}\|\bar{f}_n - f_\mathcal{H}\|^2_{L^2_{P_X}} \leq \begin{cases} O\left(\widetilde{\sigma}^2\gamma_n^{1/\alpha}n^{-1+1/\alpha} + \gamma_n^{-1}(n\gamma_n)^{-2r}\left\|\Sigma^{-r}f_\mathcal{H}\right\|^2_{L^2_{P_X}}\right), & \text{if } 0 < \zeta \leq 1/2 \\ O\left(\widetilde{\sigma}^2(n\gamma_n)^{-2+1/\alpha} + \gamma_n^{-1}(n\gamma_n)^{-2r}\left\|\Sigma^{-r}f_\mathcal{H}\right\|^2_{L^2_{P_X}}\right), & \text{if } 1/2 < \zeta < 1 \end{cases}.$$

$\square$

