# OpenReview forum: "Online robust locally differentially private learning for nonparametric regression"
_NeurIPS.cc/2025/Conference — NeurIPS 2025 poster_

### Official Review · Reviewer_HRn7 · 2025-06-25

**Clarity:** 4
**Significance:** 2
**Originality:** 3
**Rating:** 5
**Confidence:** 3

**Summary:**

This paper addresses the problem of nonparametric regression in a streaming data setting, where data arrives sequentially and privacy is a major concern. The authors propose a novel framework and two new algorithms to tackle this challenge. The first algorithm, H-FSGD (Huber Functional Stochastic Gradient Descent), is a one-pass online method that uses the Huber loss to achieve robustness against heavy-tailed errors and outliers. The second algorithm, PH-FSGD, extends H-FSGD to provide Local Differential Privacy (LDP) by injecting calibrated Gaussian process noise at each iteration, thus protecting individual data points at the source without a trusted curator. The paper provides a comprehensive non-asymptotic convergence analysis for both estimators, establishing optimal convergence rates and exploring the trade-offs involving step size and privacy budget. The theoretical findings are supported by experiments on synthetic data, demonstrating the robustness and privacy-preserving capabilities of the proposed methods.

**Questions:**

I have a few questions for the authors that I believe could help clarify some points and strengthen the paper:

1. **Real-world Performance:** I understand that setting up real-world streaming experiments can be challenging. However, could you perhaps comment on how you expect your method to perform on a benchmark streaming dataset? Even a discussion of the potential challenges (e.g., concept drift, non-i.i.d. data) and how your framework might adapt would be very insightful.

2. **Robustness in Contamination Models:** Your use of Huber loss is well-motivated for heavy-tailed noise. I'm curious if you have any intuition on how the method would perform under a stronger threat model, like Huber's contamination model, where an adversary could corrupt a small fraction of the data points. Does your analysis provide any clues about the breakdown point of the estimator?

3. **The Robustness-Privacy-Utility Trilemma:** There's a known tension in the local model between robustness, privacy, and statistical utility (see e.g. "On the Privacy-Robustness-Utility Trilemma in Distributed Learning", ICML '23). It seems your framework has all the knobs to explore this: the Huber parameter τ, the privacy budget (ϵ,δ), and the step size γ. Could you elaborate on the interplay between these? For example, how should one co-tune τ and ϵ to achieve a good trade-off, especially when outliers are present?

4. A quick note on notation: In line 185, you define $w_\tau$ in terms of a variable $u$. It's clear from the line above that $u=y−f(x)$, but for maximum clarity, you might consider making that explicit in the definition itself.

**Ethical Concerns:**

["NO or VERY MINOR ethics concerns only"]

**Final Justification:**

I agree with other reviewers that the paper clarity could be improved, e.g., definitions of differential privacy and outlier robustness. For the interplay between privacy and robustness, I recommend that the authors cite the trilemma work I referenced in my original review and any other relevant ones. The rebuttal and new experiments have addressed my initial concerns, and will introduce important changes to the paper. I maintain my positive assessment of the paper.

**Limitations:**

Yes.

**Quality:**

4

**Strengths And Weaknesses:**

### Strengths

1. **Significance and Originality:** The paper tackles an important problem at the intersection of streaming data analysis, robust statistics, and privacy. To the best of the authors' knowledge and my own, developing a scalable, statistically sound method for online, robust, and locally private nonparametric regression is an open problem. The proposed framework, which combines functional SGD, Huber loss for robustness, and LDP for privacy, is a novel contribution to this area.

2. **Theoretical Quality:** The paper provides a rigorous and comprehensive non-asymptotic convergence analysis. The authors establish finite-sample guarantees for their estimators in a general setting that does not assume the true function lies within the chosen RKHS. The analysis of convergence rates under various smoothness assumptions and step-size schemes is thorough and provides valuable theoretical insights into the behavior of the algorithms. The proofs provided in the appendix appear to be detailed and correct.

3. **Clarity:** The paper is well-written. The authors clearly motivate the problem, lay out their contributions, and present their algorithms in a way that is easy to follow. The inclusion of a comparison table and flowchart is also helpful for understanding the overall structure of their contribution.

### Weaknesses

1. **Experimental Scope:** My main reservation about this paper is the limited scope of the experiments. While the synthetic data simulations do a good job of supporting the theoretical claims about efficiency and privacy, they stop short of showing how the method would perform on a real-world streaming dataset. Given that the work is motivated by practical applications like sensor networks and autonomous vehicles, the absence of a real-data experiment feels like a missed opportunity to demonstrate the framework's utility.

2. **Characterization of Robustness:** The paper's notion of robustness is tied to handling heavy-tailed errors via the Huber loss. This is a perfectly reasonable approach, but it would be interesting to see a discussion of how this translates to robustness in an adversarial sense, for instance, in a Huber contamination model where a fraction of the data might be arbitrarily corrupted. The trade-off between the degree of robustness and the privacy guarantee is a subtle but important aspect that isn't fully explored. An example of this in the local privacy model is "On the Privacy-Robustness-Utility Trilemma in Distributed Learning", ICML '23.

3. **Incremental Technical Novelty:** While the synthesis of ideas here is original and valuable, the core components (Functional SGD, Huber loss, and the Gaussian mechanism for privacy) are all well-established tools in their respective fields. The contribution is therefore less about creating a new technical tool and more about demonstrating how existing tools can be thoughtfully combined and analyzed to solve a new, complex problem. This is a valid and useful contribution, but it's worth noting.

---

> ### Author Rebuttal · Authors · 2025-07-29
>
> Thank you for your thoughtful and critical assessment. Many of your comments will help us produce a more readable and self-contained version of the paper. Below, we address each of your specific concerns in turn.
>
> - **Real-world performance.**
>   - **Real data analysis.** To demonstrate the practical utility of our method, we evaluated it on the “Health and Fitness” dataset from Kaggle website, aiming to assess the impact of endurance on overall fitness. We used 40000 samples for training and 1000 for testing. The table below reports out-of-sample $R^2$ comparisons across methods.
>
>        | Step size \ Method | H-FSGD | L2-FSGD | Offline | (2,0.2)-LDP | (3,0.1)-LDP |  lm   |
>     | :----------------: | :----: | :-----: | :-----: | :---------: | :---------: | :---: |
>     |    **Constant**    | 0.622  |  0.617  |  0.609  |    0.589    |    0.609    | 0.581 |
>     |  **Non-constant**  | 0.623  |  0.618  |  0.609  |    0.526    |    0.596    | 0.581 |
>
>     H-FSGD consistently outperforms L2-FSGD and Offline methods in streaming settings. Under LDP constraints, PH-FSGD remains competitive with or exceeds linear regression, underscoring its robustness and practical relevance. If accepted, we will include this analysis in the final version.
>   - **Extension to non-iid data.** We agree that addressing concept drift and non-i.i.d. data is both important and challenging. Our current framework, like standard functional SGD theory, assumes i.i.d. data,  and does not directly extend to non-i.i.d. settings—a limitation shared by many SGD-based methods. Challenges include parameter drift, slower convergence  (see, e.g., "Federated learning with non-iid data", *arXiv*, 2018; "On the convergence of FedAvg on non-IID data", *ICLR*, 2020), and bias from dependent observations. While one could define the population risk as $\sum\_{n=1}^{\infty}p_nL\_{\tau}(Y_n-f_n(X_n))$ with distribution weight $p_n$, this is infeasible in online settings due to unknown distribution shifts and the ill-defined infinite-sum objective. A more practical alternative models the data as coming from $M$ sub-populations with different distributions, minimizing the global risk $\sum_{m=1}^M p_m L\_{\tau}(Y^{(m)}-f(X^{(m)}))$. This leads to a parallel SGD scheme: at each round $t$, the global estimate $\hat{f}\_t$ is sent to local devices, which compute updates $$\hat{f}^{(m)}\_{t+1}=\hat{f}\_t-\gamma_t \widehat{\nabla L}\_\tau (\hat{f}\_t)(X_n^{(m)}, Y_n^{(m)})+\gamma_t \xi_t,$$ followed by weighted aggregation $$\hat{f}\_{t+1}=\sum_{m=1}^M p_m \hat{f}\_{t+1}^{(m)}.$$ This setup accommodates distributional heterogeneity and aligns with recent federated learning approaches for non-i.i.d. data  (see, e.g., "EFL: elastic federated learning on non-IID data", *Conference on lifelong learning agents*, 2022; "Asynchronous online federated learning for edge devices with non-iid data", *2020 IEEE International Conference on Big Data*, 2020). We will add this discussion to the revised version.
> - **Robustness in contamination models.**
>   - While our method is designed for heavy-tailed noise via the Huber loss, we also assess its robustness under Huber’s $\varepsilon^\star$-contamination model, where an $\varepsilon^\star$-fraction of data is adversarially corrupted. Using Case 1 (true model) and Case 2 (contaminated model), we simulate performance across varying $\varepsilon^\star$ at $n=10000$, reporting mean and standard deviation of MSE ($\times 10^{-2}$) over $50$ runs:
>
>     |    Step size     | $\varepsilon^\star$ |        0        |      0.1       |     0.2      |     0.3      |     0.4     |     0.5     |     0.6     |    0.7     |    0.8     |    0.9     |
>     | :--------------: | :-----------------: | :-------------: | :------------: | :----------: | :----------: | :---------: | :---------: | :---------: | :--------: | :--------: | :--------: |
>     |   **Constant**   |     **H-FSGD**      | 0.0722 (0.0158) | 0.657 (0.0985) | 2.85 (0.251) | 8.08 (0.497) | 20.4 (1.27) | 56.7 (3.91) | 114 (3.47)  | 155 (2.49) | 186 (2.21) | 211 (2.10) |
>     |                  |     **L2-FSGD**     | 0.0619 (0.015)  |  2.46 (0.263)  | 9.49 (0.611) | 21.2 (0.928) | 37.4 (1.16) | 58.4 (1.60)  |  84.0 (1.70)   | 114 (1.74) | 149 (2.15) | 189 (2.14) |
>     | **Non-constant** |     **H-FSGD**      | 0.0939 (0.0326) | 0.683 (0.105)  | 2.89 (0.26)  | 8.11 (0.517) | 20.2 (1.23) | 56.4 (4.23) | 114 (3.78)  | 155 (2.61) | 186 (2.32) | 211 (2.16) |
>     |                  |     **L2-FSGD**     |  0.152 (0.136)  |  2.55 (0.31)   | 9.63 (0.683) | 21.3 (1.01)  | 37.5 (1.23) | 58.5 (1.64) | 84.2 (1.83) | 114 (1.94) | 149 (2.25) | 189 (2.18) |
>
>     H-FSGD remains stable up to $\varepsilon^\star = 0.3$, with degradation beginning around $\varepsilon^\star = 0.4$, suggesting a breakdown point near 40%. In contrast, L2-FSGD deteriorates earlier at $\varepsilon^\star = 0.3$. Across all settings, H-FSGD consistently outperforms L2-FSGD. We will include these results in the revised version.
> - **Robustness-privacy-utility trilemma.**
>   - Interplay among Huber parameter, privacy budget, and step size.
>     - In the non-private setting, smaller $\tau$ improves robustness to outliers but increases bias—a classical bias–robustness trade-off (see, e.g., "Estimation of high dimensional mean regression in the absence of symmetry and light tail assumptions", *JRSSB*, 2017). In the private setting, a larger $\tau$ amplifies gradient sensitivity, requiring stronger noise for given $(\varepsilon, \delta)$-LDP, which raises gradient variance and necessitates smaller $\gamma_0$ for stability.
>     - A tighter privacy budget (e.g., smaller $\varepsilon$) also demands more noise, further constraining $\gamma_0$.
>     - The step size follows a schedule: $\gamma_i = \gamma_0 n^{-\zeta}$ (constant step size) or $\gamma_i = \gamma_0 i^{-\zeta}$ (non-constant). Corollaries 1–4 show the optimal $\zeta$ depends on smoothness parameters $r$ and $\alpha$. Under strong privacy noise, smaller $\gamma_0$ improves stability, but overly small steps slow convergence.
>
>   - Co-tune strategy in practice.
>
>     - Huber parameter: we first compute an L2-FSGD estimate and residuals, then estimate noise scale via MAD. The Huber parameter is set to $\tau = 1.345 \times \text{MAD}$, achieving $90\\%$ efficiency under normality while preserving robustness (Lines 191–193, Algorithm 2), following standard practice (see, e.g., "Robust regression using iteratively reweighted least-squares", *Communications in Statistics-theory and Methods*, 1977).
>
>     - Privacy budget: chosen based on application needs. Where possible, a looser budget (larger $\varepsilon$) reduces noise and improves utility.
>
>     - Step size: in the absence of theory-guided tuning, we recommend grid search for $\gamma_0$ and $\zeta$ on a validation set—a practical and widely used approach (see, e.g., "The step decay schedule: a near optimal, geometrically decaying learning rate procedure for least squares", *NeurIPS*, 2019; "Benefits of learning rate annealing for tuning‑robustness in stochastic optimization", *arxiv*, 2025). To assess sensitivity, we simulate H-FSGD under Case 1 using both step-size schedules, with $\gamma_0 \in [4, 24]$ and $\zeta \in [0.3, 0.8]$, and report average MSE ($\times 10^{-3}$) over 50 repetitions at $n = 10000$ (tables below). These results will be included in the revision.
>
>       - Constant step size setting:
>
>         | $\zeta$ \ $\gamma_0$ |   4   |   8   |  12  |  16  |  20  |  24  |
>         | :------------------: | :---: | :---: | :--: | :--: | :--: | :--: |
>         |       **0.3**        | 1.52  | 1.76  | 2.05 | 2.38 | 2.79 | 3.32 |
>         |       **0.4**        | 1.59  | 1.49  | 1.56 | 1.65 | 1.76 | 1.87 |
>         |       **0.5**        | 2.06  | 1.67  | 1.54 | 1.50 | 1.49 | 1.51 |
>         |       **0.6**        | 4.03  | 2.30  | 1.93 | 1.76 | 1.67 | 1.60 |
>         |       **0.7**        | 12.28 | 5.18  | 3.37 | 2.66 | 2.31 | 2.11 |
>         |       **0.8**        | 59.14 | 17.12 | 9.69 | 6.77 | 5.21 | 4.25 |
>
>       - Non-constant step size setting:
>
>         | $\zeta$ \ $\gamma_0$ |  4   |  8   |  12  |  16  |  20  |  24  |
>         | :------------------: | :--: | :--: | :--: | :--: | :--: | :--: |
>         |       **0.3**        | 1.63 | 1.99 | 2.44 | 3.04 | 4.13 | 5.12 |
>         |       **0.4**        | 1.51 | 1.62 | 1.77 | 1.96 | 2.13 | 2.33 |
>         |       **0.5**        | 1.69 | 1.59 | 1.63 | 1.66 | 1.80 | 1.85 |
>         |       **0.6**        | 2.27 | 1.94 | 1.92 | 1.84 | 1.87 | 1.93 |
>         |       **0.7**        | 3.59 | 3.14 | 2.95 | 2.85 | 2.68 | 2.63 |
>         |       **0.8**        | 7.74 | 6.27 | 6.50 | 6.46 | 6.58 | 6.39 |
>
>       Our results indicate that the proposed estimators exhibit robustness to the choice of $\gamma_0$ and $\zeta$ under both constant and non-constant step size settings, with wide parameter regions yielding stable and comparable MSEs.
>
> - **Incremental technical novelty.** While our work builds on established tools, its key novelty lies in being the first to adapt them to online, robust, and privacy-preserving nonparametric learning in streaming environments, a setting not addressed in prior work. We simultaneously tackle three challenging constraints: robustness to heavy-tailed noise, rigorous differential privacy, and efficient streaming updates. Our theoretical analysis establishes non-asymptotic guarantees under both constant and decaying step sizes, with or without LDP, and shows that proper step size tuning yields optimal convergence rates, extending prior results from finite- to infinite-dimensional settings. Finally, our experiments demonstrate superior empirical performance.
>
> - **Notation.** We agree that making the relationship $u=y-f(x)$ explicit in the definition of $w_{\tau}(u)$ will enhance readability and prevent any potential ambiguity.
>
> Thank you again for your comments. We hope our responses resolve your concerns. If anything remains unclear, we’d be happy to provide further clarification.

---

> > ### Comment · Reviewer_HRn7 · 2025-08-04
> >
> > Thank you for the detailed rebuttal. For the interplay between privacy and robustness, please cite the trilemma work I referenced in my original review and any other relevant ones. I agree with other reviewers that the paper clarity could be improved. I maintain my positive assessment of the paper.

---

> > > ### Author Response · Authors · 2025-08-05
> > >
> > > We sincerely appreciate your constructive feedback and positive assessment of our work. Thank you for your guidance on improving the discussion of the robustness–privacy–utility trilemma. In the revised version, we will incorporate the following enhancement into the Introduction (line 56):
> > >
> > > > However, privacy protection inevitably introduces tension with two other key objectives: model robustness against adversarial perturbations and statistical utility. This fundamental trade-off known as the privacy-robustness-utility trilemma has been extensively studied across different learning paradigms, including distributed learning ("On the Privacy-Robustness-Utility Trilemma in Distributed Learning", *ICML*, 2023), adversarial learning with certified guarantees ("Scalable differential privacy with certified robustness in adversarial learning", *ICML*, 2020), and decentralized Byzantine-robust systems ("On the tradeoff between privacy preservation and Byzantine-robustness in decentralized learning", *ICASSP*, 2024).
> > >
> > > We will also incorporate a comparative discussion of our method relative to existing literature in the Introduction (line 87):
> > >
> > > > In contrast to minimizer-optimal loss alignment approaches for addressing the privacy–robustness–utility trilemma, our method achieves inherent outlier robustness via a Huber loss framework.
> > >
> > > Additionally, we will improve the paper’s clarity by including a list of notation, a summary table of key theoretical results, and a reorganized problem formulation. Thank you again for your suggestions, they have been instrumental in enhancing both the rigor and presentation of our work.

---

### Official Review · Reviewer_x3FU · 2025-07-02

**Clarity:** 3
**Significance:** 3
**Originality:** 3
**Rating:** 4
**Confidence:** 3

**Summary:**

This paper addresses nonparametric regression on streaming data by introducing two algorithms:

H-FSGD: A functional stochastic gradient descent method that employs the Huber loss to mitigate heavy-tailed noise and outliers by truncating large residuals.

PH-FSGD: A locally differentially private variant of H-FSGD that injects carefully calibrated Gaussian process noise at each iteration, ensuring each data point is protected at the source without requiring a trusted curator.

**Questions:**

1. In Corollary 1 (Line 264), under a constant step size, the bound appears independent of the privacy parameters. Could the authors clarify this?

2. In Line 144, how is the closure of H measured or defined?

**Ethical Concerns:**

["NO or VERY MINOR ethics concerns only"]

**Final Justification:**

I think my concerns have been addressed, so I increase my score.

**Limitations:**

See above.

**Quality:**

3

**Strengths And Weaknesses:**

Strengths:

1. The manuscript is well written and clearly structured.

2. Incorporating the Huber loss is a sensible choice: it projects estimates into a finite-dimensional RKHS subspace, handles heavy-tailed noise and outliers effectively, and yields a uniformly bounded gradient—ideal for private algorithm design.

3. Both algorithms operate in a single pass, making them well suited for streaming data scenarios.

Weaknesses:

1. The notation is dense and could overwhelm readers; a summary table contrasting the theoretical guarantees for the non-private and private settings would improve clarity.

---

> ### Author Rebuttal · Authors · 2025-07-29
>
> Thank you for your thoughtful and critical assessment. Many of your comments will help us produce a more readable and self-contained version of the paper. Below, we address each of your specific concerns in turn.
>
> - **Clarification of notation and theoretical results.**
>
>   - Thank you for the valuable suggestion. We agree that a notation list and a summary table contrasting the theoretical guarantees for the non-private and private settings would enhance clarity and highlight our contributions. If accepted, we will include these in the revised manuscript to improve reader comprehension.
>
>   - The list of notation:
>     - $X_n$: the $n$-th copy of the random covariate $X$;
>     - $Y_n$: the $n$-th copy of the response $Y$;
>     - $e_n$: the $n$-th copy of the error $e$;
>     - $\sigma^2$: the variance of the error $e$;
>     - $f^\star$: the target nonparametric function lying in the space $L^2(P_X)$;
>     - $K$: the reproducing kernel;
>     - $\mathcal{H}$: a reproducing kernel Hilbert space (RKHS);
>     - $\langle \cdot, \cdot\rangle_{\mathcal{H}}$: the inner product in $\mathcal{H}$;
>     - $\bar{\mathcal{H}}$: the closure of $\mathcal{H}$;
>     - $f_{\mathcal{H}}$: the best approximation of $f^\star$ in the RKHS $\bar{\mathcal{H}}$;
>     - $L_{\tau}$: the Huber loss with the parameter $\tau$;
>     - $\widehat{\nabla L_{\tau}}$: the estimator of the Fréchet gradient of the Huber loss;
>     - $B^2$: the uniform upper bound of $K(x,x)$ for all $x$;
>     - $\gamma_n$: the step size at the $n$-th iteration;
>     - $\hat{f}\_n$: the current private estimator of $f_{\mathcal{H}}$ at the $n$-th iteration;
>     - $\bar{f}\_n$: the averaged private estimator of $f_{\mathcal{H}}$ at the $n$-th iteration;
>     - $\bar{f}\_n^0$: the averaged non-private estimator of $f_{\mathcal{H}}$ at the $n$-th iteration;
>     - $\varepsilon_n$, $\delta_n$: the privacy budget at the $n$-th iteration;
>     - $\\{z_j\\}_{j=1}^m$: the function grids;
>     - $\Sigma$:  the covariance operator associated with the kernel $K$, i.e., $\Sigma=\mathbb{E}(K_X\otimes K_X)$;
>     - $\alpha$: the parameter characterizing the decay rate of the eigenvalues of the covariance operator $\Sigma$;
>     - $r$: the parameter quantifying the regularity of the target function $f_{\mathcal{H}}$ with respect to the eigenbasis of $\Sigma$.
>
>   - Summary of theoretical results: we will add a table below contrasting the theoretical guarantees under private and non-private settings across different regimes of the decay rates $r$, covering both constant and non-constant step-size schemes.
>
>     - Constant step size:
>
>       Private setting: $O\left(\left(\sigma^2+\frac{8 \tau^2 B^2\log(2/\delta)}{\varepsilon^2}\right)\left(\gamma^{1/\alpha}n^{-1+1/\alpha}+n^{-1}\right)+(1+q(\gamma, n))\gamma^{-2r}n^{-2\min\\{r,1\\}}\left\\|\Sigma^{-r}f\_{\mathcal{H}}\right\\|\_{L\_{P\_X}^2}^2\right)$;
>
>       Non-private setting: $O\left(\sigma^2\left(\gamma^{1/\alpha}n^{-1+1/\alpha}+n^{-1}\right)+(1+q(\gamma, n))\gamma^{-2r}n^{-2\min\\{r,1\\}}\left\\|\Sigma^{-r}f\_{\mathcal{H}}\right\\|\_{L\_{P_X}^2}^2\right)$.
>
>       |           $r$ range           | Optimal $\zeta$ in $\gamma_i\asymp n^{-\zeta}$ |    Private / non-private convergence rate    |
>       | :---------------------------: | :--------------------------------------------: | :------------------------------------------: |
>       | $(0, (\alpha - 1)/(2\alpha)]$ |                      $0$                       |                 $O(n^{-2r})$                 |
>       |  $((\alpha-1)/(2\alpha) ,1]$  |       $(2r\alpha+1-\alpha)/(2r\alpha+1)$       |     $O(n^{-2r\alpha / (2r\alpha + 1)})$      |
>       |      $(1, (\alpha+2)/2]$      |           $(\alpha+1)/(2r\alpha+1)$            | $O(n^{-(2r\alpha - 2r + 2)/(2r\alpha + 1)})$ |
>       |   $((\alpha+2)/2, \infty)$    |                 $1/(1+\alpha)$                 |        $O(n^{-\alpha / (1+\alpha)})$         |
>
>     - Non-constant step size:
>
>       Private setting: for $0\leq\zeta\leq 1/2$, $O\left(\left(\sigma^2+\frac{8 \tau^2 B^2\log(2/\delta)}{\varepsilon^2}\right)\gamma_n^{1/\alpha}n^{-1+1/\alpha}+\gamma_n^{-1}(n\gamma_n)^{-2r}\left\\|\Sigma^{-r}f\_{\mathcal{H}}\right\\|\_{L\_{P_X}^2}^2\right);$
>
>       ​                           for $1/2<\zeta<1$, $O\left(\left(\sigma^2+\frac{8 \tau^2 B^2\log(2/\delta)}{\varepsilon^2}\right)(n\gamma_n)^{-2+1/\alpha}+\gamma_n^{-1}(n\gamma_n)^{-2r}\left\\|\Sigma^{-r}f\_{\mathcal{H}}\right\\|\_{L\_{P_X}^2}^2\right);$
>
>       Non-private setting: for $0\leq\zeta\leq 1/2$, $O\left(\sigma^2\gamma_n^{1/\alpha}n^{-1+1/\alpha}+\gamma_n^{-1}(n\gamma_n)^{-2r}\left\\|\Sigma^{-r}f\_{\mathcal{H}}\right\\|\_{L\_{P_X}^2}^2\right);$
>
>       ​                                    for $1/2<\zeta<1$, $O\left(\sigma^2(n\gamma_n)^{-2+1/\alpha}+\gamma_n^{-1}(n\gamma_n)^{-2r}\left\\|\Sigma^{-r}f\_{\mathcal{H}}\right\\|\_{L\_{P_X}^2}^2\right).$
>
>       |                   $r$ range                    | Optimal $\zeta$ in $\gamma_i\asymp n^{-\zeta}$ |          Private / non-private convergence rate           |
>       | :--------------------------------------------: | :--------------------------------------------: | :-------------------------------------------------------: |
>       |         $(0, (\alpha - 1)/(2\alpha)]$          |                      $0$                       |                       $O(n^{-2r})$                        |
>       | $((\alpha-1)/(2\alpha) ,(1+\alpha)/(2\alpha))$ |   $(2r\alpha+1-\alpha)/(2r\alpha+1+\alpha)$    | $O(n^{-(2r\alpha + \alpha - 1)/(2r\alpha + 1 + \alpha)})$ |
>       |        $[(1+\alpha)/(2\alpha), \infty)$        |                 $1/(1+\alpha)$                 |                $O(n^{-\alpha/(1+\alpha)})$                |
>
>     - In all, the privacy term introduces an additional constant factor of $(8\tau^2 B^2 \log(2/\delta))/(\varepsilon^2)$, which increases the variance constant but does not change the asymptotic rates under optimal step sizes.
>
> - **Role of the privacy parameters in Corollary 1.**
>
>   - We appreciate the reviewer’s careful reading. Corollary 1 is a direct consequence of Theorem 2, where the dependence on the privacy parameters is made explicit. In Corollary 1, we focus on the asymptotic order in terms of the sample size $n$, all privacy parameters ($\varepsilon$, $\delta$) are treated as constants; their contribution is absorbed into the $O(\cdot)$ notation. To clarify this point, we have now restored the explicit dependence on the privacy parameters in the convergence rates. Specifically, for different regimes of $r$, the convergence rate becomes:
>
>     - When $0<r\leq (\alpha-1)/(2\alpha)$, the optimal order is $O\left(n^{-2r}\right);$
>     - When $(\alpha-1)/(2\alpha)<r\leq 1$, the optimal order is $O\left(\left(\sigma^2+\frac{8 \tau^2 B^2\log(2/\delta)}{\varepsilon^2}\right)n^{-2r\alpha/(2r\alpha+1)}\right);$
>     - When $1<r\leq (\alpha+2)/2$, the optimal order is $O\left(\left(\sigma^2+\frac{8 \tau^2 B^2\log(2/\delta)}{\varepsilon^2}\right)n^{-(2r\alpha-2r+2)/(2r\alpha+1)}\right);$
>     - When $r>(\alpha+2)/2$, the optimal order is $O\left(\left(\sigma^2+\frac{8 \tau^2 B^2\log(2/\delta)}{\varepsilon^2}\right)n^{-\alpha/(1+\alpha)}\right)$.
>
>   We will incorporate this clarification into the revised manuscript to enhance transparency regarding the role of the privacy parameters.
>
> - **Definition of the closure of $\mathcal{H}$.**
>
>   - We thank the reviewer for pointing this out. As defined in the paper, $\mathcal{X}$ is the domain of the random covariate $X$ with marginal distribution $P_X$, $\mathcal{H}$ is a reproducing kernel Hilbert space on $\mathcal{X}$ with inner product $\langle \cdot, \cdot \rangle_{\mathcal{H}}$ and reproducing kernel $K(\cdot,\cdot): \mathcal{X} \times \mathcal{X} \to \mathbb{R}$, and $L^2(P_X)$ is the space of square-integrable functions with respect to $P_X$, i.e., $L^2(P_X) = \\{f: \mathcal{X} \rightarrow \mathbb{R} | \int\_{\mathcal{X}}f(x)^2dP_X(x) < \infty  \\}$. Then the closure of $\mathcal{H}$ in $L^2(P_X)$, denoted by $\bar{\mathcal{H}}$, is defined as the set of all limits of sequences in $\mathcal{H}$. Formally,
>
>     $$\bar{\mathcal{H}} = \\{ f \in L^2(P_X): \exists \\{f_k\\} \subset \mathcal{H} \text{ such that } \\|f_k - f\\|_{L^2(P_X)} \to 0 \\}.$$
>
>     Intuitively,  $\bar{\mathcal{H}}$ consists of all functions in $L^2(P_X)$ that can be approximated arbitrarily well (in $L^2(P_X)$ norm) by a sequence of functions in $\mathcal{H}$. This construction is standard in the kernel methods literature (see, e.g., "Nonparametric stochastic approximation with large step-sizes", *The Annals of Statistics*, 2016), and it allows us to avoid assuming that $f^\star$ belongs to $\mathcal{H}$ or that $\mathcal{H}$ is dense in $L^2(P_X)$. We will clarify this point in the revised version if the paper is accepted.
>
> Thank you again for your comments. We hope our responses resolve your concerns. If anything remains unclear, we’d be happy to provide further clarification.

---

> > ### Comment · Reviewer_x3FU · 2025-08-04
> > **My concerns have been addressed.**
> >
> > Thank you to the authors for the detailed responses. My concerns have been addressed, and I will increase my score.

---

> > > ### Author Response · Authors · 2025-08-05
> > >
> > > We’re glad our responses addressed your concerns. Thank you for your careful reading and thoughtful feedback, which significantly improved the paper’s clarity and completeness.

---

### Official Review · Reviewer_nvq6 · 2025-07-02

**Clarity:** 2
**Significance:** 2
**Originality:** 2
**Rating:** 3
**Confidence:** 3

**Summary:**

The authors address the problem of online nonparametric regression and propose an algorithm that is both outlier-robust and locally differentially private. The key novelty lies in the use of the Huber loss to enhance robustness against outliers, combined with the addition of Gaussian process noise to ensure local differential privacy. The paper provides non-asymptotic theoretical guarantees for the proposed method and validates the results with synthetic numerical experiments.

**Questions:**

Please see strengths and weaknesses.

**Ethical Concerns:**

["NO or VERY MINOR ethics concerns only"]

**Final Justification:**

After reading the rebuttal and subsequent discussion, I have decided to maintain my current score. In my opinion, the paper needs a major restructuring and remains difficult to parse in its current state. The authors have promised to improve the clarity of the paper, but I believe that would require major changes in the current presentation, and should ideally go through another round of fresh reviews for a better assessment.

The authors have also included some additional guarantees for the robustness part. I have not verified these results. In any case, I believe these would be a good addition in a fresh version of the paper.

**Limitations:**

yes

**Paper Formatting Concerns:**

-

**Quality:**

2

**Strengths And Weaknesses:**

The paper addresses a compelling problem that is likely to interest the machine learning community, particularly those focused on streaming data, robustness, and privacy. Building on functional SGD methods for the squared loss [Liu et al., 2023], the authors introduce the Huber loss to enhance robustness and incorporate Gaussian process noise to achieve local differential privacy.

In my opinion, the paper is difficult to read due to heavy notation and a lack of a clear formal problem setup. A more structured presentation would help place the contributions in a better context and improve overall readability.

1. Could the authors provide formal definitions of outlier robustness and local differential privacy (LDP) in the context of this work? While the theorems address properties related to LDP, the main text does not clearly define it. Including formal definitions would help clarify the contributions and make the theoretical results more accessible.
2. The paper does not appear to provide any formal guarantees regarding outlier robustness. Interestingly, the choice of $\tau$ also influences the privacy guarantees through the term $8 \tau^2 B^2 \log (2/\delta)/\varepsilon^2$. Suppose one does not have any concern about outliers, can we keep $\tau$ to $\infty$ and essentially work with the squared loss? How does this affect the privacy guarantees?
3. Similarly, if privacy is not a concern, then one does not see any direct dependency on $\tau$ in Corollary 3. How does the choice of $\tau$ affect the guarantees in Corollary 3?
4. In my opinion, Lemma 1 should be included in the main text, as it appears to play a central role in supporting the main results.
5. Related to point 1, could the authors clarify how Proposition 1, which (as far as I understand) concerns central differential privacy, is used in the proof of Theorem 1, which provides guarantees for local differential privacy?
6. The step size depends on hyperparameters $(B, M, \kappa)$, which may not be readily available in practice. Could the authors clarify how the step size should be selected to ensure convergence when these parameters are unknown?

---

> ### Author Rebuttal · Authors · 2025-07-29
>
> Thank you for your thoughtful and critical assessment. Many of your comments will help us produce a more readable and self-contained version of the paper. Below, we address each of your specific concerns in turn.
>
> - **Clarification of notation and problem setup.** We agree that the current notation may hinder readability. Due to space constraints, we refer the reviewer to the notation table in our response to Reviewer x3FU, and a concise version will be included in the revision. While the formal setup appears in Section 2, we will reorganize and clarify its structure to better contextualize our contributions and improve clarity.
>
> - **Formal definitions.**
>   - **Outlier robustness.**  In our work, outlier robustness refers to the estimator’s stability and accuracy under contaminated or heavy-tailed data. Classical examples include Huber's M-estimators, which cap large residuals to balance efficiency and robustness ("Robust estimation of a location parameter", *The Annals of Mathematical Statistics*, 1964). Recent results (e.g., "Concentration study of M‑estimators using the influence function", *Electronic Journal of Statistics*, 2022) further establish their non-asymptotic stability under adversarial contamination, though sensitivity to high-leverage points remains due to unbounded structural influence. We will clarify this in the revised version.
>
>   - **Local differential privacy (LDP).** We agree that including key definitions such as LDP in the main text is important for clarity. Due to space constraints, the formal definition currently appears in Appendix A.2 (Definition 3). If accepted, we will move it to the main text to improve readability. For completeness, we restate the definition below.
>
>     **Definition3** (LDP, "A comprehensive survey on local differential privacy", *Security and Communication Networks*, 2020). Let $\varepsilon>0$ and $\delta\geq 0$. A randomized algorithm $M: \mathcal{X} \rightarrow \mathcal{Y}$ is said to be $(\varepsilon,\delta)$-LDP if for any pair of input individual values $x, x^\prime\in\mathcal{X}$, and any measurable set $E\subset \mathcal{Y}$,
>     $$\mathbb{P}(M(x)\in E)\leq e^{\varepsilon}\mathbb{P}(M(x^\prime)\in E)+\delta,$$
>     where the probabilities are computed over the randomness of the mechanism $M$.
>
> - **Robustness and privacy guarantee without outlier concern.**
>   - Our use of the Huber loss is motivated by its well-established robustness, balancing squared loss sensitivity for small residuals with absolute loss stability for large outliers. While we omit formal robustness guarantees here, its strong theoretical backing, from Huber’s original minimax analysis ("Robust estimation of a location parameter", *The Annals of Mathematical Statistics*, 1964) to modern concentration bounds (e.g., "Concentration study of M‑estimators using the influence function", *Electronic Journal of Statistics*, 2022), supports its effectiveness. We will elaborate on this rationale in the revised version.
>
>   - When outliers are not a concern ($\tau \to \infty$), the Huber loss reduces to the squared loss. In the DP setting, the privacy guarantee depend on gradient sensitivity. Without additional assumptions, it is standard practice to apply gradient clipping to ensure bounded sensitivity under the squared loss (see, e.g., "Evading the curse of dimensionality in unconstrained private glms", *AISTATS*, 2021). If the response variable $Y$ is further assumed to be bounded, then the gradient sensitivity is naturally finite, and the privacy guarantee directly depends on its magnitude. Thus, squared loss remains privacy-compatible with proper sensitivity control. We will clarify this connection in revision.
>
> - **Role of $\tau$ on the upper bound in Corollary 3.**
>
>   - When privacy is not a concern, the parameter $\tau$ does not appear explicitly in Corollary 3, as it is absorbed into the constant $c$ defined in Lemma 1 and further into the  $O(\cdot)$ notation. However, $\tau$ implicitly influences the bound via its role in defining constant $c$.
>
>   - To make this dependence more transparent, we provide a modified version of Corollary 3 in which $\tau$ is not absorbed into the $O(\cdot)$ notation.
>
>     **Corollary 3'** (Constant step size, without privacy).
>
>     Suppose Assumptions 1-7 hold. Consider the estimator $\bar{f}_n^0$ without privacy protection defined via the recursion (5). Take any constant choice $\gamma_i=\gamma=\Gamma(n)$, for $1\leq i\leq n$. If $c^{-1}\gamma B^2<1$ with $c$ defined in Lemma 1, i.e., $c:=c(\tau)=2m\kappa\min\left\\{1, \frac{\tau}{m+BM+\\|f^{\star}\\|\_{\infty}}\right\\}$, then $$\mathbb{E}\\|\bar{f}_n^0-f_{\mathcal{H}}\\|_{L_{P_X}^2}^2\leq O\left(\frac{c^{-2}\sigma^2}{(1-(c^{-1}\gamma B^2)^{1/2})^2}\left(c^{1/\alpha}\gamma^{1/\alpha}n^{-1+1/\alpha}+n^{-1}\right)+(1+q(c,\gamma, n))c^{-2r}\gamma^{-2r}n^{-2\min\\{r,1\\}}\\|\Sigma^{-r}f_{\mathcal{H}}\\|_{L_{P_X}^2}^2\right),$$ where $q(c,\gamma, n)=0$ for $r\leq 1/2$, and $q(c,\gamma, n)=\left(\frac{B^2}{2c-\gamma B^2}\right)^{2r-1}c^{(2r-1)/\alpha}\gamma^{(1+\alpha)(2r-1)/\alpha}n^{(2r-1)/\alpha}$ for $r>1/2$.
>
>   - The parameter $\tau$, which is embedded in $c$, establishes a trade-off between robustness and bias in the estimation, which is consistent with classical literature (see, e.g., "Estimation of high dimensional mean regression in the absence of symmetry and light tail assumptions", *Journal of the Royal Statistical Society, Series B*, 2017). In practice, we first compute an L2-FSGD estimate and residuals, then estimate the noise scale via median absolute deviation (MAD). The robustness parameter $\tau$ is set to $1.345\times \mbox{MAD}$, ensuring $90\\%$ efficiency under normal errors while resisting outliers, following standard robust regression practice (Lines 191-193, Algorithm 2). This builds on classical Huber loss methods (see, e.g., "Robust regression using iteratively reweighted least-squares", *Communications in Statistics-theory and Methods*, 1977). We will incorporate these clarifications in the revised revision.
>
> - **Placement of Lemma 1.** We agree that Lemma 1 plays a central role in supporting our main results. If the paper is accepted, we will include it in the main text to improve clarity and accessibility.
>
> - **Clarification on the use of Proposition 1 in the proof of Theorem 1.** You are right that Proposition 1 concerns central differential privacy (CDP). However, CDP reduces to local differential privacy (LDP) in the single-sample setting. In our proof of Theorem 1, we apply Proposition 1 with single sample $D = \\{(X_n, Y_n)\\}$, effectively leveraging CDP results to establish LDP guarantees.
>
> - **Selection of step size.**
>
>   - Theoretically, the choice of step size is indeed crucial for the convergence rate of the proposed estimators, both under constant and non-constant step size settings. Corollaries 1-4 demonstrate that, depending on the range of $r$, different choices of $\zeta$ are required to attain the optimal convergence rate.
>
>   - In practice, when the hyperparameters involved in the theoretical step size are unavailable, we recommend selecting the step size using a decaying schedule of the form $\gamma_i = \gamma_0 n^{-\zeta}$ (for constant step size and the total sample size $n$) or $\gamma_i = \gamma_0 i^{-\zeta}$ (for non-constant step size). The parameters $\gamma_0$ and $\zeta$ can be tuned via grid search on a validation set. This approach is widely adopted in practice and shown to be effective across various optimization settings (see, e.g., "The step decay schedule: a near optimal, geometrically decaying learning rate procedure for least squares", *NeurIPS*, 2019; "Benefits of learning rate annealing for tuning‑robustness in stochastic optimization", *arxiv*, 2025).
>
>   - To address practical concerns, we have conducted additional simulation studies to examine the sensitivity of estimation performance with respect to different values of step size. We examine the H-FSGD performance under Case 1 for both constant and non-constant step size settings, with $\gamma_0 \in [4, 24]$ and $\zeta \in [0.3, 0.8]$. The tables below report the average MSE ($\times 10^{-3}$) over 50 repetitions when $n = 10000$.
>
>     - Constant step size setting:
>
>       | $\zeta$ \ $\gamma_0$ |   4   |   8   |  12  |  16  |  20  |  24  |
>       | :------------------: | :---: | :---: | :--: | :--: | :--: | :--: |
>       |       **0.3**        | 1.52  | 1.76  | 2.05 | 2.38 | 2.79 | 3.32 |
>       |       **0.4**        | 1.59  | 1.49  | 1.56 | 1.65 | 1.76 | 1.87 |
>       |       **0.5**        | 2.06  | 1.67  | 1.54 | 1.50 | 1.49 | 1.51 |
>       |       **0.6**        | 4.03  | 2.30  | 1.93 | 1.76 | 1.67 | 1.60 |
>       |       **0.7**        | 12.28 | 5.18  | 3.37 | 2.66 | 2.31 | 2.11 |
>       |       **0.8**        | 59.14 | 17.12 | 9.69 | 6.77 | 5.21 | 4.25 |
>
>     - Non-constant step size setting:
>
>       | $\zeta$ \ $\gamma_0$ |  4   |  8   |  12  |  16  |  20  |  24  |
>       | :------------------: | :--: | :--: | :--: | :--: | :--: | :--: |
>       |       **0.3**        | 1.63 | 1.99 | 2.44 | 3.04 | 4.13 | 5.12 |
>       |       **0.4**        | 1.51 | 1.62 | 1.77 | 1.96 | 2.13 | 2.33 |
>       |       **0.5**        | 1.69 | 1.59 | 1.63 | 1.66 | 1.80 | 1.85 |
>       |       **0.6**        | 2.27 | 1.94 | 1.92 | 1.84 | 1.87 | 1.93 |
>       |       **0.7**        | 3.59 | 3.14 | 2.95 | 2.85 | 2.68 | 2.63 |
>       |       **0.8**        | 7.74 | 6.27 | 6.50 | 6.46 | 6.58 | 6.39 |
>
>   - Our results indicate that the proposed estimators exhibit robustness to the choice of $\gamma_0$ and $\zeta$ under both constant and non-constant step size settings, with wide parameter regions yielding stable and comparable MSEs. If the paper is accepted, we will clarify the step size selection procedure and provide a sensitivity analysis.
>
> Thank you again for your comments. We hope our responses resolve your concerns. If anything remains unclear, we’d be happy to provide further clarification.

---

### Official Review · Reviewer_euKv · 2025-07-03

**Clarity:** 3
**Significance:** 3
**Originality:** 3
**Rating:** 5
**Confidence:** 4

**Summary:**

The title says it all: this paper proposes an online learning method for solving nonparametric regression problems under the constraint of local differential privacy (LDP). The algorithm is stochastic gradient descent (SGD) on the reproducing kernel Hilbert space (RKHS) $\mathcal{H}$. The main contributions are:

  * finite sample analysis (theoretical)
  * optimizing convergence rates over the hyperparameters (theoretical)
  * experiments (empirical)

There are two key aspects of the problem which motivates their solution concept:

1. They do not assume the true regressor $f^* \in \mathcal{H}$ (i.e. the problem is misspecified).
2. They want the method to have good empirical performance when the noise is heavy-tailed.

The main idea is to use the Huber loss to get some robustness against the heavier noise. This is a standard approach (as they point out themselves). The benefit of this loss is that the gradient becomes bounded, avoiding the clipping needed for DP-SGD for quadratic losses like in ridge regression.

**Questions:**

**Main Question 1: Does the convergence analysis really take into account heavy-tailed noise?**

* In Section 2, the error is zero-mean but has finite bounded variance $\sigma^2$. While $\sigma^2$ could be large, assuming it is finite means that whole classes of two-tailed distributions are out, such as $\alpha$-stable distributions for $\alpha \ne 2$ or $t$-distributions with $\nu \le 2$.

* In the experiments in the submitted manuscript (the supplement has code only as far as I could see) the authors use Gaussian noise with only one value of the variance ($0.25$) and the Student-$t$ distribution with $\nu = 3$ so the variance is  $\frac{3}{3-1} = 1.5$. If I am misunderstanding, I am happy to be corrected! However, these are hardly the "extensive experiments" promised in the abstract. Obviously, space is limited, but I would have expected a better exploration of how robust the method is to actual heavy tailed noise.

* Considering this I think the claims that this method is itself robust is misleading: the loss function is chosen for its robustness properties, but neither the analysis nor the experiments properly take into account real heavy tails.

**Main Question 2: How is this algorithm supposed to be run?** This is a bit of a disconnect between the idea of the algorithm and its implementation. Since empirically everything is computed on the grid of $m$ points, everything turns into an $m$-dimensional vector gradient problem. Of course, the underlying structure comes from the RKHS assumption.

* Figure 1 suggests that for each individual, they either need privacy or do not need privacy so each gradient can be computed with or without noise. However, the experiments seem to only do private or nonprivate. Is the setting under consideration really reflected in Figure 1?

* The privacy piece is "turnkey": just add Gaussian noise at each iteration according to the gradient sensitivity. This is the same SGD algorithm which has been studied for 10+ years now. So, the novelty is in the analysis of the functional gradient. The grid determines the covariance of the noise, but how should we choose the grid?

* Because the analysis/theory is done in the RKHS, the grid size doesn't appear in the finite sample bounds. Commenting on the computational complexity in terms of the sample size alone also erases the impact of the grid: is there a way to incorporate it?

**Main Question 3: Is this really about privacy at all?** In the end, this is just a noisy gradient algorithm for nonparametric regression. The privacy "story" is an OK motivation, but the experiments are on synthetic data and have no bearing on problems involving actual privacy considerations.

* Under LDP there is no need to do privacy accounting so much of the "privacy apparatus" that has been developed over the years does not apply here.

* If minibatching were allowed then perhaps we would see some more relevance for privacy. Is there a way to incorporate minibatching into this framework?

* Several applications are mentioned in the introduction, but is there a vignette that the authors could provide in which someone has a privacy constraint, is worried about heavy-tailed noise, and has to use a nonparametric regression model (i.e. that's the right model for this kind of data)?

**Other comments:**

* There is some notation clash in using $z$ for the pair $(x,y)$ and a grid point (which is only in the $\mathcal{X}$ space, right?)

* Shouldn't the kernel function be an input to the algorithm as well?

**Ethical Concerns:**

["NO or VERY MINOR ethics concerns only"]

**Limitations:**

Yes.

**Paper Formatting Concerns:**

N/A.

**Quality:**

3

**Strengths And Weaknesses:**

**Strengths:** I am not able to evaluate the novelty of the theory w.r.t. the literature on nonparametric regression and online learning.

* The problem is straightforward to state, mathematically.
* The analysis, although quite complex, is interesting.

**Weaknesses:**

* It's not clear why this combination of constraints/properties would be needed in any particular application. In particular, the privacy piece seems almost incidental.
* The experimental evaluation is quite cursory and doesn't explore the parameter space adequately.

---

> ### Author Rebuttal · Authors · 2025-07-29
>
> Thank you for your thoughtful and critical assessment. Many of your comments will help us produce a more readable and self-contained version of the paper. Below, we address each of your specific concerns in turn.
>
> - **Heavy-tailed noise in convergence analysis.**
>
>   - Our theoretical analysis requires finite-variance noise for tractable proof techniques, excluding some extreme heavy-tailed cases. However, it still covers common heavy-tailed distributions like student-$t$ distributions with degrees of freedom $\nu > 2$, Laplace distribution, and symmetric Pareto distributions with shape parameter $\alpha>2$.
>
>   - While our main manuscript focuses on Gaussian and $t(3)$ noise, we have conducted additional experiments with heavier-tailed noise, including Student-$t(2.5)$ and $\text{Cauchy}(0,1)$ noises. The tables below show MSE (×$10^{-3}$) means and standard deviations over 200 repetitions for $n=10000, 20000$, and $40000$, under both constant and non-constant step sizes.
>
>     - $t(2.5)$ noise:
>
>       |                  |   Constant   |              |               | Non-constant |              |               |
>       | :--------------: | :----------: | :----------: | :-----------: | :----------: | :----------: | :-----------: |
>       | **Method \ $n$** |  **10000**   |  **20000**   |   **40000**   |  **10000**   |  **20000**   |   **40000**   |
>       |    **H-FSGD**    | 2.15 (0.758) | 1.25 (0.415) | 0.669 (0.214) | 2.26 (1.06)  | 1.41 (0.682) | 0.907 (0.467) |
>       |   **L2-FSGD**    | 3.97 (2.25)  |  2.04 (1.20)  | 1.11 (0.551)  |  6.13 (4.40)  | 3.38 (2.89)  |  2.02 (1.71)  |
>       |   **Offline**    | 4.21 (2.17)  | 2.17 (1.26)  | 1.16 (0.569)  | 6.75 (5.49)  | 3.65 (3.25)  |  2.12 (1.85)  |
>
>     - $\text{Cauchy}(0,1)$ noise:
>
>       |                  |      Constant       |                     |                     |    Non-constant     |                     |                     |
>       | :--------------: | :-----------------: | :-----------------: | :-----------------: | :-----------------: | :-----------------: | :-----------------: |
>       | **Method \ $n$** |      **10000**      |      **20000**      |      **40000**      |      **10000**      |      **20000**      |      **40000**      |
>       |    **H-FSGD**    |     3.64 (1.57)     |    1.97 (0.816)     |      1.00 (0.435)      |     5.22 (3.25)     |     2.98 (1.81)     |     1.65 (1.03)     |
>       |   **L2-FSGD**    | 5.83e+07 (8.2e+08)  | 1.89e+07 (2.65e+08) | 5.7e+06 (7.89e+07)  | 2.24e+08 (3.17e+09) | 1.01e+08 (1.42e+09) | 4.36e+07 (6.15e+08) |
>       |   **Offline**    | 1.73e+08 (2.45e+09) | 4.86e+07 (6.85e+08) | 1.33e+07 (1.86e+08) | 3.67e+08 (5.19e+09) | 1.38e+08 (1.95e+09) | 5.32e+07 (7.51e+08) |
>
>   - The results show H-PSGD's superior empirical performance over L2-FSGD and offline methods even under extreme heavy-tailed noise. While lacking theoretical guarantees for infinite-variance settings, this robustness motivates future theoretical investigation.
>
> - **Implementation of the algorithm.**
>
>     - **Mixed privacy.** Algorithm 1 (PH-FSGD) supports mixed privacy regimes through iteration-specific privacy budgets. Large $\varepsilon_n$ values effectively imply non-private updates. While our current experiments focus on fully private or non-private settings, the algorithm naturally accommodates hybrid cases by adjusting per-iteration noise. We will clarify this in the revision.
>
>   - Our method employs functional SGD, which differs fundamentally from finite-dimensional SGD by optimizing over infinite-dimensional function spaces  ("Online learning with kernels", *IEEE transactions on signal processing*, 2004). This involves Fréchet derivatives, RKHS-based analysis, and more complex convergence proofs (see, e.g., "Nonparametric stochastic approximation with large step-sizes", *The Annals of Statistics*, 2016). Our key contribution is a theoretically grounded private functional estimator with convergence guarantees for both step size schemes.
>
>   - **Choice of grids.** We clarify that our method and theory are developed in the infinite-dimensional RKHS, while the grid-based implementation is a discretized approximation for computational feasibility. The current bounds do not reflect the effect of grid size $m$, as the analysis assumes sufficiently fine discretization to approximate the continuous function. We acknowledge that small $m$ may introduce non-negligible error. In practice, a dense grid is needed to recover the function accurately, as also done in prior work (e.g., "Scalable statistical inference in non-parametric least squares", *arxiv*, 2023), which similarly builds theory in function space but uses discrete grids for implementation. Extending the theory to account for finite grid is left for future work.
>
>   - **Computational complexity.** Our theoretical analysis operates directly in RKHS, with finite-sample bounds depending on kernel eigenvalue decay rate $\alpha$ rather than grid size $m$. The grid size $m$ affects the computational costs, with $O(m^3+nm^2)$ total runtime and $O(m^2)$ storage. This reflects an accuracy-efficiency tradeoff that we will clarify in revision.
>
> - **Practical applicability.**
>   - Our method is grounded in LDP, which avoids the need for trusted data collectors or centralized privacy accounting, unlike CDP-based methods. With disjoint data access at each iteration, our algorithm leverages parallel composition to provide rigorous per-user privacy protection, particularly valuable in streaming or decentralized environments.
>
>   - **Minibatching integration.** Our framework can be easily extended to support minibatching. Specifically, at iteration $n$, if a mini-batch $\\{(X_{nt}, Y_{nt})\\}_{t=1}^{B_n}$ with size $B_n$ is available, the update rule becomes: $$\hat{f}_n=\hat{f}\_{n-1}+\gamma_n \frac{1}{B_n}\sum\_{t=1}^{B_n}w\_{\tau}\left(Y\_{nt}-\langle \hat{f}\_{n-1},K\_{X\_{nt}}\rangle\_{\mathcal{H}}\right)\left(Y\_{nt}-\langle \hat{f}\_{n-1},K\_{X\_{nt}}\rangle\_{\mathcal{H}}\right)K\_{X\_{nt}}+\gamma_n \xi_n,$$ $$\bar{f}_n=\frac{n-1}{n}\bar{f}\_{n-1}+\frac{1}{n}\hat{f}\_n,$$
>     where $\xi_n$ is the sample path of a Gaussian process having mean zero and covariance function $\frac{8 \tau^2 B^2\log(2/\delta_n)}{\varepsilon_n^2}K$ with $(\varepsilon_n, \delta_n)$ being the privacy budget at the $n$-th iteration. The noise term $\xi_n$ still suffices to ensure differential privacy, as the modified Fréchet gradient has the same sensitivity and the privacy mechanism remains applicable in this case. Moreover, the theoretical analysis can be readily modified by replacing the single-sample gradient with its minibatch-averaged counterpart. Minibatching may in fact help improve the privacy–utility trade-off by reducing gradient variance, allowing for smaller noise magnitudes under the same privacy budget; however, it also requires a trusted data collector to aggregate and access the mini-batch data. We will discuss the incorporation of minibatching in the revised version.
>
>   - **Applications.** Many real-world applications simultaneously require privacy protection, robustness to heavy-tailed noise, and nonparametric modeling. Examples include:
>
>     - Healthcare analytics. Wearable fitness data (e.g., heart rate, sleep patterns) demands LDP since raw physiological signals can re-identify users; exhibits heavy-tailed noise due to irregular activities (e.g., sudden spikes in heart rate during exercise or sensor artifacts); and necessitates nonparametric regression because the relationship between metrics (e.g., sleep duration vs. recovery rate) is often nonlinear and complex.
>     - Financial fraud detection. Transaction histories require LDP when shared due to their identifiable nature, legitimate spending patterns contaminated by fraudulent outliers create heavy-tailed distributions, and the adversarial nature of fraud evolution necessitates nonparametric methods to detect novel attack patterns beyond rigid rule-based systems.
>     - Consumer behavior analysis. Browsing logs need LDP protection against profiling, purchase amounts exhibit heavy-tailed distributions dominated by rare large orders, and market segmentation reveals irregular price elasticity patterns that nonparametric models can adequately capture.
>
>   - To better validate our method's practical utility, we conducted experiments on the real data "Health and fitness dataset" from Kaggle website. Our goal is to investigate how endurance levels affect overall fitness. We select 40000 samples as the training set and 1000 samples as the test set. We compared our H-FSGD and PH-FSGD methods with baseline methods on out-of-sample $R^2$ performance, as displayed in the table below.
>
>      | Step size \ Method | H-FSGD | L2-FSGD | Offline | (2,0.2)-LDP | (3,0.1)-LDP |  lm   |
>     | :----------------: | :----: | :-----: | :-----: | :---------: | :---------: | :---: |
>     |    **Constant**    | 0.622  |  0.617  |  0.609  |    0.589    |    0.609    | 0.581 |
>     |  **Non-constant**  | 0.623  |  0.618  |  0.609  |    0.526    |    0.596    | 0.581 |
>
>     The results show that H-FSGD consistently outperforms L2-FSGD and Offline methods, showing strong robustness. All non-private nonparametric methods outperform linear regression (lm) in capturing complex relationships. Under LDP, PH-FSGD remains competitive, matching or exceeding lm, confirming its effectiveness under privacy constraints. These findings validate our method’s ability to model nonlinear patterns while preserving privacy. We will include additional practical application details and real-world data analysis in the revision.
>
> - **Notation and presentation.**
>
>   - We will revise the notation to eliminate conflicts, include the kernel function as an explicit algorithm input, and review the presentation for clarity and consistency in the revision.
>
> Thank you again for your comments. We hope our responses resolve your concerns. If anything remains unclear, we’d be happy to provide further clarification.

---

> > ### Comment · Reviewer_euKv · 2025-08-03
> > **Thank you for your responses**
> >
> > Thanks for the detailed clarifications. I would like to know how you will change the description of your method and results (e.g. in the introduction). Because your theoretical analysis does not hold for infinite variance settings I think it's best to make a clean distinction between the theoretical guarantees and the experimental evaluation. Experiments showing that the method works beyond the theoretical limits can then be appreciated better (assuming you will include them).
> >
> > This also manifests in the grid discussion: the fact that the theory optimizes directly over the RKHS but the computational method requires this grid is not so clear in the manuscript.
> >
> > The computational complexity is nontrivial: the grid would involve an exponential blowup in complexity if one tried to do this for multivariate problems, right?

---

> > > ### Author Response · Authors · 2025-08-04
> > >
> > > Many thanks to the reviewer’s constructive suggestions to clarify the distinction between our theoretical and empirical contributions. We address each point below.
> > >
> > > - **Theory-experiment boundary clarification.** We fully agree with the reviewer's suggestion to clearly delineate the boundaries between our theoretical guarantees and empirical findings. In the revision, we will implement the following clarifications in two key sections:
> > >
> > >   - "Problem formulation" section: we will significantly expand the discussion of the finite-variance noise assumption and include the following discussion. Specifically, our theoretical analysis requires finite-variance noise assumption to establish tractable bounds on gradient variances, but our method can still demonstrate remarkable empirical robustness even in infinite-variance scenarios like Cauchy noise.
> > >   - "Additional experimental results" section: we will add a subsection titled "Performance beyond theoretical assumptions" that presents comprehensive results under infinite-variance noise (e.g., Cauchy) and include the following discussion. Our experimental results demonstrate that H-PSGD maintains superior robustness compared to both L2-FSGD and offline methods, even when handling infinite-variance distributions that fall outside our current theoretical framework. While formal guarantees for these extreme cases remain to be established, the observed empirical performance strongly motivates future theoretical investigation of such challenging scenarios.
> > >
> > >
> > > - **Theoretical RKHS analysis and practical grid approximation.** We thank the reviewer for highlighting the need to clarify the relationship between our theoretical RKHS framework and practical grid-based implementation. In the revised version, we will implement the following clarifications in two key parts:
> > >
> > >   - The explanation part of Algorithm 1: while our method is developed in the infinite-dimensional RKHS, the computational implementation employs grid discretization as a finite approximation, which is a standard approach in functional data analysis to balance computational feasibility with theoretical fidelity. Following established practice (e.g., "Nonparametric stochastic approximation with large step-sizes", *The Annals of Statistics*, 2016; "Scalable statistical inference in non-parametric least squares", *arxiv*, 2023), we use dense grids to ensure accurate function recovery, with approximation error diminishing as grid density increases.
> > >   - The discussion part of Theorem 2: our current theoretical bounds assume sufficiently fine discretization to guarantee the grid-based solution closely approximates the RKHS optimum. While small $m$ may introduce non-negligible error, our empirical results confirm that larger $m$ effectively mitigates this discrepancy. Extending the theory to explicitly account for finite grid effects remains an important direction for future work.
> > >
> > >
> > > - **Computational complexity in multivariate settings.** We thank the reviewer for highlighting this important computational consideration. While our current analysis focuses on univariate settings with tractable complexity, requiring $O(m^3)$ setup time for covariance matrix construction and Cholesky decomposition followed by $O(nm^2)$ operational cost for $n$ functional SGD iterations with $O(m^2)$ storage, we fully acknowledge the exponential scaling in multivariate cases. Specifically, $d$-dimensional problems would require $O(m^{3d})$ setup and $O(nm^{2d})$ operational costs with $O(m^{2d})$ storage, making naive grid-based approaches impractical for $d\geq 3$. To address this, additive kernel methods ("Minimax-optimal rates for sparse additive models over kernel classes via convex programming", *Journal of Machine Learning Research*, 2012) offer a viable alternative, reducing complexity to $O(dm^3 + ndm^2)$ and storage to $O(dm^2)$ through dimension-wise decomposition. This preserves theoretical guarantees while maintaining computational feasibility. We will emphasize this crucial point in the discussion of the revision.
> > >
> > > Thank you again for your time and insightful feedback!

---

### Note · Authors · 2025-08-12

We thank the reviewers, ACs, SACs, and PCs for their time, constructive feedback, and thoughtful discussions. We are encouraged by the positive assessments of our contributions, summarized below.

- **Significance and novelty.** We present the first unified framework for online robust differentially private nonparametric learning in RKHS, robust to heavy-tailed noise, with formal privacy guarantees and statistical efficiency in streaming settings.

- **Theoretical investigation.** We give rigorous finite-sample guarantees without assuming the true function lies in the chosen RKHS, and provide detailed non-asymptotic convergence-rate analyses—both with and without LDP—across various smoothness regimes and step-size schemes. Reviewers noted that, despite its complexity, the analysis is comprehensive and insightful.

- **Methodological soundness and clarity.** Several reviewers agreed that the Huber loss is a principled choice, offering statistical robustness and uniformly bounded gradients for privacy analysis. Our algorithm supports one-pass updates in constant time and memory, ensuring scalability for streaming data. The paper’s structure, along with its comparison table and flowchart, was also praised for clarity.

We acknowledge the main concern shared by most reviewers—that the heavy notation may challenge readers. In the revision, we will add a notation table, a concise summary of key theoretical results, and a reorganized problem formulation (see our response to Reviewer x3FU for details).

We also addressed specific points raised by Reviewer nvq6:

- **Formal outlier-robustness guarantees.** We provided a precise definition and rigorous proof that our Huber-based procedure satisfies outlier robustness, which will be included in the revision.

- **Role of Huber vs. gradient clipping.**  The Huber loss serves primarily for robust estimation, with its DP compatibility being a beneficial by-product, rather than merely acting as a gradient clipping mechanism.

Finally, we appreciate the recognition of this problem’s importance and potential impact. We believe the combination of practical relevance and theoretical contributions makes this work a valuable addition to the literature, and we hope it inspires further research in robust, private, online learning.

---

### Decision · Program_Chairs · 2025-09-17

**Decision:**

Accept (poster)

**Comment:**

This paper proposes methods for online nonparametric regression that are robust to outliers, including one that satisfies local differential privacy. The reviewers generally found the results compelling, believed the problem to be important and fundamental, and thought the analysis was high-quality and interesting. These qualities all argue in favor of acceptance at NeurIPS.

While the authors are encouraged to address all reviewer comments for the final version (and of course to make the revisions promised in their responses), they should especially focus on the clarity of the notation, which was a concern for several reviewers (as acknowledged by the authors in their final remarks).